# ReJump: A Tree-Jump Representation for Analyzing and Improving LLM Reasoning

## Abstract

Large Reasoning Models (LRMs) are Large Language Models (LLMs) explicitly trained to generate long-form Chain-of-Thoughts (CoTs), achieving impressive success on challenging tasks like math and programming. However, their underlying reasoning "algorithms" remain poorly understood. To investigate this, we propose *ReJump*, which represents a reasoning trace as a visitation order over nodes in a tree of intermediate problem-solving steps. Transitions between nodes, which we term *jumps*, include adjacent moves that capture behaviors such as calculation, and non-adjacent moves that capture behaviors such as backtracking and verification. ReJump enables analyzing LLM reasoning with diverse metrics that quantify exploration, exploitation, overthinking, forgetting, and verification. Using our proposed LLM agent to extract reasoning traces into ReJump format, we evaluate state-of-the-art LRMs on two tasks and find that models with similar accuracy can exhibit distinct reasoning behaviors, while different tasks favor different reasoning styles (e.g., varying balance between exploration and exploitation). To further understand how learning strategies shape reasoning, we use ReJump to compare distilled LRMs with their teachers, CoT-prompted LLMs with LRMs, and to examine how the number of reasoning examples and reinforcement learning affect reasoning behavior. Finally, we show that ReJump can improve reasoning quality at test time through strategies such as ReJump-guided Best-of-N selection and prompt selection.

## 1 Introduction

Chain-of-Thought (CoT) prompting improves the performance of Large Language Models (LLMs) on complex tasks, such as mathematical problem solving. This was achieved either by providing exemplars of step-by-step reasoning (Wei et al., 2022) or by simply adding the instruction "Let's think step by step" to the prompt (Kojima et al., 2022), which encourages the model to decompose problems into intermediate steps, yielding more accurate and interpretable outputs. Recent work goes further by internalizing multi-step reasoning through supervised fine-tuning or reinforcement learning, leading to the recent flourishing of Large Reasoning Models (LRMs), a class of LLMs explicitly trained to generate long-form CoT, such as DeepSeek-R1 (Guo et al., 2025), o1 (Jaech et al., 2024), and QwQ-32B (Qwen Team, 2025).

Comparisons among LRMs have so far focused primarily on final-answer accuracy. Yet, models arriving at the same answer may follow very different reasoning strategies, as illustrated in Fig. 1. Recent work has begun to explore other crucial dimensions of reasoning, such as overthinking (Chen et al., 2025) and underthinking (Wang et al., 2025), a comprehensive understanding of reasoning behavior is still lacking. For instance, analyzing how a model balances exploration and exploitation or how much it forgets during reasoning could offer deeper insights into its core capabilities. This motivates the need for tools to systematically analyze and compare reasoning processes, raising the following question:

*How can we represent an LLM's reasoning trace to facilitate a comprehensive analysis and comparison of its internal behaviors?*

A natural way to address this question is through a tree-based representation, which captures the overall structure of reasoning, including planning and action transitions. The usefulness of such frameworks has been noted by Wu et al. (2025), who introduced an abstract "reasoning tree" to detect when a models thought process plateaus. However, instead of constructing the tree directly, they rely on indirect probes of hidden states to approximate its structure. To bridge this gap, we propose

**Problem**: Make 24 with 3, 11, 11, 12 using +, −, ×, ÷, and parentheses. Use each number once.

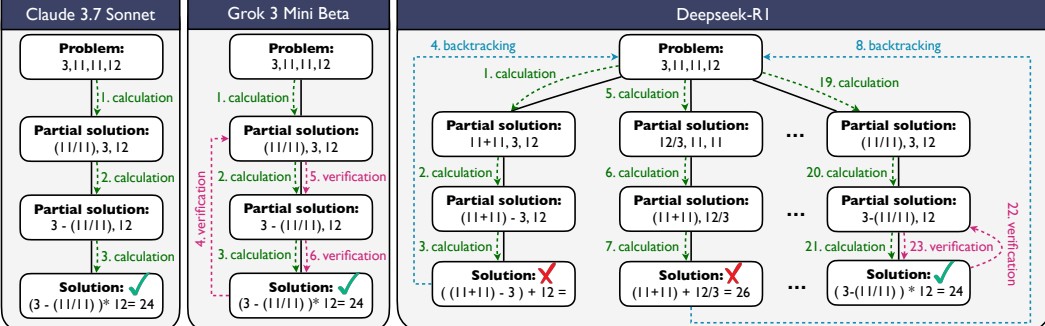

Figure 1: **ReJump representations of reasoning traces generated by Claude 3.7 Sonnet, Grok 3 Mini Beta, and DeepSeek-R1 on a Game of 24 problem.** All three models arrive at the same final answer, but their reasoning behaviors differ. Here, both Claude 3.7 Sonnet and Grok 3 Mini Beta follow a single linear reasoning path; however, Claude 3.7 Sonnet adopts the answer without verification, while Grok 3 Mini Beta verifies it before concluding. In contrast, DeepSeek-R1 explores multiple solution paths, exhibiting more deliberate behaviors such as backtracking and verification.

*ReJump*, a tree-jump representation of LLM reasoning that enables comprehensive evaluation and comparison of reasoning traces and can further be leveraged to improve reasoning accuracy. We summarize our main contributions as follows.

**We introduce *ReJump*, a tree-jump representation of LLM reasoning.** ReJump encodes a reasoning trace as a sequence of visited nodes in a hierarchical tree of intermediate problem-solving steps, where the visitation order reflects execution flow and behaviors such as verification, backtracking, and calculation. Unlike a standard tree walk (West, 2001), in which each pair of consecutive nodes is connected by an edge, reasoning traces may include transitions between non-adjacent nodes due to behaviors like backtracking or verification. We therefore term this movement between nodes as a tree *jump*. As shown in Fig. 1, this representation makes it easy to distinguish between different reasoning behaviors. Building on this representation, we define six metrics to quantify reasoning behaviors, along with tree and jump similarity metrics for comparison.

**We design an LLM agent that extracts reasoning traces into the ReJump format.** Given a reasoning trace, our agent, termed *ReJump-Extractor*, perform this via two steps: (i) it first parses it into the *tree layer*, where each node represents an intermediate step, and each edge encodes the logical dependency between steps, (ii) based on the tree layer, it constructs the *jump layer*, which captures transitions between nodes along with their corresponding action types (verification, calculation, or backtracking). In automatic evaluation, the ReJump representations produced by ReJump-Extractor reach over 0.9 tree and jump similarity with human annotations on Game of 24. In human evaluation, where annotators assess whether each generated ReJump is correct, ReJump-Extractor achieves over 80% accuracy on MATH-500.

**We utilize ReJump to evaluate and comparing reasoning traces across models, tasks, and settings.** We show that models with similar final accuracy can reason in completely different ways, and different tasks also favor different types of reasoning strategies (e.g., varying exploration-exploitation balances). Our analysis further compares reasoning traces across (i) CoT-prompted LLMs and LRMs, showing that LRMs exhibit more deliberate reasoning behaviors such as exploration and verification, and improve performance by generating more diverse solutions, though not necessarily with higher per-attempt accuracy; (ii) distilled models and their teacher LRMs, showing that distilled models inherit reasoning behaviors from their teachers; (iii) varying number of in-context reasoning examples, showing that including more reasoning examples does not always enhance problem decomposition but can induce reasoning actions such as verification and backtracking; and (iv) different checkpoints during reinforcement learning with verifiable reward (RLVR) (Guo et al., 2025), showing that RL reinforces task-preferred reasoning behaviors (e.g., models trained on tasks requiring more exploration exhibit increased exploratory reasoning throughout RL training).

**We leverage ReJump to improve the reasoning performance of LLMs.** Beyond analyzing the reasoning processes of LLMs, we show that ReJump can enhance performance. ReJump enables Best-of-N (BoN) selection and prompt selection based on desired reasoning properties (e.g., more exploration when helpful). When applied to the Game of 24 benchmark, both methods yield improvements to the pass@1 score, with performance gains ranging from +6.8% to +9.1%.

## 2 RELATED WORK

**Methods for Reasoning Visualization.** With the growing interest in analyzing LLM reasoning, multiple recent works have proposed methods to visualize and quantitatively compare different reasoning strategies (Li et al., 2025b; Zhou et al., 2025; Minegishi et al., 2025; Xiong et al., 2025; Feng et al., 2025). Zhou et al. (2025) introduce Landscape of Thoughts (LoT), a visualization method tailored for multiple-choice tasks. It represents each intermediate reasoning step as a vector by computing its perplexity-based distance to all answer options, and then projects these vectors into two dimensions using $t$-SNE for visualization of reasoning trace. They also propose three evaluation metrics: consistency, uncertainty, and perplexity, to analyze model behavior. However, LoT relies on perplexity-based vectors that lack semantic interpretability of the reasoning process. In contrast, several other studies have explored graph-based representations of reasoning traces to enhance readability, some also enable quantitative analysis (Li et al., 2025b; Minegishi et al., 2025; Xiong et al., 2025; Feng et al., 2025). ReasonGraph (Li et al., 2025b) focuses solely on readability, Minegishi et al. (2025) perform quantitative analysis via structural properties of graph such as cycles and diameter, and Xiong et al. (2025) propose metrics to evaluate the reasoning's exploration and idea integration behavior. Feng et al. (2025) also propose a graph-based view of reasoning to identify the failed-step fraction and investigate its effect on reasoning accuracy. Zhang et al. (2025a) propose DAG-Math, modeling CoT as a directed acyclic graph and using logical closeness to evaluate the fidelity of the generated trajectory. In contrast to all existing work, our representation, ReJump, a tree-jump representation that more naturally reflects the thinking process, aligning with prior work such as Tree-of-Thought (Yao et al., 2023). Moreover, ReJump enables richer and more diverse evaluations, being the first that measures exploration-exploitation balance, overthinking, verification, and forgetting, which are the key properties for quantifying reasoning behavior.

**Empirical Findings on Reasonings.** Prior empirical studies on reasoning typically fall into three categories: (i) limitations in reasoning behavior (Chen et al., 2025; Fan et al., 2025; Wu et al., 2025; Wang et al., 2025), (ii) impact of training algorithms (Yue et al., 2025; Dang et al., 2025), and (iii) factors for effective reasoning (Li et al., 2025a).

First, a well-known issue of LRMs are overthinking (Chen et al., 2025; Fan et al., 2025; Wu et al., 2025), where models continue unnecessary reasoning even after reaching a correct solution, and underthinking (Wang et al., 2025), where they abandon promising reasoning paths too early, often reflecting excessive exploration. To address overthinking, Wu et al. (2025) introduce thought calibration to dynamically terminate generation by using probes to detect when the model's reasoning tree stops growing. Second, the choice of training algorithm significantly influences reasoning behavior. Yue et al. (2025); Dang et al. (2025) observe that although RL-trained models outperform base models at small pass@$k$, they merely bias outputs toward rewarded reasoning paths without acquiring new reasoning capabilities, ultimately narrowing reasoning capacity and being surpassed by base models at large $k$. Third, recent work identifies key structural factors that contribute to effective reasoning (Gandhi et al., 2025; Li et al., 2025a). Gandhi et al. (2025) highlight behaviors in the base model such as verification and backtracking play a key role in enabling RL training to further develop reasoning ability and improve performance. Similarly, Li et al. (2025a) argue that the logical form of reasoning, rather than the content of individual steps, is key to LRM reasoning quality. The ReJump representation facilitates these analyses by systematically capturing overthinking, exploration-exploitation dynamics, and behavioral differences across different experiment settings.

## 3 REJUMP: A TREE-JUMP REPRESENTATION OF LLM REASONING

In this section, we introduce the ReJump representation and metrics for analyzing a single tree-jump and comparing pairs of tree-jumps, and will describe how reasoning traces are extracted into the ReJump representation in Sec. 4.

### 3.1 DECOMPOSING REASONING INTO TREE AND JUMP LAYERS

We extract each model-generated reasoning into a ReJump, a two-layer representation that captures both structure and actions of reasoning traces.

- **Tree layer (structure)**: We define a tree $T = (V, E)$, where $V = \{v_i\}_{v=0}^{|V|}$ is the set of nodes and $E$ is the set of edges. Following Yao et al. (2023), each node $v \in V$ represents a partial solution, with the root node corresponding to the initial state containing no solution. An edge $e \in E$ indicates that the parent's partial solution is a direct prerequisite for the child's.

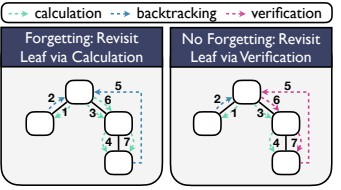

(a) Illustration of how $d_{\text{jump}}$ quantifies the exploration-exploitation trade-off in model reasoning. Given a sequence of visited leaf nodes $(v_1, v_2, v_3)$, the left panel depicts a trace exhibiting local exploration (shorter paths between nodes), while the right panel shows a trace with larger jumps to distant leaves, reflecting more global exploration.

(b) Illustration of forgetting vs. no forgetting in ReJump. Revisiting an already-seen node via `calc` indicates forgetting, while revisiting via `verify` does not.

Figure 2: Illustration of Jump Distance ($d_{\text{jump}}$) and Forgetting, as defined in ReJump.

- **Jump layer (action)**: Let $\boldsymbol{i} = (i_0, i_1, \ldots, i_K)$ denote the sequence of reasoning steps, where $i_k$ refers to the index of $k$-th visited node in the tree. The jump starts at $v_{i_0}$ (the root) and ends at $v_{i_K}$ (the final solution). Each transition between consecutive steps $(i_k, i_{k+1})$ is labeled with an action type $\phi_k \in \{\texttt{calc}, \texttt{verify}, \texttt{backtrack}\}$, where $k = 0, \ldots, K-1$. Here, `calc` refers to generating an intermediate step via calculation or derivation. Both `verify` and `backtrack` involve returning to a previously visited node: `verify` checks its correctness, while `backtrack` restarts from it to explore an alternative reasoning path. The sequence of actions in the jump layer is denoted by $\boldsymbol{\phi} = (\phi_0, \ldots, \phi_{K-1})$. A jump layer is defined as the pair $W = (\boldsymbol{i}, \boldsymbol{\phi})$, specifying the sequence of visited nodes and the corresponding transitions taken during reasoning.

## 3.2 Quantifying Reasoning Behavior: Evaluation and Comparison Metrics

We define a *derived solution step* as any step in the jump layer that reaches a leaf node via a `calc` transition, thereby contributing to the solution. Steps that visit a leaf solely for verification are excluded, whereas revisiting an already-seen leaf via `calc` still qualifies as a derived solution step.

**Evaluation Metrics.** This tool enables analysis of LLM reasoning behaviors, including solution diversity, exploration–exploitation trade-off, effectiveness in identifying correct paths, frequency of overthinking, forgetting, and verification. These aspects are quantified using the metrics below, computed across all reasonings and their corresponding ReJumps within a task. Each metric below is defined at the instance level and is directly averaged across instances to obtain the task-level score, except for $r_{\text{forget}}$, which is defined only at the task level and described accordingly.

- **Solution Count** ($\#_{\text{solution}}$): Number of distinct solutions, i.e., the total leaf nodes in the tree (including incomplete ones).
- **Jump Distance** ($d_{\text{jump}}$): Measures the exploration-exploitation balance by computing the tree distance (edge count) between the nodes of each consecutive pair of derived solution steps. For a reasoning instance, $d_{\text{jump}}$ is the average distance across all consecutive pairs of derived solution steps within the jump layer. See Fig. 2a for illustrative examples.
- **Success Rate** ($r_{\text{success}}$): Fraction of derived solution steps that yield a correct answer. Measures the efficiency and accuracy.
- **Verification Rate** ($r_{\text{verify}}$): Fraction of all transitions labeled `verify`. Indicates how deliberate and self-critical the reasoning process is.
- **Overthinking Rate** ($r_{\text{overthinking}}$): Fraction of derived solution steps that occur *after* the first correct derived solution step is found, quantifying unnecessary exploration. Quantifies unnecessary exploration and inefficiency.
- **Forgetting Rate** ($r_{\text{forget}}$): This metric is defined only at the task level. A reasoning trace is flagged as *forgetting* if it revisits the same leaf node via `calc`, indicating that the model has re-entered an already visited path; see Fig. 2b. $r_{\text{forget}}$ is the proportion of such reasoning traces within the task, highlighting poor memory or state-tracking.

See Sec. B.1 for formal mathematical definition of these metrics.

**Comparison Metrics.** To assess the similarity between reasoning process produced by different models, we introduce similarity metrics to compare tree and jump representation, respectively. As with the evaluation metrics, all comparison metrics below are defined at the instance level and averaged across instances to obtain task-level scores.

- **Tree Similarity** ($\text{Sim}_T$): Measures the similarity in problem decomposition structure between two reasoning traces by comparing their corresponding trees. These metrics assess whether models adopt similar reasoning structures, without relying on the exact content of individual steps. This choice is justified by the work of Li et al. (2025a), which demonstrates that overall logical

structure, rather than the specific content at each node, is the primary factor influencing reasoning quality. Given reasoning trees $T = (V, E)$ and $T' = (V', E')$, we compute their Zhang-Shasha Tree Edit Distance (TED) (Paaßen, 2018), which measures the minimum number of edit operations to transform one tree into another. We consider only insertions and deletions, ignoring relabeling since node semantics are not considered here. The tree similarity is then defined as $\text{Sim}_T(T, T') = 1 - \text{TED}(T, T') / \max(|V|, |V'|)$.

- **Jump Similarity** ($\text{Sim}_J$): Measures the similarity in action transition patterns between the jumps derived from two reasoning traces. For each reasoning jump $W = (\boldsymbol{i}, \boldsymbol{\phi})$, we construct a $3 \times 3$ transition probability matrix $P$, where $P_{a,b}$ is the empirical probability of transitioning from action $a$ to $b$, with $a, b \in \{\texttt{calc}, \texttt{verify}, \texttt{backtrack}\}$. Given two jump layers $W, W'$ with transition probability matrices $P$ and $P'$, we define their similarity as $\text{Sim}_J(W, W') = 1 - \text{JS}(P\|P')$, where JS is the Jensen-Shannon divergence, a symmetric and bounded variant of KL divergence defined as $\text{JS}(P\|P') = \frac{1}{2}\text{KL}(P\|\frac{1}{2}(P + P')) + \frac{1}{2}\text{KL}(P'\|\frac{1}{2}(P + P'))$. Higher $\text{Sim}_J$ values indicate greater alignment in action transition behavior.

## 4 ReJump-Extractor: Extracting CoTs into ReJump Format

In this section, we introduce ReJump-Extractor, an LLM agent for extracting reasoning traces into ReJump format via two steps:

- **Tree Layer Extraction**: We use Gemini 2.5 Pro (Google Deepmind, 2024) to extract both the tree and the jump representations from each reasoning trace. Given the original problem input and the model-generated reasoning, we prompt LLM to produce a JSON object that encodes the reasoning tree. This JSON is a dictionary where each key corresponds to a node index, and each value contains three fields: "problem" (the subproblem addressed at that node), "parent" (the index of the prerequisite node whose partial solution this node builds upon), and "solution" (the result corresponding to the subproblem). For the root node, all three fields are either left empty or labeled as "initial state."

- **Jump Layer Extraction**: We parse the JSON dictionary to construct the tree structure. Then, using the original input, the full reasoning, and the generated tree JSON as context, we prompt LLM again to extract the jump layer. The jump layer is represented as a JSON list, where each entry describes a transition between nodes, with fields "from," "to," and "category" indicating the source node, target node, and transition type (e.g., calculation, verification, or backtracking). We use this information to visualize the full reasoning trajectory overlaid on the constructed tree.

An visualization of how ReJump-Extractor extracts CoTs into ReJump format is shown in Sec. C. The extraction prompt is provided in Sec. C.1, with example ReJump representations derived from real reasoning traces in Sec. C.2.

### 4.1 Assessing the Reliability of ReJump-Extractor

To evaluate the reliability of ReJump-Extractor, we adopt two approaches: (i) automatic evaluation and (ii) human assessment. Since the ground-truth ReJump format of reasoning traces in Game of 24 are uniquely defined, we develop an automated verification pipeline using a synthetic dataset. For MATH500, where multiple ground-truth trees and jumps are possible with small variations between them, we instead rely on manual verification of the extracted ReJump representations.

**Automatic Evaluation.** The dataset includes 70 reasoning traces, each paired with manually created and verified ground-truth (GT) ReJump representations. The trees have a minimum of 4 nodes and maximum of 20. We use ReJump-Extractor to process the reasoning traces and compare its outputs with the ground-truth data, reporting tree and jump similarities in Tab. 1. We find that ReJump representations extracted by ReJump-Extractor using Gemini 2.5 Pro align closely with the ground truth, confirming the reliability of our approach. Comparison with additional LLMs is provided in Sec. C.3.

Table 1: Alignment between ReJump-Extractor outputs and GT ReJump on Game of 24.

|  | $\text{Sim}_T$ | $\text{Sim}_J$ |
|---|---|---|
| ReJump-Extractor | .943 | .940 |

Table 2: Human evaluation of decomposition of reasoning traces via ReJump accuracy on MATH500.

| Reasoning Model | Pass@1 | Pass@2 | Pass@3 |
|---|---|---|---|
| DeepSeek-R1 | 81% | 87% | 90% |
| QwQ-32B | 80% | 83% | 94% |

**Human Assessment.** We evaluate the accuracy of ReJump in decomposing tree and jump reasoning generated by DeepSeek-R1 and QwQ-32B on the MATH500 benchmark, with results shown in Tab. 2. The columns pass@1, pass@2, and pass@3 denote the accuracy after one, two, and three

Figure 3: **Reasoning performance of DeepSeek-R1, Grok 3 Mini Beta, QwQ-32B, Phi-4-reasoning-plus, and Claude 3.7 Sonnet on MATH-500 and Game of 24.** The bar plots present the final accuracy (pass@1), while the radar plots detail six reasoning metrics. For comparability, solution count and jump distance are normalized across all models and datasets. To ensure that higher values consistently reflect preferred behavior, we report the non-forgetting rate and non-overthinking rate rather than forgetting rate and overthinking rate. The results show that models display distinct reasoning behaviors across datasets. Furthermore, even when models achieve similar final performance, their underlying reasoning processes can differ significantly. To better highlight metric differences among the strongest models DeepSeek-R1, Grok 3 Mini Beta, and Claude 3.7 Sonnet, Fig. 12 focuses exclusively on these three.

extractions under different random seeds, as judged by human evaluation. Even a single extraction (pass@1) achieves strong reliability, with accuracies of 81% for DeepSeek-R1 and 80% for QwQ-32B. Multiple extractions further boost performance, surpassing 90% accuracy by pass@3.

## 5 ANALYZING LLM REASONING BEHAVIORS WITH REJUMP

Building on the ReJump representation introduced in Sec. 3, we analyze the reasoning structures of four state-of-the-art LRMs across two datasets. The two datasets we consider are: (i) MATH-500 (Lightman et al., 2024), a widely used benchmark for mathematical reasoning, and (ii) Game of 24, a task that requires strong planning and enables clear inspection of model behavior in terms of exploration, exploitation, and related strategies, as previously adopted by Yao et al. (2023). Unless otherwise specified, all experiments use a decoding temperature of 0 to ensure deterministic and reproducible outputs. In addition, we examine the effect of alternative decoding strategies on reasoning behavior in Sec. D.5.

### 5.1 COMPARING REASONING STRUCTURE ACROSS STATE-OF-THE-ART LRMS AND TASKS

This experiment focuses on the state-of-the-art LRMs for which we have access to intermediate reasoning traces: DeepSeek-R1 (Guo et al., 2025), QwQ-32B (Qwen Team, 2025), Grok 3 Mini Beta (xAI, 2025), Phi-4-reasoning-plus (Abdin et al., 2025), and Claude 3.7 Sonnet (Anthropic, 2025). Since Claude 3.7 Sonnet uses a fixed temperature of 1 in thinking mode, and Phi-4-reasoning-plus performs significantly better with temperature 1 than with 0, we set the decoding temperature to 1 for all models in this experiment. The results for DeepSeek-R1, QwQ-32B, and Grok 3 Mini Beta under temperature 0 are deferred to Sec. D.1. For Claude 3.7 Sonnet, due to high cost, the token budget is capped at 1064, whereas all other models use 8000. We generate reasoning traces and final answers for both MATH-500 and Game of 24 using each model, and analyze their reasoning behaviors through ReJump. Fig. 3 presents a unified view of final accuracy (pass@1) alongside six reasoning evaluation metrics proposed in Sec. 3.2.

**Comparison across Tasks.** MATH-500 and Game of 24 differ in the structure and demands of reasoning. MATH-500 problems are typically deterministic, with only one or two valid solution paths, encouraging focused, step-by-step reasoning. In contrast, Game of 24 requires generating diverse arithmetic expression to reach the target number, promoting trial-and-error and exploratory strategies. This difference is evident in the results: as shown in Fig. 3, all five models yield a much lower average solution count on MATH-500 compared to Game of 24. The jump distance is also substantially higher for Game of 24. Meanwhile, MATH-500 shows much higher success and verification rates, likely due to its proof-like structure that favors thorough verification and local exploitation over broad exploration. To systematically analyze which reasoning behaviors contribute most to final accuracy, we compute the feature importance of six reasoning metrics and summarize them in Tab. 3. We find that $r_{success}$ has the strongest impact on pass@1 for MATH-500, while both $d_{jump}$ and $r_{success}$ are key predictors of pass@1 on Game of 24.

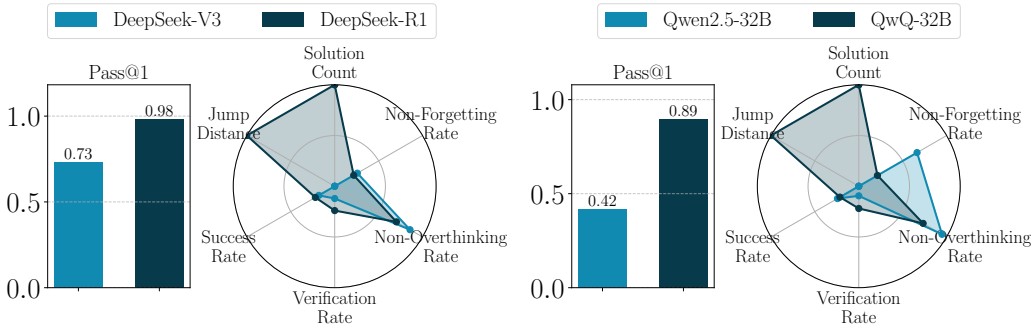

(a) DeepSeek-V3 v.s. DeepSeek-R1      (b) Qwen2.5-32B v.s. QwQ-32B

Figure 4: **Comparison of base LLMs (DeepSeek-V3, Qwen-2.5-32B) and their corresponding LRMs (DeepSeek-R1, QwQ-32B) on pass@1 and reasoning metrics for the Game of 24.** The bar plots present the final accuracy (pass@1), while the radar plots detail six reasoning metrics. For comparability, solution count and jump distance are normalized across all models and datasets. To ensure that higher values consistently reflect preferred behavior, we report the non-forgetting rate and non-overthinking rate rather than the forgetting rate and overthinking rate. Despite similar $r_{\text{success}}$, LRMs achieve higher pass@1 by generating more and diverse solutions, as reflected in higher average solution counts and jump distances. LRMs also exhibit increased verification, overthinking, and forgetting behaviors.

**Comparison across LRMs.** Among the five models, DeepSeek-R1 and Grok 3 Mini Beta achieve the highest pass@1, followed by QwQ-32B, with Phi-4-reasoning-plus and Claude 3.7 Sonnet lag behind. Although DeepSeek-R1 and Grok 3 Mini Beta reach similar final accuracy, their reasoning behaviors differ substantially. Compared to DeepSeek-R1, Grok 3 Mini Beta adopts a narrower approach: it explores fewer paths and makes shorter jumps, yet reaches correct solutions more efficiently, as reflected in its higher success rate. By contrast,

Table 3: **Contribution of reasoning metrics to solution correctness on MATH-500 and Game of 24 for DeepSeek-R1, Grok 3 Mini Beta, QwQ-32B, Phi-4-reasoning-plus, and Claude 3.7 Sonnet.** "Contribution" is quantified by the feature-importance scores produced when an XGBoost model is trained on six reasoning-evaluation metrics to classify each solution as correct or incorrect; the model attains accuracies of 0.9197 (MATH-500) and 0.7980 (Game of 24) versus majority-class baselines of 0.6928 and 0.6768. Metrics with importance scores $> 0.2$ are bold-faced. Game of 24 emphasizes exploration (i.e., $d_{\text{jump}}$), while MATH-500 emphasizes exploitation (i.e., $r_{\text{success}}$).

| Dataset | $\#_{\text{solution}}$ | $d_{\text{jump}}$ | $r_{\text{success}}$ | $r_{\text{verify}}$ | $r_{\text{overthinking}}$ | $r_{\text{forget}}$ |
|---|---|---|---|---|---|---|
| MATH-500 | .0177 | .0541 | **.8548** | .0173 | .0219 | .0341 |
| Game of 24 | .1402 | **.2742** | **.2146** | .0980 | .1413 | .1317 |

DeepSeek-R1 engages in broader exploration, producing more candidate solutions and making longer jumps, though at the cost of a slightly lower success rate. Despite this, it ultimately achieves accuracy comparable to Grok 3 Mini Beta. QwQ-32B exhibits less exploration than DeepSeek-R1 and a lower success rate than Grok 3 Mini Beta, resulting in worse performance compared to both. Phi-4-reasoning-plus and Claude 3.7 Sonnet perform the worst among all models, with even lower levels of exploration and success rates. Notably, Claude 3.7 Sonnet exhibits the least deliberate reasoning behavior, as both exploration ($\#_{\text{solution}}$ and $d_{\text{jump}}$) and $r_{\text{verify}}$ are low. Phi-4-reasoning-plus demonstrates slightly more deliberate reasoning behaviors, with higher exploration than Claude 3.7 Sonnet, which contributes to its relatively better performance on Game of 24. All models exhibit overthinking, which is an issue previously observed in LRMs (Chen et al., 2025; Yang et al., 2025), as well as forgetting, both of which reflect inefficient reasoning behaviors.

> *Findings*:
> • *Task characteristics shape and favor distinct reasoning behaviors.*
> • *Models achieving comparable accuracy may employ distinct reasoning strategies.*
> • *Overthinking and forgetting are prevalent across LRMs.*

## 5.2 COMPARING REASONING STRUCTURE: STANDARD LLMS VS. LRMS

While LRMs are optimized for multi-step reasoning, general-purpose LLMs can still reason effectively when prompted (e.g., with CoT). Their differing training objectives lead to distinct reasoning behaviors. Yue et al. (2025) find that RL favors high-reward paths with less exploration: RL-trained LRMs outperform base models at low $k$ in pass@$k$, but underperform at high $k$. Using ReJump, we compare reasoning behaviors of two LRM-base pairs: (i) Qwen2.5-32B vs. QwQ-32B, and (ii) DeepSeek-V3 vs. DeepSeek-R1.

Fig. 4 shows results on Game of 24. Consistent with Yue et al. (2025), LRMs outperform LLMs in pass@1, but not by favoring high-reward paths. Instead, their success rates are similar, and gains stem from generating more solutions. LRMs also exhibit more exploratory reasoning, reflected in higher jump distances and more frequent shifts in approach. This does not contradict Yue et al. (2025), as their analysis considers exploration across samples, while ours focuses on single-trace exploration. LRMs further show higher verification rates, but also more overthinking and forgetting.

> **Findings**:
>
> - *LRMs achieve higher pass@1 by generating more numerous and diverse solutions, despite not necessarily improving per-attempt accuracy.*
>
> - *Compared to LLMs, LRMs demonstrate more deliberate reasoning behaviors, such as increased exploration and verification, but also suffer more from overthinking and forgetting.*

The results for MATH-500, presented in Sec. D.2, further support our findings.

### 5.3 IMPACT OF DISTILLATION ON REASONING STRUCTURE: COMPARING TEACHER AND DISTILLED MODELS

Model distillation transfers the capabilities of large LRMs to smaller, more efficient models (Guo et al., 2025), often preserving task performance. However, its effect on underlying reasoning structure and actions remains unclear. To investigate this, we compare three model types (base, teacher, and distilled) at two scales: the 14B group uses Qwen-2.5-14B and DeepSeek-R1-Distill-Qwen-14B, while the 32B group uses the 32B counterparts of the same models, with DeepSeek-R1 as the teacher in both cases.

We report the similarity to the teacher model before and after distillation for the 14B comparison group in Tab. 12; results for the 32B group and detailed per-metric comparisons for both groups are deferred to Sec. D.3. The results show that the distilled model consistently moves closer to the teacher in both tree similarity ($\text{Sim}_T$) and jump similarity ($\text{Sim}_J$). Further analysis in Sec. D.3 confirms that distilled models inherit reasoning behaviors from teacher LLMs, including broader exploration, verification, and backtracking, though success rates are not improved.

Table 4: **Tree similarity ($\text{Sim}_T$) and jump similarity ($\text{Sim}_J$) between each model and the teacher model.** Base: Qwen2.5-14B; Distilled: DeepSeek-R1-Distill-Qwen-14B; Teacher: DeepSeek-R1. Both metrics improve after distillation, showing that distilled model more closely replicates the teacher's reasoning structure.

| Reference Model | Metric | | | |
| | $\text{Sim}_T$ | | $\text{Sim}_J$ | |
| | Base | Distilled | Base | Distilled |
| MATH-500 | .715 | **.728** | .771 | **.878** |
| Game of 24 | .360 | **.426** | .873 | **.905** |

> **Finding**: *Distilled models inherit reasoning behaviors from teacher models, as evidenced by gains in both tree and jump similarities.*

In Sec. D.3, we further compare the distilled model (DeepSeek-R1-Distill-Qwen-32B) with the RL-trained model (QwQ-32B) as an initial exploration of how SFT and RL differ in their impact on reasoning behavior.

### 5.4 IMPACT OF REASONING EXAMPLES ON REASONING STRUCTURE

Although prior work (Agarwal et al., 2024; Zhang et al., 2025b) has explored the use of reasoning examples in prompts to enhance LLM reasoning capabilities, how in-context examples reshape the reasoning behavior remains underexplored. A natural question arises: How does the presence and number of examples affect reasoning characteristics? To investigate these questions, we vary the number of in-context reasoning examples ($\{0, 1, 2, 3\}$) included in the prompt and analyze resulting changes in reasoning behavior. We evaluate DeepSeek-V3 and Gemini 2.0 Flash, with the latter following prior work (Agarwal et al.,

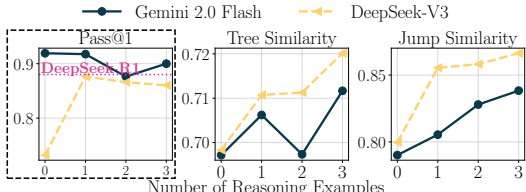

Figure 5: **Effect of the reasoning examples on reasoning behaviors (MATH-500).** Gemini 2.0 Flash and DeepSeek-V3 are prompted with DeepSeek-R1 examples. The dashed boxes indicate final accuracy for different number of in-context examples, while the remaining plots show tree and jump similarity to DeepSeek-R1.

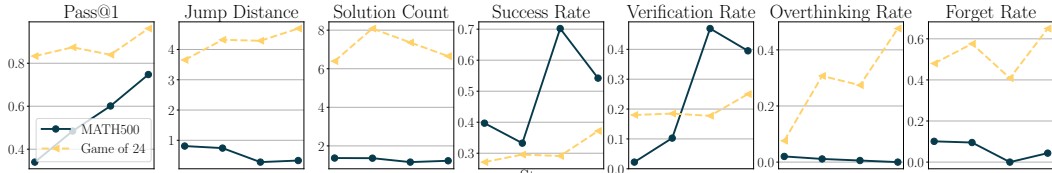

Figure 6: **Evolution of reasoning metrics during RL.** We use Qwen3-1.7B for the Game of 24 task and Qwen3-8B for the MATH500 task, applying DAPO (Yu et al., 2025) to enhance reasoning performance without supervised fine-tuning, with max sequence length set to 2048. RL encourages more exploratory reasoning on Game of 24, as indicated by increased jump distance, while promoting more exploitative behavior on MATH500, reflected by higher success rates and reduced jump distance. These results show that RL-induced improvements align with the inherent reasoning characteristics of each task. Note that our pass@1 computation also accounts for the correct formatting of `<answer></answer>` tags.

2024; Zhang et al., 2025b). Since Gemini-family LRMs like Gemini 2.5 Pro do not expose reasoning traces, we use DeepSeek-R1-generated examples from other samples within the same dataset.

Fig. 5 shows how pass@1, tree similarity, and jump similarity vary with the number of in-context examples on MATH-500. Accuracy does not consistently improve; in fact, Gemini 2.0 Flash even outperforms DeepSeek-R1 without demonstrations. However, jump similarity rises steadily, indicating stronger imitation of LRM-style behaviors (e.g., verification, calculation, backtracking) with more examples. In contrast, tree similarity shows no clear trend, suggesting limited impact on problem decomposition. Results on Game of 24 (Sec. D.4) show similar patterns.

> ***Finding***: *Increasing the number of in-context reasoning examples has a stronger and more consistent influence on reasoning actions (e.g., verification and backtracking) than on high-level problem decomposition strategies, which remain relatively invariant.*

## 5.5 EVOLUTION OF REASONING DYNAMICS UNDER REINFORCEMENT LEARNING

Guo et al. (2025) demonstrates the effectiveness of RLVR in enhancing the reasoning capabilities of standard LLMs. In this experiment, we use the ReJump representation to visualize the evolution of models' reasoning behavior during RL. Specifically, we apply DAPO (Yu et al., 2025) on both the MATH-500 and Game of 24 tasks. For MATH-500, we employ Qwen3-8B, and for Game of 24, Qwen3-1.7B. Each model is evaluated at four checkpoints, corresponding to one-quarter intervals throughout training (i.e., 1/4, 2/4, 3/4, and 4/4 of the training process).

As discussed in Sec. 5.1, MATH-500 favors higher success rates, indicating stronger exploitation, whereas Game of 24 benefits from both increased jump distance and higher success rates, reflecting a need for balanced exploration and exploitation. The results in Fig. 6 show that RL progressively shapes the models reasoning dynamics to match these task-specific requirements: promoting more exploitative reasoning (i.e., higher success rates) on MATH500 and both greater exploration (i.e., higher jump distance) and higher success rates on Game of 24.

> ***Finding***: *Reinforcement learning progressively aligns a model's reasoning behavior with the demands of the target task.*

## 6 ENHANCING LLM REASONING WITH REJUMP

Beyond using ReJump to dissect LLM reasoning and compare behaviors across different LRMs, tasks, and settings, we further demonstrate in this section that ReJump can also be leveraged to enhance LLM reasoning.

A key advantage of ReJump is that they enable selecting outputs with desired reasoning characteristics, as measured by the six metrics, without requiring ground-truth solutions. We focus on the Game of 24 because its difficulty depends less on an LLM's raw capability (e.g., success rate) and more on its reasoning behavior (e.g., exploration), which can be more effectively improved by enhancing reasoning patterns. As established in Sec. 5.1, Game of 24 benefit from more explorative reasoning. Therefore, our experiments in this section center on this task instead of MATH500. All reported results are averaged over three random seeds.

### 6.1 IMPROVING REASONING VIA BEST-OF-N SELECTION WITH REJUMP

Accordingly, we apply a Best-of-N (BoN) strategy: generate multiple responses and use ReJump to select the one exhibiting the highest exploration (i.e., the largest jump distance, $d_{\text{jump}}$).

Table 5: **Performance of the majority vote and Best-of-N (BoN) with ReJump on Game of 24 using QwQ-32B and Phi-4-reasoning-plus.** BoN with ReJump improves both jump distance ($d_{jump}$) and pass@1.

| Model | Method | pass@1 | $d_{jump}$ |
|---|---|---|---|
| QwQ-32B | Majority Vote | 0.76 | 4.20 |
| | BoN w. ReJump | **0.82** | **5.70** |
| Phi-4-reasoning-plus | Majority Vote | 0.77 | 3.32 |
| | BoN w. ReJump | **0.84** | **5.53** |

Table 6: **Comparison of performance between the default prompt and the prompt selected by ReJump from four candidate prompts on Game of 24.** The ReJump-chosen prompt yields better performance.

| Model | Prompt | pass@1 | $d_{jump}$ |
|---|---|---|---|
| QwQ-32B | Default | 0.73 | 4.09 |
| | ReJump-chosen | **0.78** | **4.28** |
| Phi-4-reasoning-plus | Default | 0.76 | 3.42 |
| | ReJump-chosen | **0.82** | **3.98** |

We consider QwQ-32B and Phi-4-reasoning-plus in this experiment since their performance is relatively limited and has more room for improvement comparing to other LRMs as shown in Sec. 5.1. In this experiment, we set $N = 3$, which means, for each prompt, we generate 3 response, and use ReJump to choose the one with highest jump distance. As a baseline, we use majority vote to ensemble the three responses. The comparison between the majority vote and our BoN with ReJump are shown in Tab. 5, demonstrating the effectiveness of this method compared to the baseline. In Sec. E, we extend our analysis to two additional datasets requiring different reasoning characteristics, further supporting our findings.

### 6.2 PROMPT SELECTION WITH REJUMP

Another natural application of ReJump's comprehensive measurement is prompt selection. There are several ways to leverage ReJump for this purpose. For efficient reasoning, one can design multiple prompts, test them on a development set, and use ReJump to evaluate the generated responses' reasoning behavior. The prompt that achieves a higher success rate and lower overthinking rate can be selected. For search-heavy tasks, such as maze solving or the Game of 24, ReJump can help identify the prompt that provides the best exploration-exploitation tradeoff.

In this experiment, we focus on prompt selection for improving exploration in the Game of 24. Similar to the previous experiment, we consider QwQ-32B and Phi-4-reasoning-plus. We design four prompts (see Sec. E.2 for details) intended to encourage broader exploration and use ReJump to select the one yielding the highest jump distance. Tab. 6 shows that without ground-truth labels, prompt selection guided by ReJump improves the performance of both models on the Game of 24.

### 7 CONCLUSION

In this work, we propose ReJump, a tree-jump representation of reasoning traces, with the tree layer capturing the hierarchical structure of partial solutions and their dependencies, and the jump layer, tracing the sequential execution of reasoning steps. Using our proposed ReJump-Extractor method to extract reasoning traces into ReJump, the resulting representations allows us to quantitatively analyze reasoning behaviors such as the exploration-exploitation trade-off, overthinking, and forgetting, which enables comparison of reasoning processes themselves beyond final accuracy across models, tasks, and settings. Because these metrics reveal potential weaknesses in reasoning regardless of final performance, ReJump shows what needs to be improved during training and helps decide which inference strategy works best for a task. Beyond serving as an analytic tool that indirectly guides LRM development, ReJump can also directly enhance reasoning performance, as demonstrated through two applications: Best-of-N response selection and prompt selection.

**Limitations & Future Work.** Despite these contributions, several limitations remain. First, cost and efficiency are a challenge: ReJump-Extractor requires a separate, capable LLM to process each reasoning trace, which is computationally expensive and slow, limiting large-scale use such as real-time feedback during training. Overcoming this limitation would make it possible to analyze the dynamics of how reasoning evolves during training, including under outcome- vs process-supervised reinforcement learning and supervised fine-tuning. In addition, incorporating ReJump-derived signals into reward modeling represents another promising direction. Second, when comparing reasoning traces, our current tree and jump similarity metric capture only logical structure and action transition distributions. This simplifies computation but can mask important differences: two models may yield identically structured trees yet differ semantically, and a perfect jump similarity score (i.e., 1.0) may still hide distinct temporal behaviors. For instance, one model might perform derivation first and verify at the end, whereas another may interleave computation and verification throughout. Incorporating semantic similarity and temporal dynamics is therefore interesting future work. Finally, the method still requires defining partial solutions for each task, necessitating task-specific prompting; automating this adaptation would greatly improve usability.

## REPRODUCIBILITY STATEMENT

We support the reproducibility of all our experiments. All datasets and implementation are publicly available at our GitHub repository: https://anonymous.4open.science/r/ReJump-C826.

## ETHICS STATEMENT

This work provides a structured framework for analyzing the reasoning process of LLMs, offering insights into how different models handle complex reasoning tasks. While the study itself is foundational and does not involve deployment or real-world decision-making, it informs future efforts to design more interpretable and trustworthy LLMs.

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

# Appendix

## A  EXTENDED SEC. 2: RELATED WORK

We provide an extended version of Sec. 2, including additional discussion of techniques for eliciting reasoning and a more detailed comparison between related works and ours.

**Techniques for Eliciting Reasoning.**  There are two main approaches to eliciting the reasoning capabilities of LLMs: (i) training-free methods (Wei et al., 2022; Kojima et al., 2022; Yao et al., 2023), and (ii) training-based methods (Guo et al., 2025) that explicitly internalize CoT reasonings. For training-free methods, two early CoT works demonstrate their effectiveness: Wei et al. (2022) show that inserting step-by-step reasoning examples in a few-shot setting significantly improves reasoning, while Kojima et al. (2022) find that simply prompting with "let's think step by step" yields similar gains in a zero-shot setting. Unlike sequential CoT, Yao et al. (2023) propose Tree-of-Thought, which extends CoT by exploring multiple reasoning paths as a tree more for strategic problem-solving. Following the success of o1 (Jaech et al., 2024), recent efforts focus on training LRMs to elicit stronger reasoning abilities (Guo et al., 2025; Qwen Team, 2025; Kimi Team et al., 2025; Seed et al., 2025). A notable example is DeepSeek-R1 (Guo et al., 2025), which begins with cold-start supervised fine-tuning on structured CoT data, followed by reinforcement learning using Group Relative Policy Optimization (Shao et al., 2024) with rule-based rewards, such as correctness and format adherence, resulting in strong performance on reasoning tasks.

**Methods for Visualizing Reasoning.**  With the growing interest in analyzing LLM reasoning, many recent works have proposed methods to visualize and quantitatively compare different reasoning strategies (Lee et al., 2024; Ko et al., 2024; Li et al., 2025b; Zhou et al., 2025; Feng et al., 2025). Lee et al. (2024) aim to explore how different math-solving approaches in reasoning affect the final performance, by categorizing math-solving approaches into text, code, and cumulative reasoning, prompting LLMs with each and using Venn diagrams to show the overlap and divergence in solvable problem spaces that each method explores. Ko et al. (2024) compare the reasoning of different models by introducing the DEPTHQA dataset by decomposing complex questions into a hierarchy of sub-questions, enabling analysis of how LLMs reason across knowledge depths and revealing reasoning inconsistencies such as "forward discrepancies" (failing complex tasks despite prerequisite success) and "backward discrepancies" (succeeding on complex tasks but failing simpler ones). Five recent works (Li et al., 2025b; Zhou et al., 2025; Minegishi et al., 2025; Xiong et al., 2025; Feng et al., 2025) are more closely related to ours, as they also aim to visualize individual reasoning traces. Zhou et al. (2025) introduce Landscape of Thoughts (LoT), a visualization method tailored for multiple-choice tasks. It represents each intermediate reasoning step as a vector by computing its perplexity-based distance to all answer options, and then projects these vectors into two dimensions using t-SNE for visualization of reasoning trace. They also propose three evaluation metrics, consistency, uncertainty, and perplexity, to analyze model behavior. In contrast to LoT, our method parses the actual reasoning content to construct structured graphs that preserve both plan decomposition and reasoning actions. This allows us to analyze fine-grained behaviors such as explorationexploitation balance, overthinking, verification, and forgetting. In contrast, our method constructs structural representations directly from free-form outputs and enables quantitative analysis of six distinct reasoning behaviors under various experimental conditions. ReasonGraph (Li et al., 2025b) instead prompts the model to generate reasoning traces in graph form, improving interpretability by reducing the difficulty of analyzing long outputs. While this line of work emphasizes interpretability, Minegishi et al. (2025); Xiong et al. (2025) go further by introducing quantitative metrics for graph-based reasoning representations. Specifically, Minegishi et al. (2025) analyze structural properties of graphs (e.g., cycles and diameter), while Xiong et al. (2025) propose metrics to capture exploration and idea integration. Meanwhile, Feng et al. (2025) also propose a graph-based view of reasoning to identify the failed-step fraction and investigate its effect on reasoning accuracy. In contrast, ReJump enables richer and more diverse evaluation, including measures of exploration vs. exploitation, overthinking, verification, and forgetting. Crucially, it can be applied not only for analysis but also to directly improve reasoning quality.

**Empirical Findings on Reasonings.**  Prior empirical studies on reasoning typically fall into three categories: (i) limitations in reasoning behavior (Chen et al., 2025; Fan et al., 2025; Wang et al., 2025), (ii) impact of training algorithms (Yue et al., 2025; Dang et al., 2025), and (iii) factors for effective reasoning (Li et al., 2025a; Stechly et al., 2024; Gandhi et al., 2025). First, the most well-known issue of LRMs are overthinking overthinking (Chen et al., 2025; Fan et al., 2025), continuing

unnecessary reasoning even after reaching a correct solution, and underthinking (Wang et al., 2025), where models abandon promising reasoning paths too early, often reflecting excessive exploration. Second, the choice of training algorithm significantly influences reasoning behavior. Yue et al. (2025); Dang et al. (2025) observe that although RL-trained models outperform base models at small pass@$k$, they merely bias outputs toward rewarded reasoning paths without acquiring new reasoning capabilities, ultimately narrowing reasoning capacity and being surpassed by base models at large $k$. Third, recent work identifies key structural factors that contribute to effective reasoning. Gandhi et al. (2025) highlight behaviors such as verification, backtracking, and subgoal setting as essential for RL-trained models, and suggest that Qwen outperforms Llama as a base model due to higher appearance of such behaviors in CoT. Similarly, Li et al. (2025a) argue that the logical form of reasoning, rather than the content of individual steps, is key to LRM reasoning quality. Stechly et al. (2024) further shows that reasoning performance heavily depends on highly specific instruction for various tasks. The ReJump framework facilitates these analyses by systematically capturing overthinking, explorationexploitation dynamics, and behavioral differences across training settings. In Sec. 5.1, we further quantify the importance of factors such as solution count, success rate, and verification frequency in predicting reasoning correctness, and show how different tasks favor distinct reasoning strategies.

# B EXTENDED SEC. 3: REJUMP

In this section, we follow the structure of Sec. 3, but provide additional details such as the prompts and formal mathematical definitions of the metrics. To make the discussion self-contained, we also repeat some key content from the main text.

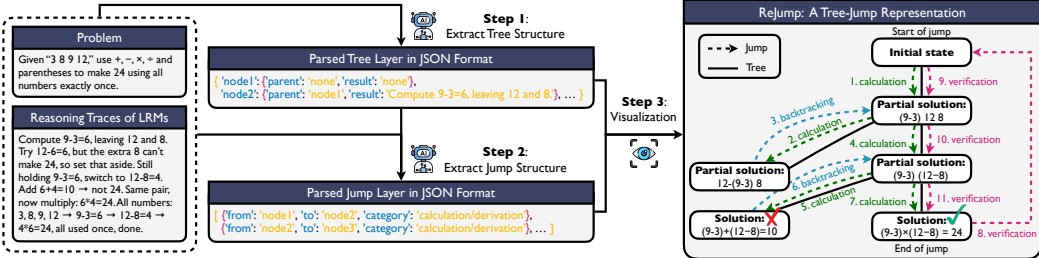

(a) ReJump for a Game-of-24 instance. Partial solution here is defined as an intermediate arithmetic expression.

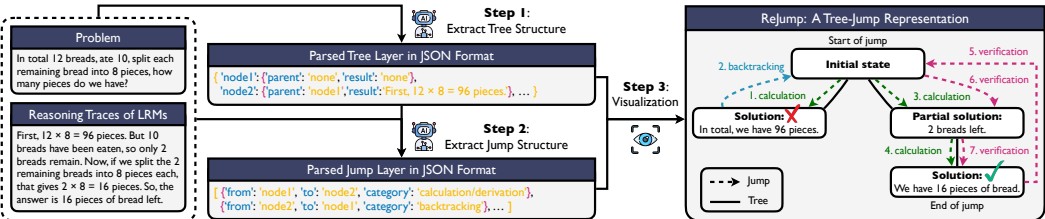

(b) ReJump for a math word problem. Partial solution here is defined as an intermediate computed results.

Figure 7: **Illustration of ReJump representation of reasoning traces for two different tasks.** These examples are crafted for demonstration purposes. Nodes represent partial solutions, and tree edges indicate prerequisite relationships. The dashed jump traces the solvers reasoning trajectory, with transitions labeled by action type: calculation/derivation, verification, or backtracking, highlighted in different colors.

- **Tree layer (structure)**: We define a tree $T = (V, E)$, where $V = \{v_i\}_{v=0}^{|V|}$ is the set of nodes and $E$ the set of edges. Following Yao et al. (2023), each node $v \in V$ represents a partial solution, and an edge $e \in E$ indicates that the parent's partial solution is a direct prerequisite for the child's. Let $S_{\text{leaf}}(T) \subset V$ denote the set of leaf nodes in $T$, each representing a single solution attempt, either a completed solution under one approach or a deadend where the approach failed to yield a correct or full solution. Among these, we define

$S^\star_{\text{leaf}}(T) = \{v \in S_{\text{leaf}}(T) : v \text{ encodes a fully correct solution}\}$ as the subset of leaves that represent correct solutions.

- **Jump layer (action)**: Let $\boldsymbol{i} = (i_0, i_1, \ldots, i_K)$ denote the sequence of reasoning steps, where $i_k$ refers to the index of $k$-th visited node in the tree. The jump starts at $v_{i_0}$ (the root) and ends at $v_{i_K}$ (the final solution). Each transition between consecutive steps $(i_k, i_{k+1})$ is labeled with an action type $\phi_k \in \{\texttt{calc}, \texttt{verify}, \texttt{backtrack}\}$, where $k = 1, \ldots, K-1$. Here, $\texttt{calc}$ refers to generating an intermediate step via calculation or derivation; $\texttt{verify}$ denotes revisiting a node to check its correctness; and $\texttt{backtrack}$ indicates returning to a previous node to explore an alternative reasoning path. A jump is the pair $W = (\boldsymbol{i}, \boldsymbol{\phi})$, fully specifying both the node sequence and how the solver moves through the tree. A *derived solution step* is any step in the jump that reaches a leaf node via a $\texttt{calc}$ transition. Even if the leaf has been visited before, it still counts as a derived step if reached via $\texttt{calc}$; by contrast, visits for verification do not count. We mathematically define the sequence of such steps as $\boldsymbol{i}_{\text{leaf}}(T, W) = (i_{k_1}, \ldots, i_{k_M})$, where each $v_{i_{k_j}} \in S_{\text{leaf}}(T)$ and the corresponding transition is $\phi_{k_j} = \texttt{calc}$, for all $j = 1, \ldots, M$. Among these, we further define the *correct derived solution steps* as $\boldsymbol{i}^\star_{\text{leaf}}(T, W)$, a subsequence of $\boldsymbol{i}_{\text{leaf}}(T, W)$, consisting of indices $i_{k_j}$ such that $v_{i_{k_j}} \in S^\star_{\text{leaf}}(T)$.

**Notations.** Define $\text{consec}(\cdot)$ as an operator that takes a sequence as input and returns the set of all consecutive pairs; that is, for a sequence $(x_1, \ldots, x_M)$, $\text{consec}(x_1, \ldots, x_M) = \{(x_1, x_2), \ldots, (x_{M-1}, x_M)\}$. Let $\text{set}(\cdot)$ be the operator that convert a sequence into a set. For any sequence $\boldsymbol{i} = (i_1, \ldots, i_n)$, we denote its $j$-th element (1-based indexing) by $\boldsymbol{i}[j]$. When consider $N$ generated reasonings and their corresponding ReJump, we use subscript $(n)$ to denotes the $n$-th reasoning instance.

Next, we present more rigorous definitions of the evaluation metrics.

## B.1 EVALUATION METRICS

This tool enables analysis of LLM reasoning behaviors, including solution diversity, exploration-exploitation trade-off, effectiveness in identifying correct paths, frequency of overthinking, forgetting, and verification. These aspects are quantified using the following metrics, computed across all reasonings and their corresponding ReJumps within a task.

**Solution Count ($\#_{\text{solution}}$).** This metric quantifies the model's ability to discover diverse solution attempts, measured by the number of leaf nodes in the reasoning tree. We define $\#_{\text{solution}}(\{T\}) = |S_{\text{leaf}}(T)|$ as the total number of leaf nodes representing distinct solutions within a single tree $T$. To assess the average performance over $N$ reasoning instances for a given task, we calculate the Average Solution Count as the mean number of unique solutions found across all instances: $\#_{\text{solution}}(\{T^{(n)}\}^N_{n=1}) = \sum^N_{n=1} \#_{\text{solution}}(\{T^{(n)}\})/N$.

**Jump Distance ($d_{\text{jump}}$).** This metric captures the exploration-exploitation tradeoff of the reasonings by averaging how far the reasoning "jumps" between newly visited leaf nodes (see Fig. 2a). Define $d(u, v)$ as the number of edges on the path between nodes $u, v \in V$. The jump distance of a single jump trace is $d_{\text{jump}}(\{(T, W)\}) = \frac{1}{|\text{consec}(\boldsymbol{i}_{\text{leaf}}(T,W))|} \sum_{(i_j, i_l) \in \text{consec}(\boldsymbol{i}_{\text{leaf}}(T,W))} d(v_{i_j}, v_{i_l})$, and the task-level average is $d_{\text{jump}}(\{(T^{(n)}, W^{(n)})\}^N_{n=1}) = \frac{1}{N} \sum^N_{n=1} d_{\text{jump}}(\{(T^{(n)}, W^{(n)})\})$.

**Success Rate ($r_{\text{success}}$).** The metric measures how frequently a reasoning path produces a correct solution. For a single reasoning with tree $T$ and jump $W$, the success rate is computed as $r_{\text{success}}(\{(T, W)\}) = |\boldsymbol{i}^\star_{\text{leaf}}(T, W)|/|\boldsymbol{i}_{\text{leaf}}(T, W)|$, i.e., the proportion of newly visited leaf nodes that are correct solutions. The overall average is computed across all $N$ reasoning instances: $r_{\text{success}}(\{(T^{(n)}, W^{(n)})\}^N_{n=1}) = \frac{1}{N} \sum^N_{n=1} r_{\text{success}}(\{(T^{(n)}, W^{(n)})\})$.

**Verification Rate ($r_{\text{verify}}$).** This metric quantifies how frequently the model invokes verification steps during its reasoning process. For a given reasoning instance with jump $W = (\boldsymbol{i}, \boldsymbol{\phi})$, the verification rate is computed as the number of $\texttt{verify}$ transitions divided by the total number of steps in the jump: $r_{\text{verify}}(\{W\}) = \sum_{\phi_k \in \text{set}(\boldsymbol{\phi})} \mathbb{I}\{\phi_k = \texttt{verify}\}/(K-1)$. We report the average verification rate across all $N$ reasoning instances. $r_{\text{verify}}(\{W^{(n)}\}^N_{n=1}) = \sum^N_{n=1} r_{\text{verify}}(\{W^{(n)}\})/N$.

Table 7: Prompt sensitivity of the six metrics. Value close to $1$ indicate low sensitivity to prompt wording.

| Model | #sol | $d_{\text{jump}}$ | $r_{\text{success}}$ | $r_{\text{verify}}$ | $r_{\text{overthinking}}$ | $r_{\text{forget}}$ |
|---|---|---|---|---|---|---|
| QwQ-32B | 1.45 | 1.04 | 1.08 | 1.39 | 1.21 | 0.99 |
| Phi-4-Reasoning-Plus | 0.82 | 0.93 | 1.25 | 0.96 | 0.95 | 1.08 |

**Overthinking Rate ($r_{\text{overthinking}}$).** This metric quantifies the extent of unnecessary exploration after a correct solution has already been found. For a given reasoning instance with tree $T$ and jump $W$, let $k_0^\star$ denote the first index in $i_{\text{leaf}}^\star(T, W)$, the step at which a correct leaf is first reached. The overthinking rate is defined as the fraction of newly visited leaf nodes that appear *after* this first correct solution: $r_{\text{overthinking}}(\{(T, W)\}) = |\{i_k \in \text{set}(i_{\text{leaf}}(T, W)) : k > k_0^\star\}|/|i_{\text{leaf}}(T, W)|$. In other words, it measures how many additional leaf nodes are explored via `calc` transitions after a correct solution has been identified. The task-level overthinking rate is then given by the average over all $N$ reasoning instances: $r_{\text{overthinking}}(\{(T^{(n)}, W^{(n)})\}_{n=1}^N) = \frac{1}{N} \sum_{n=1}^N r_{\text{overthinking}}(\{(T^{(n)}, W^{(n)})\})$.

**Forgetting Rate ($r_{\text{forget}}$).** This metric tracks how often the model forgets its earlier steps and recomputes a previously derived result. Specifically, forgetting is flagged when a previously visited leaf node is revisited via a `calc` transition (see Fig. 2b). We define a binary indicator for each reasoning instance as $\mathbf{1}_{\text{forget}}(T, W) = 1 - \prod_{m=2}^M \left( \prod_{j=1}^{m-1} \mathbb{I}\{i_{\text{leaf}}[j] \neq i_{\text{leaf}}[m]\} \right)$, which returns 1 if any earlier leaf is re-entered, and 0 otherwise. The forgetting rate is then reported as the proportion of instances where forgetting occurred: $r_{\text{forget}}(\{(T^{(n)}, W^{(n)})\}_{n=1}^N) = \frac{1}{N} \sum_{n=1}^N \mathbf{1}_{\text{forget}}(T^{(n)}, W^{(n)})$.

## B.2 PROMPT SENSITIVITY.

To assess the robustness of our metrics to variations in prompt wording, we perform a prompt-sensitivity ablation. The goal is to modify the prompt while preserving all semantic requirements needed for the LLM to correctly parse the reasoning. We construct three meaning-preserving prompt variants: (i) the original *default* prompt, (ii) a *shuffle* variant that permutes the order of instructions describing the three transition types (calculation, backtrack, verification), and (iii) a *rephrase* variant that rewrites the instructions in natural language without altering their semantics.

Let $\text{std}_{\text{seed}}(M)$ denote the standard deviation of metric $M$ under the default prompt across three runs with different random seeds, and let $\text{std}_{\text{prompt}}(M)$ denote the standard deviation of $M$ across the three prompt variants under a fixed seed. We define the *Prompt Sensitivity* of metric $M$ as

$$\text{PromptSensitivity}(M) = \frac{\text{std}_{\text{prompt}}(M)}{\text{std}_{\text{seed}}(M)}.$$

A value close to $1$ indicates that the variability introduced by changing the prompt is comparable to natural seed-level fluctuations, implying that the metric is robust to prompt wording. The prompt-sensitivity results for all six metrics and two representative models are reported in Tab. 7.

## B.3 REDUNDANCY ANALYSIS

To assess whether the six proposed metrics capture complementary aspects of reasoning behavior, we conduct an information-theoretic redundancy analysis. For each metric $M$, we compute its redundancy as

$$\text{Redundancy}(M) = \frac{I(M; \text{others})}{H(M)},$$

where $H(M)$ is the entropy of $M$ and $I(M; \text{others})$ is the mutual information between $M$ and the remaining metrics. Lower values indicate that the metric contains information not recoverable from the others. Redundancy scores for MATH-500 and Game of 24 are reported in Tab. 8.

Table 8: Redundancy scores of the six metrics on MATH-500 and Game of 24. Lower values indicate less dependence on other metrics.

| Dataset | #sol | $d_{\text{jump}}$ | $r_{\text{success}}$ | $r_{\text{verify}}$ | $r_{\text{overthink}}$ | $r_{\text{forget}}$ |
|---|---|---|---|---|---|---|
| MATH-500 | 0.789 | 0.624 | 0.437 | 0.105 | 0.944 | 0.277 |
| Game of 24 | 0.731 | 0.761 | 0.882 | 0.687 | 0.840 | 0.840 |

Table 9: Direct LLM extraction vs. ReJump-Extractor on the synthetic ground-truth dataset. Lower MAE and higher accuracy are better.

| Method | #sol (MAE ↓) | $r_{\text{success}}$ (MAE ↓) | $r_{\text{forget}}$ (Acc ↑) |
|---|---|---|---|
| Direct Query | 2.12 | 0.11 | 0.87 |
| ReJump-Extractor | **0.62** | **0.08** | **0.89** |

### B.4 COMPARISON TO SIMPLER LLM-BASED ANALYSIS

We evaluate whether the metrics could be obtained by directly prompting a large model (Gemini 2.5 Pro), instead of extracting ReJump trees. For metrics not defined on the graph (#solution, $r_{\text{success}}$, $r_{\text{forget}}$), we use the synthetic ground-truth dataset introduced in Sec. 4.1, where each reasoning instance is manually annotated with correct metric values. We compare (i) directly querying the LLM for each metric and (ii) computing the metric from the extracted ReJump representation. As shown in Tab. 9, ReJump provides substantially more accurate measurements.

For graph-defined metrics ($d_{\text{jump}}$, $r_{\text{verify}}$), direct querying is not feasible because these quantities require structural information absent in the raw text. As an alternative, we prompt the model to classify exploration level and apply Best-of-$N$ (BoN). As shown in Tab. 10, BoN with ReJump consistently achieves the strongest performance, demonstrating that ReJump provides a more faithful basis for analysis.

## C EXTENDED SEC. 4: REJUMP-EXTRACTOR

In addition to the illustrative example on a math word problem, we include one more example showcasing the construction and visualization of ReJump, with both examples shown in Fig. 7.

### C.1 LLM PROMPTS FOR REJUMP CONSTRUCTION

In this section, we present the prompt used by the LLM to parse results across all experiments.

We use different prompts for the two datasets. Listing 1 and Listing 2 show the prompts used to extract the tree and jump from the generated reasoning for MATH-500, respectively, while Listing 3 and Listing 4 show the corresponding prompts for Game of 24.

```
1 def get_tree_prompt_math(input_str, output_str):
2     return f"""
3 Your task is to analyze a detailed thinking process for solving a math
      problem (provided below) and convert it into a reasoning tree. This
      tree must represent the **chronological flow of solving substantive,
      mathematically well-posed subproblems or distinct attempts**,
      starting from an initial state and culminating in answering the
      original question.
4
5 Represent this structure as a **single JSON object** where keys are
      unique node IDs (e.g., "node1", "node2") and values are node objects
      detailing each state or subproblem attempt.
6
7 **Core Principles for Tree Generation:**
8
```

Table 10: Majority Vote vs. BoN using direct exploration classification vs. BoN with ReJump. Higher is better.

| Model | Majority Vote | BoN (Direct) | BoN (ReJump) |
|---|---|---|---|
| Phi-4-Reasoning-Plus | 0.77 | 0.77 | **0.84** |
| QwQ-32B | 0.76 | 0.82 | **0.82** |

9 * **Chronological Flow & Dependency:** The tree follows the order of
    substantive steps/attempts in the reasoning. Parent links indicate
    the preceding step whose `Result` provides necessary mathematical
    input.
10   **BRANCHING AND SUBSTEP RULE:**
11   - Create a new branch **if and only if** the reasoning process
    explicitly abandons or gives up on a previous approach and then
    starts a new, distinct solution plan. In other words, a new branch is
     created always and only when the previous line of reasoning is
    abandoned and a fundamentally different method is attempted. The new
    branch should start from the most recent shared node. Even if the
    solver does not immediately abandon the previous approach, we still
    consider it an Abandoned Attempt Node and mark it with [Path
    abandoned] if a different method is initiated that departs from the
    original direction.
12   - Importantly, whenever a new branch is created, the leaf node where
    the previous method ended must be explicitly marked with [Path
    abandoned].
13   - Conversely, if the current node is marked with [Path abandoned], a
    new branch must always be created.
14   - Importantly, for all subproblems or calculations within a single
    uninterrupted attempt, even if subcalculations are mathematically
    independent, represent these steps sequentially in the order they are
     performed in the reasoning: each node's parent must be the
    immediately preceding node within that attempt.
15    That is, substeps within any one attempt always form a single chain.
16 * **Substantive, Well-Posed Steps Only:** Nodes must represent **major**
    intermediate calculations or logical deductions constituting a clear,
     self-contained mathematical task (like a homework sub-problem). **
    Aggressively filter out** setup actions, strategy descriptions,
    narrative, verification, and trivial calculations/manipulations.
    Minor algebraic steps within a larger logical step must be grouped.
17 * **Include Failed Attempts:** Represent distinct, substantive
    calculation or derivation attempts that were **explicitly abandoned**
     in the reasoning as separate nodes in the chronological flow. **Do
    not filter these out.**
18 * **Focus on Mathematical Task:** Intermediate `Problem` fields must
    state a clear mathematical objective based on **all necessary given
    mathematical conditions and inputs**, avoiding descriptions of the
    reasoner's process or assumptions *within the Problem text*.
19 * **Special Final Node:** The node performing the last calculation for
    the final answer uses the original problem statement as its `Problem
    `.
20
21 **Node Object Structure:**
22 Each node object must contain: `Problem`, `parent`, `Result`.
23
24 1.  **`Problem` (String): Defines the specific mathematical task for this
     node.**
25   * **`node1` (Root):** Must be exactly "Initial State".
26   * **Intermediate Nodes (`node2` to `node(N-1)`):** Formulates a **
    clear, mathematically well-posed, and self-contained task
    representing a substantive step or distinct attempt.** Each node
    represents achieving a distinct intermediate objective through
    calculation or deduction.

27        * **Format:** Start with "Given..." listing **all essential mathematical conditions, constraints, equations, and input values** ( often from parent `Result` or established context like 'point P is on curve C') needed to define and solve *this specific task*. End with a specific mathematical question/instruction (e.g., "Calculate...", " Solve...", "Derive...").

28        * **Content:** The formulation must focus purely on the ** mathematical task**, making it **understandable and solvable in isolation** like a homework sub-problem, using only the provided " Given..." information and general mathematical knowledge. **CRITICAL RULE:** The `Problem` text **must not** include descriptions of the reasoner's strategy, assumptions, or procedural instructions reflecting the reasoning flow. State only the necessary mathematical conditions and the objective. The task must be **substantive**. ** CRITICAL FILTERING RULE:** **DO NOT** create separate nodes for individual algebraic manipulations... [rest of filtering rule stays the same – GROUP minor operations]. Also filter out narrative, setup, verification. No meta-tags or node ID references.

29    * **`nodeN` (Final Calculation Node):** **This node represents the very last calculation step that produces the final answer.** Its ` Problem` field **must contain the verbatim Original Problem Statement .**

30

31 2.  **`parent` (String): Identifies the immediately preceding substantive step providing necessary input.**

32    * **`node1`:** Must be "none".

33    * **Other Nodes (`node2` to `nodeN`):** Must be the ID of the node whose `Result` provides the direct mathematical prerequisite for the task in the current node's `Problem`. (For abandoned attempts, the parent is the node preceding the attempt).

34

35 3.  **`Result` (String): Records the mathematical outcome of completing the task.**

36    * **`node1`:** "Original problem statement provided **as** context." (or similar).

37    * **Intermediate Nodes (`node2` to `node(N-1)`):** The direct mathematical outcome of achieving the task defined in `Problem`. Summarizes the result of grouped operations.

38    * **Abandoned Attempt Nodes:** Must state any partial outcome and explicitly end with "[Path abandoned]".

39    * **`nodeN` (Final Calculation Node):** Must be the **final answer** to the Original Problem Statement.

40

41 **Instructions for Analysis:**

42 1.  **Inputs:** Use the "Original Problem Statement" and "Input Reasoning Process".

43 2.  **Identify & Filter Steps:** Read the reasoning chronologically. Identify **major** calculation phases, key logical deductions, or distinct attempts. **Crucially, ensure that distinct, substantive attempts explicitly marked as abandoned in the reasoning are identified and *not* filtered out.** Apply the **CRITICAL FILTERING and GROUPING RULES** aggressively: Group sequences of trivial algebraic steps into the single larger objective they serve. Filter out non-mathematical content, setup, strategy descriptions/ assumptions-as-actions, and verification. Only create nodes for the remaining substantive steps and distinct abandoned attempts.

44 3.  **Create Nodes Sequentially:**

45    * Create `node1`.

46    * For each identified **substantive step/objective/attempt** *before* the final answer calculation: Create the corresponding intermediate node (`node2`, `node3`, ...). Determine `parent`. Formulate the ` Problem` strictly according to Rule 1 (well-posed, self-contained task including **all necessary conditions/constraints**, no process descriptions). Record `Result`. Link abandoned attempt nodes chronologically.

```
47      * For the **final calculation step**: Create `nodeN`. Determine `
        parent`. Set `Problem` to verbatim Original Problem Statement. Set `
        Result` to final answer.
48  4.   **Formatting:** Use LaTeX (`$...$`) for all math notation.
49  5.   **Output:** Produce a single JSON object.
50
51  ---
52  **BEGIN ORIGINAL PROBLEM STATEMENT**
53  ---
54  {input_str}
55  ---
56  **END ORIGINAL PROBLEM STATEMENT**
57  ---
58
59  ---
60  **BEGIN INPUT REASONING PROCESS**
61  ---
62  {output_str}
63  ---
64  **END INPUT REASONING PROCESS**
65  ---
66
67  Generate the JSON output based on these instructions.
68      """
69
70  # After obtaining the tree, use a separate prompt to evaluate the
        correctness of each leaf node for refining the tree.
71  def get_result_parsing_and_comparison_prompt(result_string,
        ground_truth_string):
72      return f"""You are an expert AI assistant. Your task is to analyze a
        'Result' string from a mathematical reasoning step and compare its
        final numerical answer to a 'Ground Truth' value.
73
74  Instructions:
75  1.   Extract the final numerical value(s) from the 'Result' string.
76      -  If multiple numbers are present, focus on the one that seems to be
        the conclusive answer of that step.
77      -  Handle approximations (e.g., "approx 46.0", "is about 3.14").
78      -  If the result explicitly states abandonment (e.g., "[Path abandoned
        ]"), extract the numerical value derived *before* abandonment, if any
        . If no clear numerical value was derived, use "N/A" for the parsed
        value.
79      -  If no specific numerical answer can be clearly identified, use "N/A
        " for the parsed value.
80
81  2.   Compare the extracted numerical value with the 'Ground Truth' value.
82      -  The comparison should determine if they are essentially the same,
        considering potential minor differences in formatting or precision (e
        .g., "46" vs "46.0", "1.03" vs "1.035" if context implies rounding).
83      -  If the parsed value is "N/A", the comparison result should be "
        NOT_APPLICABLE".
84      -  If the ground truth is empty or clearly not a comparable numerical
        value, and the parsed value is numerical, consider it a "MISMATCH"
        unless specified otherwise.
85
86  3.   Output a single JSON object with two keys:
87      -   `"parsed_value"`: The extracted numerical value as a string (e.g
        ., "46", "3.14", "N/A").
88      -   `"match_status"`: A string indicating the comparison result. Must
         be one of: "MATCH", "MISMATCH", "NOT_APPLICABLE".
89
90  Example:
91  Result string: "Using the approximations, $tan x^\circ \\approx \\frac
        {{1.3270 + 6.3138}}{{1.3270 \\times 6.3138 - 1}} \\approx \\frac
        {{7.6408}}{{8.381 - 1}} \\approx \\frac{{7.6408}}{{7.381}} \\approx
```

```
     1.0355$. This implies $x \\approx arctan(1.0355) \\approx 46.0^\circ$
      . [Path abandoned]"
92 Ground Truth string: "46"
93 Expected JSON Output: {{"parsed_value": "46.0", "match_status": "MATCH"}}
94
95 Result string: "The answer is $y=3$."
96 Ground Truth string: "3.0"
97 Expected JSON Output: {{"parsed_value": "3", "match_status": "MATCH"}}
98
99 Result string: "The calculation leads to $10/2 = 5$. However, this path
      is incorrect."
100 Ground Truth string: "7"
101 Expected JSON Output: {{"parsed_value": "5", "match_status": "MISMATCH"}}
102
103 Result string: "[Path abandoned] No value obtained."
104 Ground Truth string: "10"
105 Expected JSON Output: {{"parsed_value": "N/A", "match_status": "
      NOT_APPLICABLE"}}
106
107 ---
108 Result string to analyze:
109 {result_string}
110
111 Ground Truth value:
112 {ground_truth_string}
113 ---
114
115 JSON Output:"""
```

Listing 1: Prompt for extracting a tree from the reasoning trace in JSON format for MATH-500.

```
1 def get_jump_prompt(input_str, output_str, tree_json):
2     return f"""
3 You are an AI assistant specialized in analyzing mathematical reasoning
      processes. Your task is to trace the provided reasoning text against
      a structured reasoning tree and generate a "walk" representing the
      trajectory of the thought process.
4
5 **Inputs:**
6
7 1.  **Problem Description:**
8     ```
9     {input_str}
10    ```
11 2.  **Reasoning Text:** A step-by-step textual explanation of how the
      problem was solved, including potential errors, corrections,
      explorations of different paths, and verifications.
12    ```text
13    {output_str}
14    ```
15 3.  **Reasoning Tree:** A JSON object representing the structured steps
      and dependencies of the solution(s). Each key is a node ID, and the
      value contains information about that step, including its parent node
       and specifically a "Problem" field describing the task of that node.
16    ```json
17    {tree_json}
18    ```
19
20 **Task:**
21
22 Analyze the `Reasoning Text` to determine the sequence in which the
      solver mentally visited or considered the steps represented by the
      nodes in the `Reasoning Tree`. Identify the transitions between these
       nodes and categorize each transition. **Crucially, for verification
```

steps, visiting a node X implies the text shows evidence of re-doing the specific task described in the "Problem" field of node X.**

**Output Format:**

Generate a JSON list of dictionaries, where each dictionary represents a single step in the reasoning walk. Each dictionary must have the following keys:

* `from`: The ID (string) of the node the reasoning is moving *from*.
* `to`: The ID (string) of the node the reasoning is moving *to*.
* `category`: A string indicating the type of transition. Must be one of:
    * `calculation/derivation`: Represents forward progress in the reasoning, moving from one step to the next logical step (often parent to child in the tree) to derive new information or explore a solution path.
    * `backtracking`: Represents abandoning a current line of thought or calculation (often because it's incorrect, inefficient, or a dead end) and returning to a previous state (node) to try a different approach. This is typically a move from a node to one of its ancestors (not necessarily the direct parent).
    * `verification`: Represents checking or confirming a result or step **by re-doing the work associated with previous nodes**. This is determined based on the text:
        * **Specific Re-work:** If the text explicitly describes actions that precisely match the **problem description** defined within an intermediate node (e.g., node X) as part of checking a later result (node Z), trace the path reflecting that specific re-work (e.g., Z -> X -> Z). This requires clear evidence in the text of **re-solving the problem defined in node X**.
        * **General Check:** If the text indicates verification of a result (node Z) but ***does not*** show actions matching the specific **problem description** of any intermediate node, interpret this as checking consistency with the initial problem statement/conditions (node 1). Represent this path as Z -> 1 -> Z. ***Note: Simply using a formula or result from a previous node (e.g., node X) without showing the steps to re-solve the problem defined in node X does NOT count as re-doing the work of node X.***

**Instructions:**

1.  Read the `Reasoning Text` carefully, paying attention to the flow, changes in direction, calculations, statements of intent (e.g., "Let me **try**...", "No, that's wrong...", "Let me verify..."), and results.
2.  Map segments of the `Reasoning Text` to the corresponding nodes in the `Reasoning Tree`. Use the "Problem" and "Result" fields in the tree nodes to help with mapping *initial* derivations.
3.  Identify the sequence of nodes visited or considered based on the flow of the `Reasoning Text`.
4.  For each transition from one node (`from`) to the next (`to`) in the sequence, determine the appropriate `category` based the definitions above.
5.  Pay close attention to parts of the reasoning text that indicate:
    * Starting a calculation or derivation (maps to `calculation/derivation`).
    * Realizing an error or deciding a path is not fruitful and returning to an earlier idea (maps to `backtracking`).
    * Re-checking results (maps to `verification`). **When mapping `verification`:** First, check if the text describes actions that precisely match the **problem description** of an intermediate node (Node X), essentially re-doing the work defined in that node. If yes, trace the walk through the node being re-worked (e.g., Z -> X -> Z). If the text indicates verification but ***does not*** show such a specific re-work of a prior node's problem, assume it implies checking against the initial problem conditions (node 1) **and**

```
       represent the path as Z -> 1 -> Z. Remember: Simply *using* a result
       or formula from node X does not qualify as re-doing the problem of
       node X according to this definition.
47 6.  The walk should reflect the *actual* path taken in the `Reasoning
       Text`, including explorations of dead ends (like `node2` in the
       example) and subsequent backtracking.
48
49     **Mandatory Backtracking Rule:**
50     Only when the reasoning process explicitly abandons or gives up on
       the current approach at node A and then starts a new, distinct
       attempt at node B must you include a backtracking transition from A
       to the parent of B, followed by a calculation/derivation transition
       from the parent of B to B. Never allow a direct calculation/
       derivation transition from A to B in these cases. Do not include
       backtracking transitions except in such abandonment cases.
51
52 7.  Ensure the output is strictly the JSON list as specified, with no
       additional explanatory text.
53 8. The output MUST be perfectly valid JSON, parseable by standard
       libraries.
54 9. The walk must always start at node1: The first transition in your
       output should always be `"from": "node1"`, `"to": ...`. Never use `"
       from": "none"`, `"from": null`, or any other alternative. Assume
       reasoning always conceptually begins at node1.
55
56 **Example Analysis (Based on Provided Inputs with Stricter Verification
       Logic):**
57
58 * Reasoning starts, defining the problem (maps to `node1`).
59 * Text explores calculating AB with specific points (maps to `node2`). `
       node1` -> `node2` (`calculation/derivation`).
60 * Text says "That seems messy... Let me think differently." and abandons
       the `node2` approach, returning to the setup phase (conceptually `
       node1`). `node2` -> `node1` (`backtracking`).
61 * Text introduces symmetry and points B(x,y), C(-x,y) (maps to `node3`).
       `node1` -> `node3` (`calculation/derivation`). This step involves *
       doing* the problem in `node3` (calculating distances).
62 * Text derives relationship between AB and BC, sets them equal (maps to `
       node4`). `node3` -> `node4` (`calculation/derivation`).
63 * Text solves for x and y using parabola equation (maps to `node5`). `
       node4` -> `node5` (`calculation/derivation`).
64 * Text calculates final side length (maps to `node6`). `node5` -> `node6`
        (`calculation/derivation`).
65 * Text says "Let me verify with the distance." It then shows:
66    1.  `AB = sqrt(x^2 + y^2) = ...` This ***uses*** the formula derived
       in `node3` and values from `node5`. It does ***not*** show a re-
       derivation of the distance formula as described in `node3`'s problem
       ("Calculate the distances...").
67    2.  `BC is 2x = ...` This ***uses*** the formula derived in `node3`
       and value from `node5`. It does ***not*** show a re-derivation.
68 * **Applying the strict verification rule:** Does the text show actions
       matching the *problem description* of an intermediate node (like re-
       deriving the formulas as defined in `node3`'s problem, or re-solving
       for x,y as defined in `node5`'s problem)? **No**, the text only shows
        the *application* of results from previous nodes.
69 * Therefore, according to the rule, since no specific re-work of a prior
       node's **problem** is detailed, we default to the **General Check**
       case. The path should be represented as checking the final result (`
       node6`) against the initial state (`node1`).
70 * The expected verification path for this text, under this strict
       interpretation, would be: `node6` -> `node1` (`verification`),
       potentially followed by `node1` -> `node6` (`verification`) or
       repeated. A simple `node6 -> node1 -> node6` sequence for the overall
        verification check is likely.
71
```

```
72 **Final Output Request:**
73
74 Now, analyze the provided inputs ('{{problem_description}}', '{{
       reasoning_text}}', '{{reasoning_tree_json}}') using **this strict
       interpretation of verification** (visiting a node requires re-doing
       its specific "Problem") and generate the reasoning walk as a JSON
       list. Output *only* the JSON list.
75     """
```

Listing 2: Prompt used to extract jump from reasoning as a JSON structure for MATH-500.

```
1 def get_tree_prompt(input_str, output_str):
2     return f"""
3 Given the problem statement and reasoning process below. Your task is to
      analyze a detailed thinking process for solving a math problem (
      provided below) and convert it into a reasoning tree. **Do not try to
       solve the problem yourself, fully use the given reasoning process
      and just convert it!**
4
5 ---
6 **BEGIN ORIGINAL PROBLEM STATEMENT**
7 ---
8 {input_str}
9 ---
10 **END ORIGINAL PROBLEM STATEMENT**
11 ---
12
13 ---
14 **BEGIN INPUT REASONING PROCESS**
15 ---
16 {output_str}
17 ---
18 **END INPUT REASONING PROCESS**
19 ---
20
21 Here are some instructions:
22
23 **Node Object Structure:**
24
25 Each node object must contain: 'Problem', 'parent', 'Result'.
26
27 1. **'Problem' (String): A partial solution containing the four numbers
      and any calculation has been tried. Only use numbers, + - * / and
      parentheses.
28
29 * **'node1' (Root):** Must be exactly the four initial numbers in the
      problem. For example, "9,3,12,8".
30
31 * **Non-leaf Nodes:** Each node describes the partial solution being
      explored. For example, for problem 9,3,12,8, an intermediate node "
      9-3, 12, 8" means that we have tried (9-3), and need to try 2 more
      calculations with numbers 12 and 8 to get 24. Give all these nodes
      indexes number to keep tracking (after node1).
32
33 * **Leaf node:** **This node represents the very last calculation step
      that produces the final answer after three calculation steps.** For
      example, for problem 9,3,12,8, this could be "9-3+128", which is a
      leaf node that is unsuccessful. Another successful leaf node could be
       "(9-3)*(128)". Also use an index number for each one (after node1).
34
35 Pay attention that the problem statement of each node should be unique.
      If two nodes have the same description (i.e., the same partial
      calculation and the numbers not calculated so far), merge them into
      one.
36
```

```
37  2. **`parent` (String):
38
39  * **`node1` (root):** Must be None.
40
41  * **Other nodes:** Must be the previous partial solution that the current
       node builds on. For example, the parent of the node "9-3, 12, 8" is
       "9,3,12,8". But here just use the index number to indicate the index
       of its parent node.
42
43  3. **`Result` (String):
44
45  * **`root`:** None.
46
47  * **Intermediate Nodes:** None.
48
49  * **Leaf node** Must be the **final answer**. For example, the result of
       node "9-3+12-8" is 10. Written in latex.
50
51  Please generate a single JSON output. This output must be a **single JSON
       object** where keys are unique node IDs (e.g., "node1", "node2",
       corresponding to the index numbers assigned to track the nodes) and
       values are the node objects (containing 'Problem', 'parent', 'Result
       ') as detailed above.
52
53          """
```

Listing 3: Prompt used to extract tree from reasoning as a JSON structure for Game of 24.

```
1  def get_jump_prompt(input_str, output_str, tree_json):
2      return f"""
3  You are an AI assistant specialized in analyzing mathematical reasoning
      processes. Your task is to trace the provided reasoning text against
      a structured reasoning tree and generate a "walk" representing the
      trajectory of the thought process.
4
5  **Inputs:**
6
7  1.  **Problem Description:**
8      ```
9      {input_str}
10     ```
11 2.  **Reasoning Text:** A step-by-step textual explanation of how the
      problem was solved, including potential errors, corrections,
      explorations of different paths, and verifications.
12     ```text
13     {output_str}
14     ```
15 3.  **Reasoning Tree:** A JSON object representing the structured steps
      and dependencies of the solution(s). Each key is a node ID, and the
      value contains information about that step, including its parent node
       and specifically a "Problem" field describing the task of that node.
16     ```json
17     {tree_json}
18     ```
19
20 **Task:**
21
22 Analyze the `Reasoning Text` to determine the sequence in which the
      solver mentally visited or considered the steps represented by the
      nodes in the `Reasoning Tree`. Identify the transitions between these
       nodes and categorize each transition. **Crucially, for verification
      steps, visiting a node X implies the text shows evidence of re-doing
      the specific task described in the "Problem" field of node X.**
23
24 **Output Format:**
```

25

26 Generate a JSON list of dictionaries, where each dictionary represents a
    single step in the reasoning walk. Each dictionary must have the
    following keys:

27

28 * `from`: The ID (string) of the node the reasoning is moving *from*.
29 * `to`: The ID (string) of the node the reasoning is moving *to*.
30 * `category`: A string indicating the type of transition. Must be one of:
31    * `calculation/derivation`: Represents forward progress in the
      reasoning, moving from one step to the next logical step (often
      parent to child in the tree) to derive new information or explore a
      solution path.
32    * `backtracking`: Represents abandoning a current line of thought or
      calculation (often because it's incorrect, inefficient, or a dead end
      ) and returning to a previous state (node) to try a different
      approach. This is typically a move from a node to one of its
      ancestors (not necessarily the direct parent).
33    * `verification`: Represents checking or confirming a result or step
      **by re-doing the work associated with previous nodes**. This is
      determined based on the text:
34        * **Specific Re-work:** If the text explicitly describes actions
      that precisely match the **problem description** defined within an
      intermediate node (e.g., node X) as part of checking a later result (
      node Z), trace the path reflecting that specific re-work (e.g., Z ->
      X -> Z). This requires clear evidence in the text of **re-solving the
       problem defined in node X**.
35        * **General Check:** If the text indicates verification of a
      result (node Z) but ***does not*** show actions matching the specific
       **problem description** of any intermediate node, interpret this as
      checking consistency with the initial problem statement/conditions (
      node 1). Represent this path as Z -> 1 -> Z. ***Note: Simply using a
      formula or result from a previous node (e.g., node X) without showing
       the steps to re-solve the problem defined in node X does NOT count
      as re-doing the work of node X.***

36

37 **Instructions:**

38

39 1.  Read the `Reasoning Text` carefully, paying attention to the flow,
      changes in direction, calculations, statements of intent (e.g., "Let
      me **try**...", "No, that's wrong...", "Let me verify..."), and results.
40 2.  Map segments of the `Reasoning Text` to the corresponding nodes in
      the `Reasoning Tree`. Use the "Problem" and "Result" fields in the
      tree nodes to help with mapping *initial* derivations.
41 3.  Identify the sequence of nodes visited or considered based on the
      flow of the `Reasoning Text`.
42 4.  For each transition from one node (`from`) to the next (`to`) in the
      sequence, determine the appropriate `category` based the definitions
      above.
43 5.  Pay close attention to parts of the reasoning text that indicate:
44    * Starting a calculation or derivation (maps to `calculation/
      derivation`).
45    * Realizing an error or deciding a path is not fruitful and returning
       to an earlier idea (maps to `backtracking`).
46    * Re-checking results (maps to `verification`). **When mapping `
      verification`:** First, check if the text describes actions that
      precisely match the **problem description** of an intermediate node (
      Node X), essentially re-doing the work defined in that node. If yes,
      trace the walk through the node being re-worked (e.g., Z -> X -> Z).
      If the text indicates verification but ***does not*** show such a
      specific re-work of a prior node's problem, assume it implies
      checking against the initial problem conditions (node 1) **and**
      represent the path **as** Z -> 1 -> Z. Remember: Simply *using* a result
      **or** formula **from** node X does **not** qualify **as** re-doing the problem of
      node X according to this definition.

6.  The walk should reflect the *actual* path taken **in** the `Reasoning Text`, including explorations of dead ends (like `node2` **in** the example) **and** subsequent backtracking.

    **Mandatory Backtracking Rule:**
    Only when the reasoning process explicitly abandons **or** gives up on the current approach at node A **and** then starts a new, distinct attempt at node B must you include a backtracking transition **from** A to the parent of B, followed by a calculation/derivation transition **from** the parent of B to B. Never allow a direct calculation/ derivation transition **from** A to B **in** these cases. Do **not** include backtracking transitions **except in** such abandonment cases.

7.  Ensure the output **is** strictly the JSON **list as** specified, **with** no additional explanatory text.
8.  The output MUST be perfectly valid JSON, parseable by standard libraries.
9.  The walk must always start at node1: The first transition **in** your output should always be `"from": "node1"`, `"to": ...`. Never use `" from": "none"`, `"from": null`, **or any** other alternative. Assume reasoning always conceptually begins at node1.

**Example Analysis (Based on Provided Inputs **with** Stricter Verification Logic):**

* Reasoning starts, defining the problem (maps to `node1`).
* Text explores calculating AB **with** specific points (maps to `node2`). `node1` -> `node2` (`calculation/derivation`).
* Text says "That seems messy... Let me think differently." **and** abandons the `node2` approach, returning to the setup phase (conceptually `node1`). `node2` -> `node1` (`backtracking`).
* Text introduces symmetry **and** points B(x,y), C(−x,y) (maps to `node3`). `node1` -> `node3` (`calculation/derivation`). This step involves * doing* the problem **in** `node3` (calculating distances).
* Text derives relationship between AB **and** BC, sets them equal (maps to `node4`). `node3` -> `node4` (`calculation/derivation`).
* Text solves **for** x **and** y using parabola equation (maps to `node5`). `node4` -> `node5` (`calculation/derivation`).
* Text calculates final side length (maps to `node6`). `node5` -> `node6` (`calculation/derivation`).
* Text says "Let me verify with the distance." It then shows:
    1.  `AB = sqrt(x^2 + y^2) = ...` This ***uses*** the formula derived **in** `node3` **and** values **from** `node5`. It does ***not*** show a re-derivation of the distance formula **as** described **in** `node3`'s problem ("Calculate the distances...").
    2.  `BC is 2x = ...` This ***uses*** the formula derived in `node3` and value **from** `node5`. It does ***not*** show a re-derivation.
* **Applying the strict verification rule:** Does the text show actions matching the *problem description* of an intermediate node (like re-deriving the formulas as defined in `node3`'s problem, **or** re-solving **for** x,y **as** defined **in** `node5`'s problem)? **No**, the text only shows the *application* of results from previous nodes.
* Therefore, according to the rule, since no specific re-work of a prior node's **problem** **is** detailed, we default to the **General Check** case. The path should be represented **as** checking the final result (`node6`) against the initial state (`node1`).
* The expected verification path **for** this text, under this strict interpretation, would be: `node6` -> `node1` (`verification`), potentially followed by `node1` -> `node6` (`verification`) **or** repeated. A simple `node6 -> node1 -> node6` sequence **for** the overall verification check **is** likely.

**Final Output Request:**

```
74  Now, analyze the provided inputs ('{{problem_description}}', '{{
        reasoning_text}}', '{{reasoning_tree_json}}') using **this strict
        interpretation of verification** (visiting a node requires re-doing
        its specific "Problem") and generate the reasoning walk as a JSON
        list. Output *only* the JSON list.
75      """
```

Listing 4: Prompt used to extract jump from reasoning as a JSON structure for Game of 24.

## C.2 EXAMPLE REJUMP REPRESENTATIONS CONVERTED FROM REAL REASONING TRACES

In this part, we present example ReJump representations derived from real reasoning traces on both the MATH-500 and Game of 24 tasks. Specifically, we randomly select one problem from each task:

- **MATH-500**: The operation $\&$ is defined for positive integers $a$ and $b$ as $a\&b = \dfrac{\sqrt{ab+a}}{\sqrt{ab-b}}$. What is the value of $9\&2$? Express your answer as a common fraction in simplest radical form.
- **Game of 24**: Given the numbers 3, 6, 6, and 7, use each exactly once with basic arithmetic operations $(+, -, \times, \div)$ and optional parentheses to obtain 24.

For MATH500, we present the reasoning traces generated by Grok 3 Mini Beta and DeepSeek-R1 in Listing 5 and Listing 6, together with their corresponding ReJump representations shown in Fig. 8a and Fig. 8b. For Game of 24, we include the reasoning traces produced by DeepSeek-R1, Phi-4-reasoning-plus, and Claude 3.7 Sonnet in Listing 7, Listing 8, and Listing 9, along with their respective ReJump representations visualized in Fig. 9a, Fig. 9b, and Fig. 9c.

```
1  First, the operation is defined as \( a \& b = \frac{\sqrt{a b + a}}{\
       sqrt{a b - b}} \). I need to find the value for \( a = 9 \) and \( b
       = 2 \).
2
3  So, plugging in the values: \( 9 \& 2 = \frac{\sqrt{9 \cdot 2 + 9}}{\sqrt
       {9 \cdot 2 - 2}} \).
4
5  Let me calculate inside the square roots.
6
7  First, \( 9 \cdot 2 = 18 \).
8
9  So, numerator: \( \sqrt{18 + 9} = \sqrt{27} \).
10
11 Denominator: \( \sqrt{18 - 2} = \sqrt{16} \).
12
13 Okay, so \( 9 \& 2 = \frac{\sqrt{27}}{\sqrt{16}} \).
14
15 I know that \( \frac{\sqrt{a}}{\sqrt{b}} = \sqrt{\frac{a}{b}} \), so this
        simplifies to \( \sqrt{\frac{27}{16}} \).
16
17 Now, \( \frac{27}{16} \) is already a fraction, and 27 is 9 times 3,
       which is 3 squared times 3, so \( \sqrt{27} = \sqrt{9 \cdot 3} = 3\
       sqrt{3} \).
18
19 And \( \sqrt{16} = 4 \).
20
21 So, originally, \( \frac{\sqrt{27}}{\sqrt{16}} = \frac{3\sqrt{3}}{4} \).
22
23 Is that the simplest form? Yeah, 3 and 4 have no common factors, and   3
        is already simplified.
24
25 The problem says "express your answer as a common fraction in simplest
       radical form." So, \( \frac{3\sqrt{3}}{4} \) is a common way to write
       it.
26
27 Is this a fraction? It's a fractional expression with a radical, which I
        think qualifies.
```

```
28
29  I could write it as \( \frac{3}{4} \sqrt{3} \), but that's the same thing
        .
30
31  Let me double-check my calculations.
32
33  a = 9, b = 2
34
35  ab + a = 9*2 + 9 = 18 + 9 = 27,   2 7 =    (9*3) = 3   3
36
37  ab - b = 18 - 2 = 16,   1 6 = 4
38
39  So, yes, \( \frac{3\sqrt{3}}{4} \)
40
41  I think that's the final answer.
42
43  The problem says "Your response should just be the answer containing only
        answer with no additional t e x t  f o r example, 0"
44
45  So, I should just write \( \frac{3\sqrt{3}}{4} \)
46
47  In the example, it's just "0", so probably as a mathematical expression.
48
49  Sometimes they write it as a fraction.
50
51  I think it's fine.
52
53  One more thing: is there any simplification I missed? Is the expression
        inside the square roots positive? For a=9, b=2, ab + a = 27 > 0, ab -
        b = 16 > 0, so yes.
54
55  Also, b is positive integer, as stated.
56
57  Alright, I think that's it.
```

Listing 5: Example reasoning generated by Grok 3 Mini Beta on MATH-500.

```
1  Okay, let's try to solve this problem step by step. The operation is
       defined as a & b = sqrt(ab + a) / sqrt(ab - b). We need to find the
       value of 9 & 2. Alright, so first, I should substitute a = 9 and b =
       2 into the formula. Let me write that down.
2
3  So, substituting, we get:
4
5  9 & 2 = sqrt(9*2 + 9) / sqrt(9*2 - 2)
6
7  Let me compute the numerator and denominator separately to avoid
       confusion.
8
9  Starting with the numerator: 9*2 + 9. Let's calculate that. 9*2 is 18,
       and then adding 9 gives 18 + 9 = 27. So the numerator is sqrt(27).
10
11 Now the denominator: 9*2 - 2. That's 18 - 2 = 16. So the denominator is
       sqrt(16).
12
13 So now the expression becomes sqrt(27) / sqrt(16). Let me simplify each
       square root.
14
15 sqrt(27) can be simplified. Since 27 = 9*3, sqrt(9*3) = sqrt(9)*sqrt(3) =
       3*sqrt(3).
16
17 sqrt(16) is straightforward. sqrt(16) = 4.
18
19 So substituting back, we have 3*sqrt(3) / 4. That simplifies to (3/4)*
       sqrt(3). But let me check if this is the simplest radical form.
```

20

21 The problem asks for a common fraction in simplest radical form. A common
    fraction means that the numerator and denominator are integers, and
    the radical should be simplified as much as possible. Here, sqrt(3)
    can't be simplified further, and the fraction 3/4 is already in
    simplest terms. So I think that's the answer.

22

23 Wait, let me double-check my steps to make sure I didn't make any
    mistakes.

24

25 First, substituting a=9 and b=2:

26

27 Numerator: ab + a = 9*2 + 9 = 18 + 9 = 27. Correct.

28

29 Denominator: ab − b = 9*2 − 2 = 18 − 2 = 16. Correct.

30

31 sqrt(27) = 3*sqrt(3), sqrt(16) = 4. So 3*sqrt(3)/4. Yes, that seems right
    .

32

33 Hmm. Let me check if there's any other way to simplify or if I messed up
    the order of operations. For example, in the denominator, ab − b:
    since it's 9*2 first, then subtract 2, which is 18 − 2 = 16. Correct.
     If it had been 9*(2 − 2), that would be 0, but that's not the case
    here. Parentheses are around ab in both numerator and denominator, so
     the operations are multiplication first, then addition or
    subtraction. So, yes, that's correct.

34

35 Another thing to check: sqrt(ab + a) / sqrt(ab − b). So the entire ab
    term is under the square roots, not just parts of them. So yes, sqrt(
    ab +a) is sqrt((ab) + a) which is sqrt(27), and sqrt(ab − b) is sqrt
    (16). All correct.

36

37 Therefore, 3*sqrt(3)/4 is the simplified form, which is a common fraction
     with a radical in the numerator. Since 3 and 4 have no common
    factors besides 1, the fraction is reduced completely. The radical
    sqrt(3) cannot be simplified further. So this should be the answer.

38

39 Just to be thorough, let me compute the decimal values to see if it makes
     sense. sqrt(27) is about 5.196, and sqrt(16) is 4. So 5.196 / 4
    1.299. If I compute 3*sqrt(3)/4, sqrt(3) is approximately 1.732, so
    3*1.732    5.196, divided by 4 is the same 1.299. That seems
    reasonable. Let me check with the original formula: sqrt(ab +a) is
    sqrt(27)    5 .196, sqrt(ab −b) is sqrt(16)=4, so the ratio is indeed
    approximately 1.299. So 3  3 /4 is correct.

40

41 Alternatively, is there a different way to approach this problem? Maybe
    factoring out terms?

42

43 Let's see. Let's take the original expression:

44

45 sqrt(ab + a) / sqrt(ab − b)

46

47 Factor numerator and denominator:

48

49 Numerator: sqrt(a(b + 1))

50

51 Denominator: sqrt(b(a − 1))

52

53 So the expression becomes sqrt(a(b + 1)) / sqrt(b(a − 1))

54

55 Which is sqrt[ (a(b + 1)) / (b(a − 1)) ) ]

56

57 But maybe that's not helpful here. Let's plug in a=9 and b=2:

58

```
59 sqrt(9*(2 + 1)) / sqrt(2*(9 - 1)) = sqrt(9*3) / sqrt(2*8) = sqrt(27)/sqrt
       (16) = same as before. So same result. So that method also gives 3
        3 /4. So yes, same answer.
60
61 Therefore, I'm confident that the value of 9 & 2 is 3  3 /4. Since the
       problem asks for the answer as a common fraction in simplest radical
       form, and 3/4 is a common fraction, and   3  is simplified, this
       should be correct.
62
63 **Final Answer**
64 \boxed{\dfrac{3\sqrt{3}}{4}}
```

Listing 6: Example reasoning generated by DeepSeek-R1 on MATH-500.

```
1 Okay, let's see. I need to get 24 using the numbers 3, 6, 6, and 7. Each
       number exactly once. Hmm. Let's start by thinking about possible
       combinations.
2
3 First, maybe multiply some numbers. 7 and 3 could be 21. Then I have two
       6s left. 21 plus 6 is 27, minus 6 is 21. That's not enough.
       Alternatively, 7 minus 3 is 4. Then 6 times 6 is 36. 4 times 36 would
        be way too big. Hmm.
4
5 Wait, maybe division can help. Let's see. If I do 6 divided by something.
        Like, 6 divided by (7 - 3) is 6/4 = 1.5. Then multiply by the other
       6: 1.5 *6=9. Not enough. That's 9, but how to get to 24?
6
7 Another approach: combining 6 and 6 first. 6 +6=12. Then 7-3=4. 12*4=48.
       That's double. Not helpful. But maybe divide by something? Wait, 48
       divided by 2 is 24, but I don't have a 2. Hmm.
8
9 Wait, what if I do (7 - 6) first? That gives 1. Then 3 times 6 is 18. 18
       plus 1 is 19. Not helpful. Maybe other combinations.
10
11 What about 6 multiplied by (something). Let's see. Let's think of 24 as
       6*4. So if I can make 4 with 3, 6, and 7. 7 - 3 is 4. Then 6*4=24,
       but I still have another 6 left. Oh, right, need to use all numbers.
       So that approach leaves a 6 unused. Not allowed.
12
13 Alternatively, maybe (3 * 7) is 21. Then 21 plus 6 is 27, minus 6 is 21
       again. Not helpful.
14
15 Wait, maybe combining division and multiplication. Let's see. For example
       , (6 / (7 - 3)) * 6. Let's calculate that: 7-3 is 4. 6/4 is 1.5. 1.5
       *6=9. Still no. Not enough.
16
17 Another idea: (6 * 7) is 42. Then subtract 6*3. 42 - 18=24. Wait, that's
       possible. Let's check: 6*7=42, then 3*6=18. 42-18=24. But wait, how
       are the numbers used? I have 6,7,3,6. So each number is used once.
       Yes! Because the first 6 and 7, then 3 and the other 6. So that's 6*7
        - 3*6 = 42-18=24. But let's verify the order. To use each exactly
       once, yes. 6,7,3,6. All four numbers used. So the equation would be
       6*7 - 3*6 =24. Let me check the math again. 6  7 =42, 3  6 =18.
       42-18=24. Yes! That works. So that's the solution. So the expression
       is 6*7 - 3*6 =24. But let me make sure parentheses aren't needed here
       . Since multiplication is done before subtraction, so it's (6*7) -
       (3*6) =24. So maybe written as 6  7  3  6 =24. Alternatively, maybe
       with parentheses for clarity, but it's not necessary. So the answer
       is 6*7 -3*6=24. But let me check if that's correct. Yes, using all
       numbers once. Each 6,7,3,6. Yes, all used once. So that's the
       solution
```

Listing 7: Example reasoning generated by DeepSeek-R1 on Game of 24.

1  We are asked: "Given 4 numbers: 3, 6, 6, 7. use these four numbers
      exactly once with basic operations (+ - * /) to obtain 24. Provide
      one valid solution. Provide thinking process and then answer enclosed
      in <answer> tags. So answer must be according to instructions. It
      says: "Please provide your thinking process in" and then " tags after
      your final answer: <answer> ... </answer>" The expected answer is
      formatted as we want output: When forced to output the final answer
      using <answer> and </answer> tags.

2

3  I must confirm instructions: "Your final answer should be enclosed in <
      answer> and </answer> tags, containing only letter of the correct
      answer with no additional textfor example, <answer>2*9+18/3=24</
      answer>" So answer final formula. Thus I answer with some manipulated
      : "3 * 7 - 6/6??" but that yields 3*7=21, then plus 6/6=1 equals 22,
      not 24. Let me see variations. We can test. We have two 6's: one can
      be used to produce another effect. Also note that can use division to
      create fraction 1. Let's experiment:

4

5  Goal: obtain 24.

6

7  We note that we have 7,3,6,6.

8

9  One idea: we can try: 6 / 6 = 1, then 7 + 3 = 10? then multiply together:
      10*1 ? not equals 24. Alternatively possibly: (7-3) * 6? equals
      4*6=24 is perfect? We just used two numbers: Use (7-3) equals 4, then
      multiplied by 6 equals 24 but then we didn't use the second 6? Wait
      no: 7-3 uses two numbers, and then the remaining 6 multiplied, but
      then we would get: (7-3) *6 = 24 anyway, but then second 6 is not
      used. Because we want to use all four numbers exactly one time.
      Actually then must incorporate the second 6 in multiplication or
      addition but not null. Possibly: Multiply by 6 and then division by 6
      gives 24? Let's try: (7 - 3)/6? That equals 4/6=2/3 no.

10

11  The pair (7-3)=4, but then multiplied by 6 gives 24 if there was only one
      6. But we have two sixes. I can try: maybe use one of them to
      simplify a fraction? Possibly: 7-3 = 4. Then use one 6 to multiply to
      get 24? But then the remaining 6 equals 6, we can then do 6/? Wait,
      perhaps: (7 - 3) * 6 /6 equals derivative but then it equals 4 not
      equals 24. So that's not.

12

13  We want to incorporate both 6's, but wait: We have two 6 numbers; idea:
      use one operation with division by 6/6, which equals 1, so then
      multiplication by something equals? For instance: (7 - something).
      But then maybe: (6 * 7 - 6)/3 = 42 - 6 / 3 = 36/3=12, no.

14

15  Wait re-read instructions: "Given a game 24 problem, we have 4 numbers:
      3, 6, 6, 7. Use these four numbers exactly once etc." We need to use
      +, -, *, / in any order. One possible candidate: "((7*6*6)/3) - ?"
      But that squared: 7*6*6 = 252, divided by 3 gives 84, not 24.
      Possibly: (7* (6+6/3)? etc.

16

17  I want to try a combination: We want to obtain 24. Look at a potential
      candidate: 6/6 produce 1 then used with 7 and 3 to produce 24, maybe:
      7+3=10, 10+1=... no.

18

19  Alternatively: maybe generating fraction: (7-3/6)*6 ... Let's try:
      3/6=0.5, then 7-0.5=6.5, multiplied by 6 equals 39? Not 24. Another
      combination: 7*(6/3 - 6) negative fraction multiply? Let me attempt:
      We want to create 24. Since one of the numbers is 7, maybe 7 +
      something equals 24, so maybe 17? Something: 6/6 equals 1. So maybe:
      7 + (6+3) equals 16, not 24.

20  Alternatively: Use subtraction like: 7 * something etc.

21

22  Wait: maybe combine 7 and 3 to get 21, then add something: 21 + 6/6?
      equals 21+1=22. not 24. Alternatively: 7 *3 equals 21, plus second 6

divided by 6 equals 1, then equals 22. Alternatively: 7+3+6/6 equals:
10 + 1 = 11? No.

23

24 Maybe consider: (7 - 3/6) *6. That equals (7 -0.5)=6.5*6=39, not 24.

25

26 Perhaps we can use fraction with denominator and numerator not just
simple. Let's call the numbers a=3, b=6, c=6, d=7 in some arrangement
. I attempt to combine two numbers to make 2, and then multiply them.
For instance: one idea: get 7 - 6/6 = 7 -1 = 6, then 6+3 = 9, not
24. Alternatively: 7 * 6/ (6+? =? no.

27

28 Maybe incorporate division to get a fraction that can yield 1/2 etc.
Possibly: (6/ (7 - 3))=6/4=1.5. Then perhaps multiply by other 6
equals 9. Then not 24.

29

30 Alternatively: (6*6 - 7)/3. Let me calculate:6*6=36, minus 7=29, divided
by 3 equals approx 9.67, not 24.

31

32 Maybe: (7*6 - 3)/? Actually, maybe: (7*6 - 3) equals 42-3=39, and if
divided by (6/?) but no.

33

34 Or maybe: (7*3 + 6/6) equals 21+1=22. Not 24.

35

36 What about: (7+3/6)*6, equals (7+0.5)*6=7.5*45, not 24.

37

38 Alternatively: 6*(7-3/6)? again: 7-0.5=6.5, multiplied yields 39 not 24.

39

40 Maybe incorporate parentheses: Since multiplication and division have
same precedence, maybe check: 7*(6-3)*(6/?) But wait, we have two
identical 6's; maybe distinguish them as 6a and 6b.

41

42 Maybe attempt: 6*(7-6/6) - 3. Evaluate inside parentheses: 6/6=1, then 7
-1=6, times 6 equals 36, minus 3 equals 33, not 24.

43

44 What about: 6*(7-6/6) - (something)? Not sure.

45

46 Alternatively attempt: (7-6/6) * (6-3)? That equals (7-1)*3=6*3=18, not
24.

47

48 Alternatively: 6*(7-3/6)? Already did that: 6*6.5=39, no.

49

50 What about: (7*6 - 6)/3? That equals (42-6)/3=36/3=12. So two 12's give
24 maybe: you can sum them. But need use both 6 maybe: Actually, (7*6
- 6)/3 is using both 6's? Let's parse: numbers used: 7, a 6, and
then - 6, and then divided by? Actually, we then
use 3, two 6 and one 7. But that's only three numbers? Wait: We have
lots: Actually, we have 3,6,6,7, so we need to use one 3, both 6's,
and the 7. In that expression " (7*6 - 6)/3", which numbers are used?
7, first 6, second 6, and then 3. Yes, that uses all four numbers:
They appear as 7, 6, 6, and 3. And it equals (7*6=42-6=36)/3 = 12.
That equals 12. Then if we use multiplication by 2 (but then where do
we get 2?) Because we have already used all numbers; maybe we can do
: (7*6 - 6)/3 = 12, then add something? No, that's too many numbers.

51

52 Maybe: If we do 2*12 that yield 24, but can we get the extra factor 2
from somewhere without using extra numbers? But we are not allowed
using additional number 2. Need to combine the four numbers to yield
24. We can try: ((7*6 - 6)/3)*? Wait, no.

53

54 Maybe: 6*(7-3/6) -? Already tried.

55

56 Let's solve algebraically. We need to get result 24 using arrangement of
the four numbers: a, b, c, d are 3, 6, 6, 7. I want to incorporate a
subtraction maybe: maybe a formula: (7+3) *6/6 = (10)* (6/6=1) equals
10, no not 24.

```
57
58 Think in terms of 24 factorization: 24 = 8*3, 24 = 4*6, 24= (7+something)
       etc.
59
60 What about (7+3/6)*6? That equals (7+0.5)*6=7.5*45, works not.
61
62 How about (7*3-6)/6? That equals (21-6)/6=15/6=2.5, not 24.
63
64 Maybe try: (6*(7-3/6)) -? That equals 6*(7-0.5)=6*6.5=39, too high.
65
66 Maybe: (7+6/6)*3 = (7+1)*3=8*3=24. That's promising. Let's check: There
       is 7, then use 6/6 which equals 1, then 7+1=8. Then multiply by 3
       gives 24. But then what happened to the second 6? Wait, used: 7, one
       6, another 6, and 3 appear: Actually, this expression " (7+6/6)*3 "
       uses 7, uses one 6 in the fraction 6/6 (using the two 6's, because
       6/6 uses one 6 numerator and one 6 denominator), then multiply by 3
       uses the number 3. Yes, we used 7, plus fraction 6/6, plus 3. And 6/6
        uses both 6 numbers. But note: This expression uses exactly the four
        numbers: Actually, check: Numbers: 7, 6, 6, 3 appear in the
       expression: 7 is there, 6 is in numerator, 6 is in denominator, and 3
        is outside. So yes, it uses them. And the result equals (7 + (6/6))
       * 3 = (7+1)*3=8*3=24. That works.
67
68 Therefore, one feasible solution is: (7 + (6/6))*3=24. But note:
       Sometimes the order of operations: Actually, re-check: (7 + (6/6))=
       (7 + 1)=8, then multiplied by 3 = 24. Yes.
69
70 This is a valid answer. Another possibility: (3*(7+ (6/6)))=24. So one
       can even write (3*(7+6/6)).
71
72 Let's be explicit: Use one of two 6's as 6 in numerator and the other as
       6 in denominator. Guarantee that parentheses wrap that fraction. So I
       'd write: 3*(7+6/6)=? But then equals 3*(7+1) equals 3*8=24.
73
74 Yes! That exactly uses each number once: digits: 7, a 6, another 6, and a
        3. So using result = 24.
75
76 Let's denote our answer: We want to provide final answer inside <answer>
       tags. But our instructions "Your final answer should be enclosed in <
       answer> and </answer> tags, containing only letter of the correct
       answer with no additional text" implies that the answer should be
       free of explanation in the final answer block. But the final answer
       is such: <answer>(7+6/6)*3=24</answer>, or <answer>3*(7+6/6)=24</
       answer> can be both.
77
78 Wait, but instruction says: "containing only letter of the correct answer
        with no additional textfor example, <answer>2*9+18/3=24</answer>" It
       's a sample text. So maybe I'll provide: <answer>(7+6/6)*3=24</answer
       >. But wait, the instruction says: "Your final answer should be
       enclosed in <answer> and </answer> tags, and containing only letter
       of the correct answer with no additional text" This may mean that
       output should be solely that expression. Perhaps the answer
       requirements are "ex: <answer>2*9+18/3=24</answer>" but then they
       said "example", so it's example meaning exactly one valid answer is
       needed.
79
80 Yes, my final answer would be: (7+6/6)*3=24. I want to check if any
       potential alternative check: For instance, 6 in numerator, 6 in
       denominator give 1. So 7+1=8, then multiplied by 3 gives 24. So
       indeed valid. Also note that expression: 3*(7+6/6)=24 equals 3*(7+1)
       =24 correct.
81
82 I'll now provide step-by-step thinking process followed by the final
       result in the required format.
83
```

84 I'll produce final answer: <answer>3*(7+6/6)=24</answer>. Also note
     instructions says include my thinking process, I must provide that
     along with my final answer.

85

86 Wait instructions say: " provide your thinking process in tags? The
     instructions: "Now given a game 24 problem ... Provide your thinking
     process in" wait instructions not truncated? "Your goal is to use all
      the 4 numbers and basic arithmetic to obtain 24. Please provide one
     feasible solution. Please provide your thinking process in". The text
      is truncated: "tags." Possibly the instruction might be: "Please
     provide your thinking process in our answer." But some systems
     require explanation inside solution. But instructions state "your
     final answer should be enclosed in <answer> and </answer> tags
     containing only letter of the correct answer." That means that I have
      to separate the explanation (external explanation text) and final
     answer block, the final line's block must only be expression. But
     instructions tell if code text inside a format.

87

88 The instructions: "Your final answer should be enclosed in <answer> and
     </answer> tags, containing only letter of the correct answer with no
     additional textfor example, <answer>2*9+18/3=24</answer>" So I must
     not include extra text in the answer block, so I include only the
     expression in <answer> and then my explanation separated from that
     block.

89

90 I'll provide a final answer and then explanation above it. But
     instructions "Please provide your thinking process in tags" might
     mean that the final answer's thinking process should be included as
     well, but then the final answer although hmm! The sample said "with
     your final answer" but did it ask "I need to provide the whole
     process"? So apparently I must provide my chain-of-thought reasoning
     outside the answer block maybe. But instructions say " your final
     answer should be enclosed in <answer> and </answer> tags, containing
     only letter of the correct answer, no additional text" that means
     what? Possibly "only share the exact answer and not additional
     commentary?"

91

92 Wait, it says: "with no additional text" likely means that inside the
     answer block I will not have text, only letters. In our final answer
     block, we must put the mathematical expression. But then the chain-of
     -thought must be provided before final answer block, but the final
     answer block must be only the correct answer itself.

93

94 Wait, instructions: "containing only letter of the correct answer with no
      additional text." That means the final answer block should only be
     the answer. But then we want to include my chain-of-thought. But
     instructions says "Please provide your thinking process in [maybe
     code tags]" after answer. Actually instructions: "Please provide your
      thinking process in tags" might be some instruction. Let me re-read
     the prompt: It says: "Now given a game 24 problem, we have 4 numbers:
      3, 6, 6, and 7. ..." then "please provide one feasible solution to
     this problem. Please provide your thinking process in [what?" The
     source says "in tags", but possibly it was truncated. Possibly
     instructions are: "Write your final answer with a chain-of-thought
     explanation then a final answer between <answer> and </answer> tags."

95

96 I must check the original problem text: "Now given a game 24 problem, we
     have 4 numbers: 3, 6, 6, and 7. Your goal is to use all the 4 numbers
      and basic arithmetic operations (+ - * /) to obtain 24. You must use
      each number exactly once, and you can use parentheses to change the
     order of operations. Please provide one feasible solution to this
     problem. Please provide your thinking process in tags. After that,
     your final answer should be enclosed in <answer> and </answer> tags,
     containing only letter of the correct answer with no additional
     textfor example, <answer>2*9+18/3=24</answer>"

97
98 I conclude that I must print my chain-of-thought process in the answer
     possibly using some plain text style, then final answer in tags as
     requested.

99
100 However, instructions in this conversation say: "your final answer should
     be enclosed in...". So must output chain-of-thought first (maybe
     under separate explanation, but careful instructions of the assistant
     policy say that I must not reveal chain-of-thought. This is slight
     contradiction: rules says my chain-of-thought should not be revealed
     in the final answer.

101
102 The instructions in the final answer instruct me: " provide your thinking
     process in tags" which is likely the chain-of-thought. But
     instructions from other problems: "Don't output chain-of-thought
     explicitly" in our conversation. But let me double-check: The prompt
     instructs "please provide your thinking process" and then final
     answer in tags. However instructions from "Chain-of-thought policy"
     in the conversation instructions if the chain-of-thought should be
     hidden? The instructions says: "Don't include any part of your
     internal chain-of-thought in final answer." But then instructions
     mention "Your final answer should be enclosed in ... tags but no
     explanation." However the user prompt doesn't explicitly mention from
      policy "Don't use chain-of-thought" but instructions mention user "
     explain your thinking process". But actually instructions explicitly
     say "explain your thinking process", so do I provide chain-of-thought
     ? We must not provide chain-of-thought in final result. But
     instructions "Include your chain-of-thought in your answer" may not
     be allowed because instructions "The assistant must not output chain-
     of-thought".

103
104 Wait, instructions: "explanations: The chain-of-thought is hidden and not
      output." The conversation instructs chain-of-thought cannot be
     included with final answer due to rules if not requested. Wait – let
     me check our policies: "Don't output any additional chain-of-thought
     ." But instructions says "include your thinking process in your final
      answer in tags" but the meta-intro said "Don't reveal chain-of-
     thought."

105
106 Let's re-read summary output from previous problems: Some chain-of-
     thought reasoning internal chain-of-thought that should not be
     included in the final answer chain-of-thought. However, instructions
     said: "explain my thinking process" may be sabotage. Wait, careful:
     instructions from the user ask: "Please provide your thinking process
      in tags." The user instructs: "Please provide your thinking process
     in tags" does that permit chain-of-thought? Actually, instructions
     from guidelines: "Don't include chain-of-thought." Possibly from
     internal policies: chain-of-thought internal explanation is not to be
      output. So I must provide some meta summary reasoning if it's
     requested? Or does the answer instructions ("explain your thinking
     process") mean that I output a separate explanation outside the
     answer block?

107
108 Wait, the instructions say: " After that, your final answer should be
     enclosed in ... tags, containing only letter of the correct answer
     with no additional text" That means that my final answer block should
      only consist of the final expression, no explanation. But
     instructions "explain your thinking process" must be provided perhaps
      separate text outside the answer block. But instructions say: "your
     final answer should be enclosed in <answer> and </answer> tags,
     containing only letter of the correct answer with no additional text"
      for example that sample: <answer>2*9+18/3=24</answer>. That means
     the final answer block should only be that. But then I must also
     include my chain-of-thought? But guidelines say we should hide chain-
     of-thought if possible.

109
110 According to our Microsoft policies instructions modifications: "Don't output internal chain-of-thought reasoning." I'm not allowed to show it. But then instructions from user instruct me: "please provide your thinking process in tags" might be interpreted as "please provide your chain-of-thought that leads to your answer."
111
112 However, the instructions may refer to "explain your thinking process briefly" but as multiple turns sometimes get this. But instructions " with no additional text" refer to "final answer" block. So I must not include a separate chain-of-thought inside that block.
113
114 But instructions which are part of the sample are "for example, <answer >2*9+18/3=24</answer>" indicates the answer is provided in a formatted enclosed tag.
115
116 I must now produce the final answer explanation first if required? But instructions "explain your thinking process in ..." might be meaning my chain-of-thought internal analysis that I'll include outside the final answer tags. But then are final answer result should be minimal .
117
118 However, the instructions say: "Your final answer should be enclosed in < answer> and </answer> tags, containing only letter of the correct answer with no additional text" so the final answer block must contain only that chain-of-thought's result. But if I add any additional text around the answer block that might be considered additional text. The instructions specify: "Your final answer should be enclosed in <answer> and </answer> tags, containing only letter of the correct answer with no additional text" For example, " <answer >2*9+18/3=24</answer>" So they indicated that the answer block must only contain the textual characters of the expression.
119 Thus I'll produce, outside of that block, I might provide chain-of-thought explanation. But maybe that counts as additional text not allowed. The instructions "containing only letter of the correct answer with no additional text" is the final answer block.
120
121 I need to check whether the chain-of-thought explanation I produce near the answer block is considered "additional text" that need to be omitted from the final answer block. The sample instructions "for example, <answer>2*9+18/3=24</answer>" then my final answer, which I' ll replicate.
122
123 So I'll produce: "Chain-of-thought: ..." etc. But instructions "explain your thinking process in tags" might refer to my internal chain-of-thought but I'm not to output that.
124
125 Wait, sorry, I try to re-read instruction:
126
127 "Now given a game 24 problem, we have 4 numbers: 3, 6, 6, and 7.
128  ...
129  Please provide one feasible solution to this problem.
130  Please provide your thinking process in (tags). After that, your final answer should be enclosed in <answer> and </answer> tags, ..."
131
132 Maybe the parentheses "in tags" might be literal formatting instruction like: "in words" but I'm not sure.
133
134 I think I'll produce initial explanation text in plain text, then my final answer block enclosed in <answer>: which I wrote as <answer >3*(7+6/6)=24</answer>.
135
136 Wait, but instructions "with no additional text" might mean I must not include any explanation text within the "<answer>" block. So the answer block should exactly contain an expression. So I'll produce

```
2106         answer block as: "<answer>3*(7+6/6)=24</answer>" with no extra words.
2107          But I must produce text output of my chain-of-thought? Possibly I
2108          can include it before the answer block.
2109 137
2110 138 Given the request includes "explain your thought process", I'll produce a
2111          separate reasoning explanation as separate text at the beginning,
2112          then after that a line that says "Answer:". But instructions "with no
2113          additional text" inside answer block implies that answer block must
2114          not contain any text other than the expression.
2114 139
2115 140 I recall guidelines: "don't reveal chain-of-thought." Possibly I might
2116          provide a final chain-of-thought summary in a separate text. But
2117          instructions "explain your chain-of-thought" if they are reading the
2118          final answer, I'll produce a message like: "Chain-of-thought: ...."
2119          then "" then produce the answer in bold. But in final answer block,
2120          only the textual expression "3*(7+6/6)=24".
2120 141
2121 142 I wonder if I'll produce "3*(7+6/6)=24" or " (7+6/6)*3=24"? They are
2122          equivalent mathematically. I might prefer "3*(7+6/6)=24" because it's
2123          more symmetric. But I must check: It uses the multiplication plus
2124          parentheses: 3*(7+6/6). That's valid.
2124 143
2125 144 I'll produce that final answer.
2126 145 I'll produce text "Final Answer:" and then show the answer in <answer>
2127          ... </answer> tags, but instructions "with no additional text" means
2128          that inside solution block I must produce text "3*(7+6/6)=24" exactly
2129          . But then instructions say "explain your chain-of-thought in tags"?
2130          Possibly I'll produce my chain-of-thought in summarily but not in the
2131           final answer block.
2131 146
2132 147 To be safe, I'll produce a final explanation with text outside of the
2133          final answer block, then separate final answer block.
2133 148
2134 149 But instructions say "your final answer should be enclosed in <answer>
2135          and </answer> tags, with no additional text in the final answer block
2136          ." However, instructions instruct "explain your thinking process in
2137          tags" might refer to additional paragraphs preceding the final answer
2138          .
2138 150
2139 151 I recall instructions "explain your chain-of-thought" might violate my
2140          instructions. But in any case I'll produce explanation and then final
2141           answer.
2141 152
2142 153 Thus my answer text: "One solution is: 3*(7+6/6)=24. Explanation: Because
2143          6/6=1, so 7+1=8, then 8*3=24, and we have used all numbers exactly
2144          once."
2145 154
2146 155 I must produce final answer in simple tags: <answer>3*(7+6/6)=24</answer
2147          >.
2147 156
2148 157 I'll produce that now.
```

Listing 8: Example reasoning generated by Phi-4-reasoning-plus on Game of 24.

```
2153 1     In this 24 game problem, I need to use the numbers 3, 6, 6, and 7
2154         exactly once each with basic arithmetic operations (+, -, *, /) to
2155         get a result of 24.\n\n
2155 2     One solution is: (7 - 6) * 6 * 3 + 6 = 24\n\nWait, let me verify:\n(7
2156         - 6) = 1\n1 * 6 = 6\n6 * 3 = 18\n18 + 6 = 24\n\nSo we have: (7 - 6)
2157         * 6 * 3 + 6 = 24
```

Listing 9: Example reasoning generated by Claude 3.7 Sonnet on Game of 24.

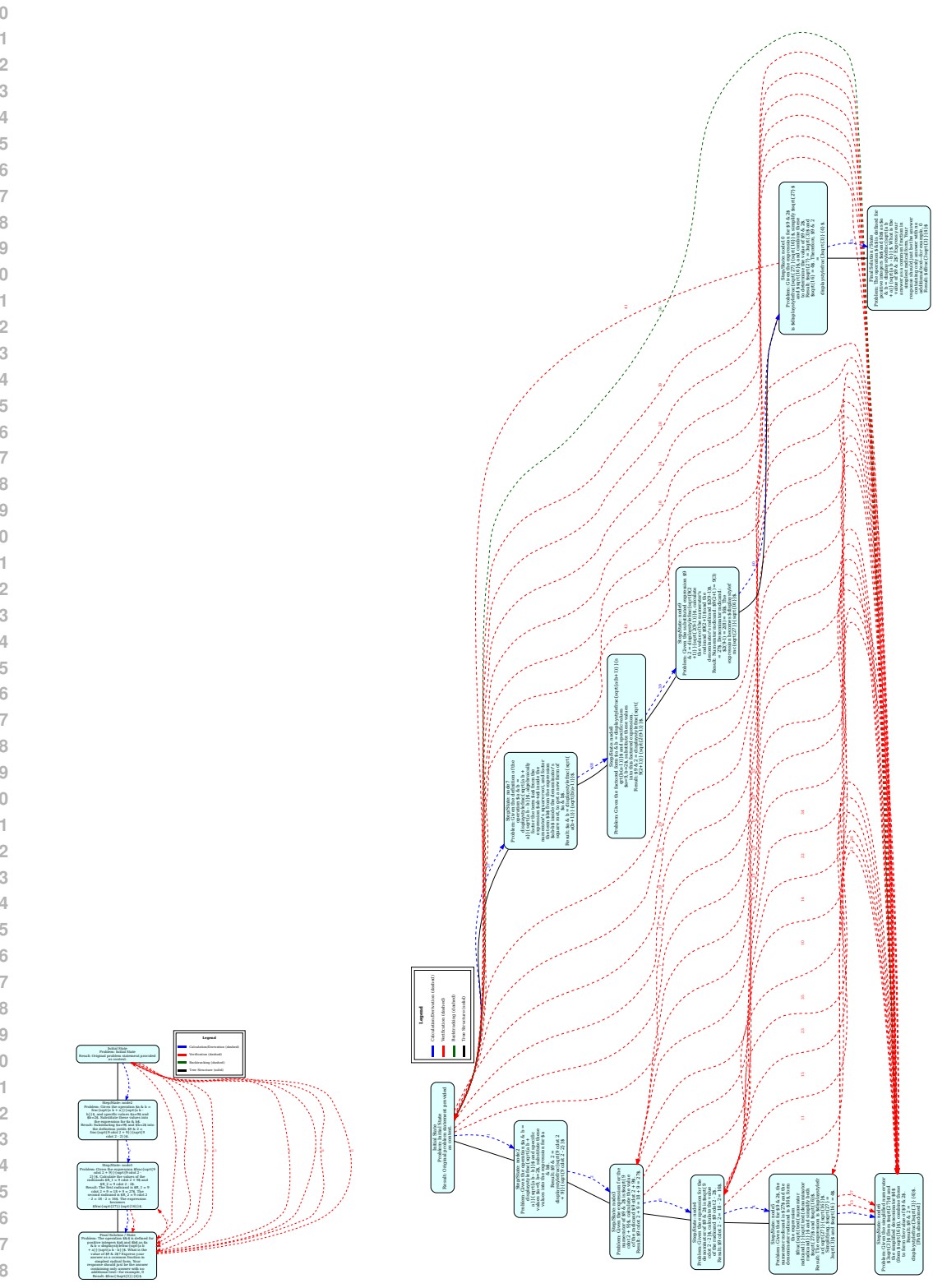

(a) Grok 3 Mini Beta. Corresponding reasoning traces are shown in Listing 5.

(b) DeepSeek-R1. Corresponding reasoning traces are shown in Listing 6.

Figure 8: ReJump representations of reasoning traces generated by Grok 3 Mini Beta and DeepSeek-R1 for a MATH-500 problem.

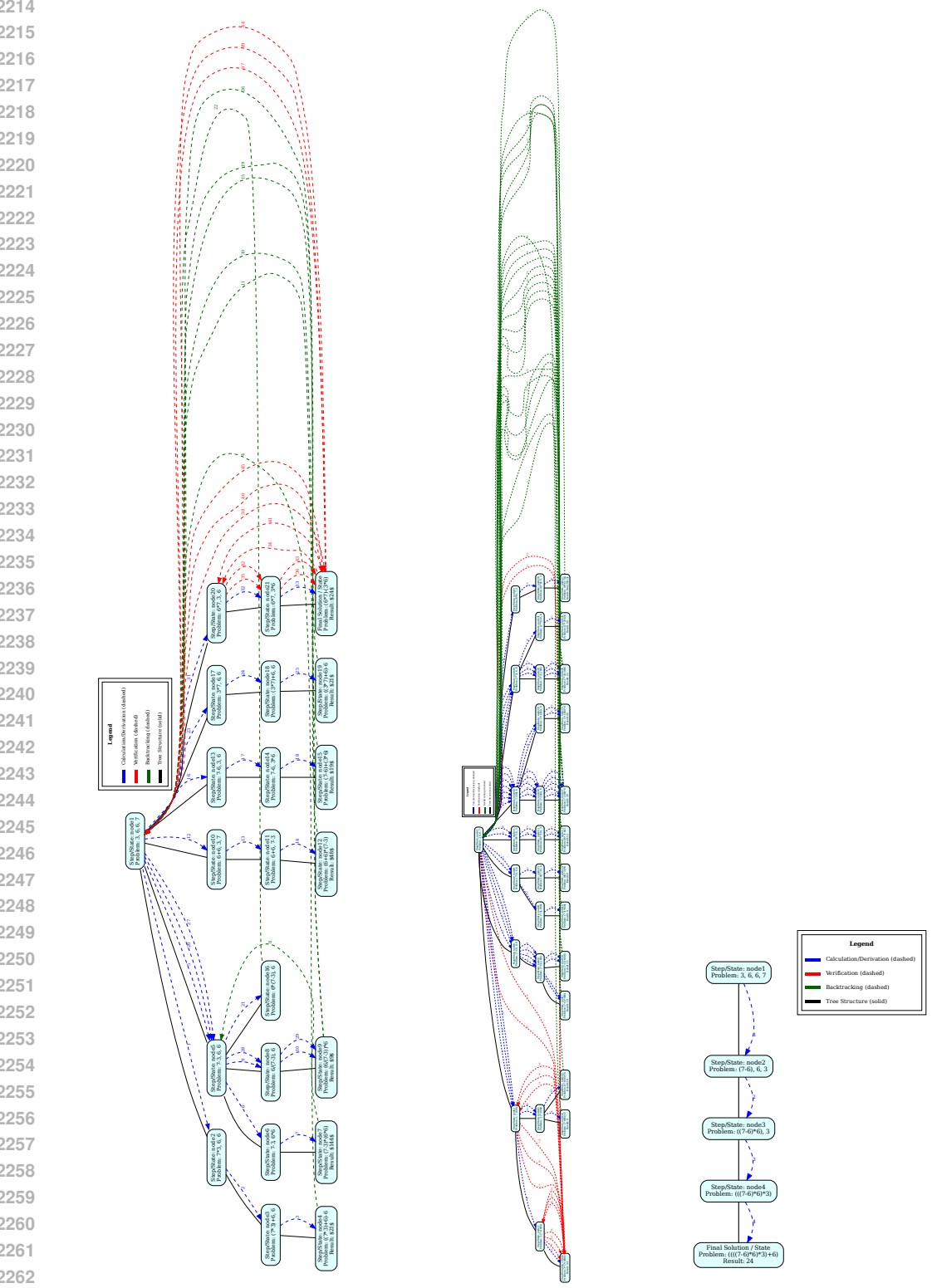

(a) DeepSeek-R1. Corresponding reasoning traces are shown in Listing 7.

(b) Phi-4-reasoning-plus. Corresponding reasoning traces are shown in Listing 8.

(c) Claude 3.7 Sonnet. Corresponding reasoning traces are shown in Listing 9.

Figure 9: ReJump representations of reasoning traces generated by DeepSeek-R1, Phi-4-reasoning-plus, and Claude 3.7 Sonnet for a Game of 24 problem.

Table 11: Alignment between the ReJump representations extracted by ReJump-Extractor and the ground-truth ReJump on the Game of 24, comparing Gemini 2.5 Pro and Claude 3.7 Sonnet.

| Evaluation LLM | $\mathrm{Sim}_T$ | $\mathrm{Sim}_J$ |
|---|---|---|
| Gemini 2.5 Pro | .943 | .940 |
| Claude 3.7 Sonnet | .867 | .672 |

Figure 10: **Reasoning performance of DeepSeek-R1, Grok 3 Mini Beta, and QwQ-32B on MATH-500 and Game of 24 with temperature set to 0.** The bar plots present the final accuracy (pass@1), while the radar plots detail six reasoning metrics: solution count ($\#_{\mathrm{solution}}$), jump distance ($d_{\mathrm{jump}}$), success rate ($r_{\mathrm{success}}$), verification rate ($r_{\mathrm{verify}}$), overthinking rate ($r_{\mathrm{overthinking}}$), and forgetting rate ($r_{\mathrm{forget}}$). For comparability, $\#_{\mathrm{solution}}$ and $d_{\mathrm{jump}}$ are normalized across all models and datasets, denoted as $\overline{\#_{\mathrm{solution}}}$ and $\overline{d_{\mathrm{jump}}}$. To ensure that higher values consistently reflect preferred behavior, we report the complements $1 - r_{\mathrm{overthinking}}$ and $1 - \mathbf{1}_{\mathrm{forget}}$. The results support the same findings as in Fig. 3, which shows performance for DeepSeek-R1, Grok 3 Mini Beta, QwQ-32B, Claude 3.7 Sonnet, and Phi-4-reasoning-plus at temperature 1.

### C.3 COMPARISON OF GEMINI 2.5 PRO WITH ALTERNATIVE LLM

Among various state-of-the-art closed-source models, we select Gemini 2.5 Pro for its low cost and strong performance. Alternatives like o1 and Claude 3.7 Sonnet (or Claude Sonnet 4) are more expensive. Claude 3.7 Sonnet costs twice as much as Gemini 2.5 Pro, while o1 is five times Claude's price. Due to o1's prohibitive cost, we designed experiments comparing Gemini 2.5 Pro against Claude 3.7 Sonnet (with thinking mode enabled) on tree and jump extraction tasks. Tab. 11 reports tree and jump similarities (as defined in our paper) for extractions by Claude 3.7 Sonnet. Claude 3.7 Sonnet performs comparably worse than Gemini 2.5 Pro.

## D EXTENDED SEC. 5: REJUMP-BASED BEHAVIORAL COMPARISONS

### D.1 EXTENSION OF SEC. 5.1: COMPARING REASONING STRUCTURE ACROSS STATE-OF-THE-ART LRMS AND TASKS

In Sec. 5.1, we compare the performance of five state-of-the-art LRMs at temperature 1, as both Claude 3.7 Sonnet and Phi-4-reasoning-plus use this setting by default: Claude 3.7 Sonnet does not support temperature control in thinking mode, and Phi-4-reasoning-plus performs poorly with low temperatures. Here, we additionally report the performance of DeepSeek-R1, Grok 3 Mini Beta, and QwQ-32B at temperature 0, as well as tree and jump similarity results for both temperature settings. One caveat is that the Anthropic API requires specifying a token limit in advance. In our main experiments, we set this limit to 1,048 tokens. However, we observed that increasing the limit to 10,000 tokens can greatly improve performance. For example, achieving pass@1 = 1 on Game of 24. Due to the significantly higher cost of using the Anthropic API ($7\times$ that of DeepSeeks and even more compared to others), we report results using the 1,048-token setting for Claude models in the main paper.

The results in Fig. 10 show the pass@1 accuracy and six reasoning evaluation metrics for temperature 0. We observe that the performance of DeepSeek-R1, Grok 3 Mini Beta, and QwQ-32B remains consistent with their temperature-1 counterparts, further supporting the findings in Sec. D.5

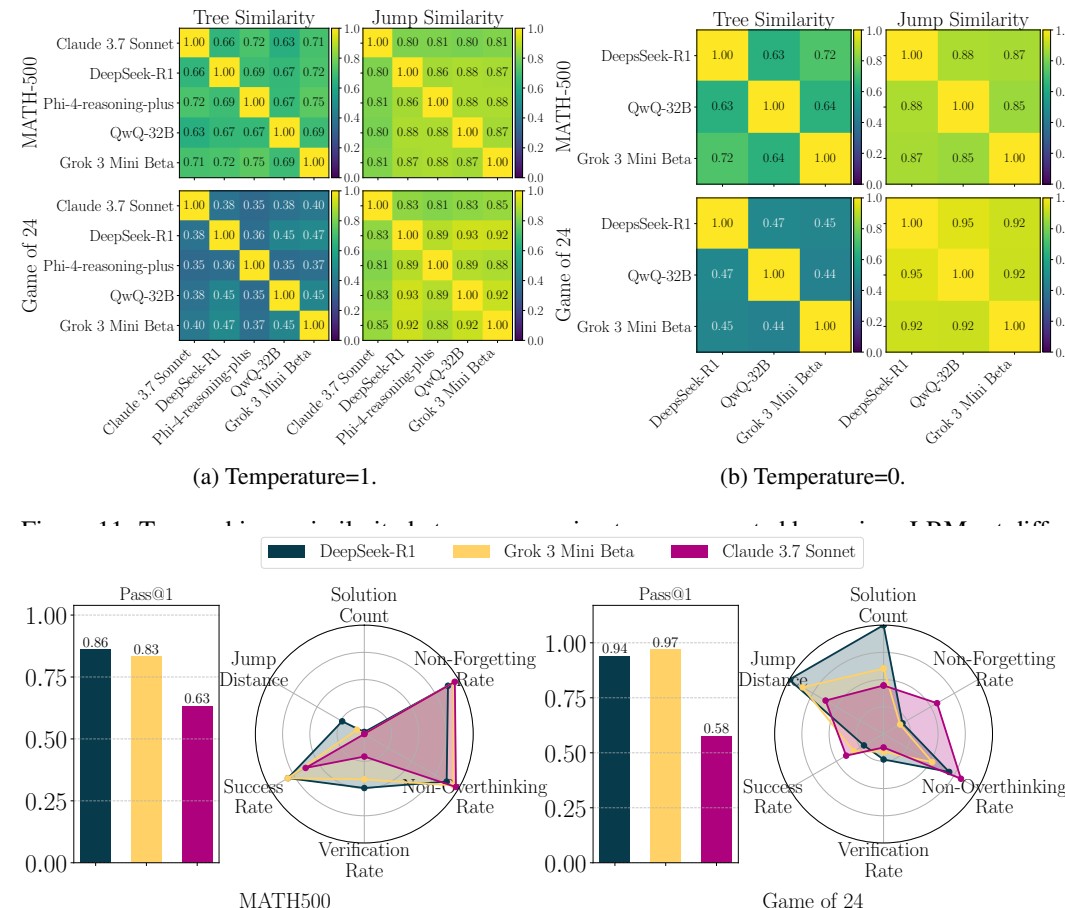

(a) Temperature=1.            (b) Temperature=0.

Figure 11: Tree and jump similarities between reasoning trees generated by various LRMs at different

Figure 12: **Reasoning performance of DeepSeek-R1, Grok 3 Mini Beta, and Claude 3.7 Sonnet on MATH-500 and Game of 24.** The bar plots present the final accuracy (pass@1), while the radar plots detail six reasoning metrics. For comparability, solution count and jump distance are normalized across all models and datasets. To ensure that higher values consistently reflect preferred behavior, we report the non-forgetting rate and non-overthinking rate rather than forgetting rate and overthinking rate. The results show that models display distinct reasoning behaviors across datasets. Furthermore, even when models achieve similar final performance, their underlying reasoning processes can differ significantly.

that temperature has limited impact on reasoning behavior. We also compare reasoning structures across models using tree and jump similarity metrics, as shown in Fig. 11. On MATH-500, tree similarities are notably higher than those on Game of 24, likely because MATH-500 encourages more exploitation and yields less diverse tree structures. On MATH-500, Grok 3 Mini Beta and Phi-4-reasoning-plus exhibit the highest tree and jump similarities, while QwQ-32B and Claude 3.7 Sonnet score the lowest in both. For Game of 24, DeepSeek-R1 and Grok 3 Mini Beta show the highest tree similarity, while QwQ-32B and DeepSeek-R1 achieve the highest jump similarity.

Furthermore, to better visualize the metric values for the top models DeepSeek-R1, Grok 3 Mini Beta, and Claude 3.7 Sonnet, we provide a version of Fig. 3 that includes only these three models in Fig. 12.

## D.2 EXTENSION OF SEC. 5.2: COMPARING REASONING STRUCTURE: STANDARD LLMS VS. LRMS

In Sec. 5.2, we compare base LLMs (DeepSeek-V3, Qwen-2.5-32B) with their corresponding LRMs (DeepSeek-R1, QwQ-32B) on pass@1 accuracy and reasoning evaluation metrics for Game of 24. Here, we present the results for MATH-500 in Fig. 13, which further support the findings from

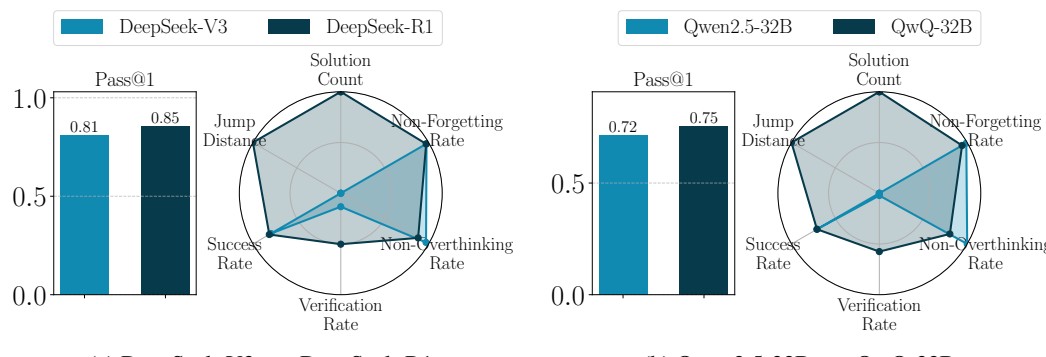

(a) DeepSeek-V3 v.s. DeepSeek-R1       (b) Qwen2.5-32B v.s. QwQ-32B

Figure 13: **Comparison of base LLMs (DeepSeek-V3, Qwen-2.5-32B) and their corresponding LRMs (DeepSeek-R1, QwQ-32B) on pass@1 and reasoning metrics for the MATH500.** The bar plots present the final accuracy (pass@1), while the radar plots detail six reasoning metrics: solution count ($\#_{\text{solution}}$), jump distance ($d_{\text{jump}}$), success rate ($r_{\text{success}}$), verification rate ($r_{\text{verify}}$), overthinking rate ($r_{\text{overthinking}}$), and forgetting rate ($r_{\text{forget}}$). For comparability, $\#_{\text{solution}}$ and $d_{\text{jump}}$ are normalized across all models and datasets, denoted as $\overline{\#_{\text{solution}}}$ and $\overline{d_{\text{jump}}}$. To ensure that higher values consistently reflect preferred behavior, we report the complements $1 - r_{\text{overthinking}}$ and $1 - \mathbf{1}_{\text{forget}}$. Despite similar $r_{\text{success}}$, LRMs achieve higher pass@1 by generating more and diverse solutions, as reflected in higher average solution counts and jump distances. LRMs also exhibit increased verification, overthinking, and forgetting behaviors.

Sec. 5.2: LRMs achieve better performance not by higher success rates, but through increased exploration, verification, and other reasoning behaviors.

### D.3 EXTENSION OF SEC. 5.3: IMPACT OF DISTILLATION ON REASONING STRUCTURE

In Sec. 5.3, we compare the similarity of the distilled model to both its base and teacher models within the 14B group. The full similarity results for both the 14B and 32B groups are presented in Tab. 12, and detailed reasoning evaluation metrics for each model on the two datasets are shown in Fig. 14.

Fig. 14 reveals that distilled models exhibit more deliberate reasoning behaviors, such as exploration, verification, overthinking, and forgetting, compared to their base models. However, this does not translate into a higher success rate; in fact, the success rate often decreases. As a result, the distilled model may underperform the base model on MATH-500 (which emphasizes correctness), while outperforming it on Game of 24 (which benefits more from exploratory behavior). These findings corroborate those in Sec. 5.1, which show that MATH-500 favors success rate, whereas Game of 24 rewards exploration. They also reinforce the conclusion in Sec. 5.3 that distilled models inherit reasoning behaviors from their teachers. Additionally, we highlight a new insight:

> **Finding:** *Distillation can reduce the success rate of the base model.*

Lastly, we conduct a preliminary comparison between Supervised Fine-Tuning (SFT) and Reinforcement Learning (RL) by evaluating DeepSeek-R1-Distill-Qwen-32B and QwQ-32B. This comparison is not strictly controlled, as the training datasets and durations may differ significantly. Nonetheless, the results visualized in Fig. 15 offers an initial perspective: the RL-trained model appears to exhibit more deliberate reasoning behaviors, including increased exploration, verification, and overthinking.

### D.4 EXTENSION SEC. 5.4: IMPACT OF REASONING EXAMPLES ON REASONING STRUCTURE

While Sec. 5.4 shows that reasoning examples tend to have a stronger effect on jump similarity, while having little impact on pass@1 and tree similarity on MATH-500, we present the full results for both MATH-500 and Game of 24 in Fig. 16. The results further support this observation.

Table 12: **Tree similarity** ($\text{Sim}_T$) **and jump similarity** ($\text{Sim}_J$) **between each distilled model and its corresponding base and teacher models.** Across both datasets and model scales, distilled models are more similar to the teacher LRMs than to the base models in most cases.

| Comparison Group | vs. DeepSeek-R1-Distill-Qwen-14B | | | | vs. DeepSeek-R1-Distill-Qwen-32B | | | |
|---|---|---|---|---|---|---|---|---|
| Metric | $\text{Sim}_T$ | | $\text{Sim}_J$ | | $\text{Sim}_T$ | | $\text{Sim}_J$ | |
| Reference Model | Base | Teacher | Base | Teacher | Base | Teacher | Base | Teacher |
| MATH-500 | .724 | **.728** | .777 | **.878** | **.745** | .716 | .790 | **.879** |
| Game of 24 | .354 | **.426** | .852 | **.905** | .294 | **.435** | .834 | **.893** |

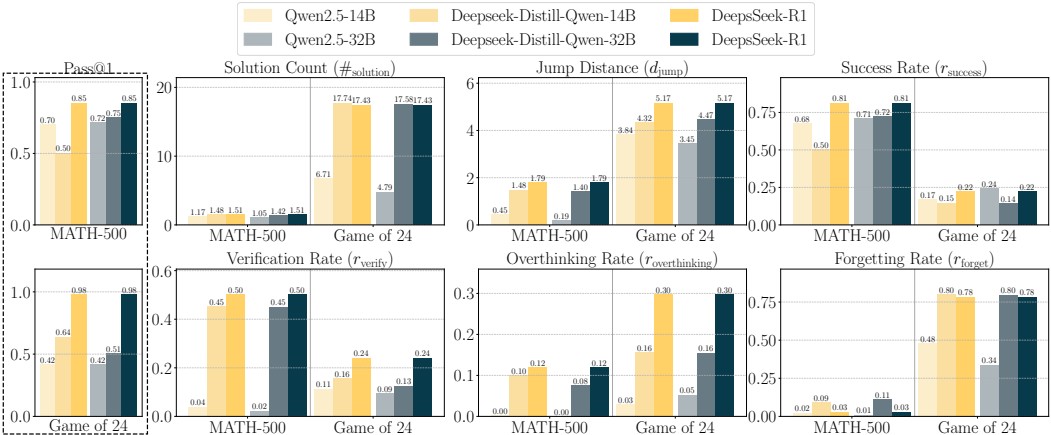

Figure 14: **Comparison of base, teacher, and distilled models across pass@1 and six reasoning evaluation metrics on MATH-500 and Game of 24.** Distilled models inherit reasoning pattern from teacher LRMs. Distilled models exhibit lower success rates than base models but achieve higher pass@1 by generating more and diverse solutions. They also show increased verification, overthinking, and forgetting, close to the teacher LRMs.

### D.5 IMPACT OF DECODING STRATEGY ON REASONING STRUCTURE

Greedy decoding picks the most likely token each step, while temperature sampling adds randomness by adjusting probability distribution. Lower temperatures mimic greedy behavior, while higher temperatures increase sampling randomness by favoring less likely tokens. We test if higher temperatures enhance exploration and impact reasoning, using DeepSeek-R1 and Grok 3 Mini Beta with temperatures $\{0.0, 0.33, 0.66, 1.0\}$. As shown in Fig. 17, we do not observe a consistent pattern in how reasoning behaviors change with temperature.

## E EXTENDED SEC. 6: ENHANCING LLM REASONING WITH REJUMP

### E.1 EXTENSION OF SEC. 6.1: IMPROVING REASONING VIA BEST-OF-N SELECTION WITH REJUMP

**Additional Datasets.** To further demonstrate that the characteristics captured by ReJump can enhance performance, we include additional datasets: Sudoku and ZebraLogic (Lin et al., 2025). Compared to math reasoning tasks, which rely more on a model's fundamental abilities such as applying mathematical knowledge and where high success rates are the only focus (see Sec. 5), these tasks require more sophisticated reasoning strategies, making them more suitable for improvement through test-time adaptation. Unlike Game of 24, Sudoku and ZebraLogic strike a balance between exploration and exploitation: they require iterative refinement to reach a consistent solution rather than creativity in generating diverse path with high jump distance. **(Sudoku)** Sudoku is a logic-based number puzzle. The standard form uses a $9 \times 9$ grid divided into nine $3 \times 3$ subgrids (called boxes).

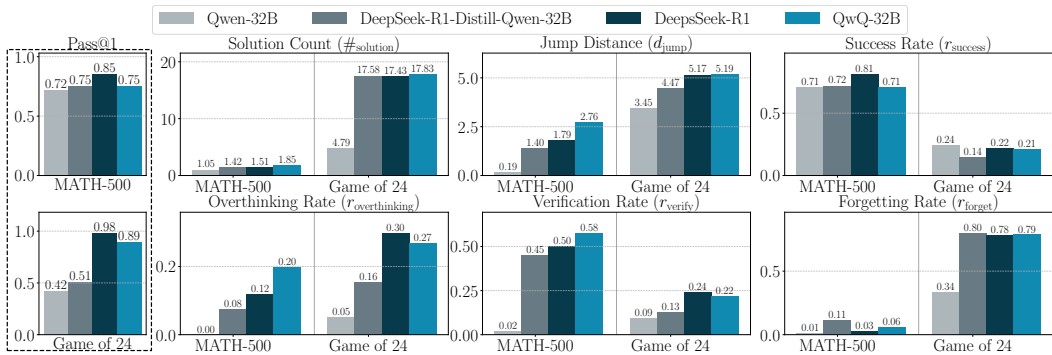

Figure 15: **Comparison of reasoning performance between supervised and RL-trained models.** We compare Qwen-32B (base), DeepSeek-R1-Distill-Qwen-32B (supervised fine-tuning), DeepSeek-R1 (RL-trained), and QwQ-32B (RL-trained) on MATH-500 and Game of 24. RL-trained models exhibit more deliberate reasoning behaviors (e.g., higher exploration, verification, and overthinking), while supervised models maintain higher success rates on MATH-500. This comparison provides only an initial perspective, as training setups (e.g., data and compute) are not fully aligned.

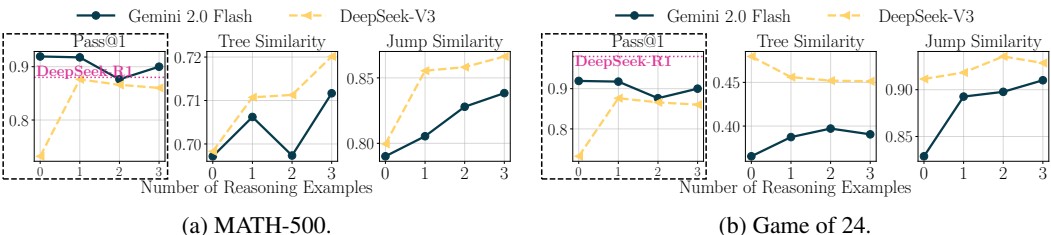

(a) MATH-500.                                    (b) Game of 24.

Figure 16: **Effect of the number of in-context reasoning examples on reasoning behaviors.** We include reasoning examples generated by DeepSeek-R1 in the prompt to guide LLMs (Gemini 2.0 Flash and DeepSeek-V3) to reason more like LRMs. The dashed boxes indicate final accuracy for different number of in-context examples, while the remaining plots show tree similarity and jump similarity to DeepSeek-R1. Neither pass@1 nor tree similarity exhibits a consistent correlation with the number of examples. In contrast, jump similarity increases nearly monotonically, suggesting that fine-grained reasoning actions (e.g., verification, calculation, backtracking) are more influenced by reasoning examples, whereas high-level problem decomposition shows no consistent change.

The goal is to fill every cell with a digit from 1 to 9 so that (i) each row contains all digits 1-9 exactly once, (ii) each column contains all digits 1-9 exactly once, and (iii) each $3 \times 3$ box contains all digits 1-9 exactly once. A Sudoku puzzle starts with some numbers already filled in (called clues). There is only one correct solution if the puzzle is well-formed. To reduce output length and computation cost, we consider a simplified version with a $6 \times 6$ grid, where the solution must satisfy only two constraints: (i) each row contains all digits 1-6 exactly once, and (ii) each column contains all digits 1-6 exactly once. We generate 500 such puzzles. **(ZebraLogic (Lin et al., 2025))** ZebraLogic extends the classic Einstein's Riddle, also known as the Zebra Puzzle. The Zebra Puzzle is a well-known logic puzzle that tests deductive reasoning. It describes a set of entities (typically five houses in a row), each with several attributes such as color, nationality, pet, drink, and occupation. A series of clues defines relationships among these attributes, and the goal is to determine the unique configuration that satisfies all clues. For instance, a clue might state, "The Brit lives in the red house," or "The person who drinks coffee lives next to the one who keeps a cat." Solving the puzzle involves systematically ruling out contradictions until only one consistent assignment remains. ZebraLogic generalizes this setup to $N$ entities and $M$ attributes, denoted as $N \times M$. Increasing the number of entities or attributes makes the reasoning task substantially harder. From their datasets, we select problems of sizes $5 \times 6$, $6 \times 4$, $6 \times 5$, and $6 \times 6$ to ensure sufficient difficulty, and then randomly sample 500 instances from this subset.

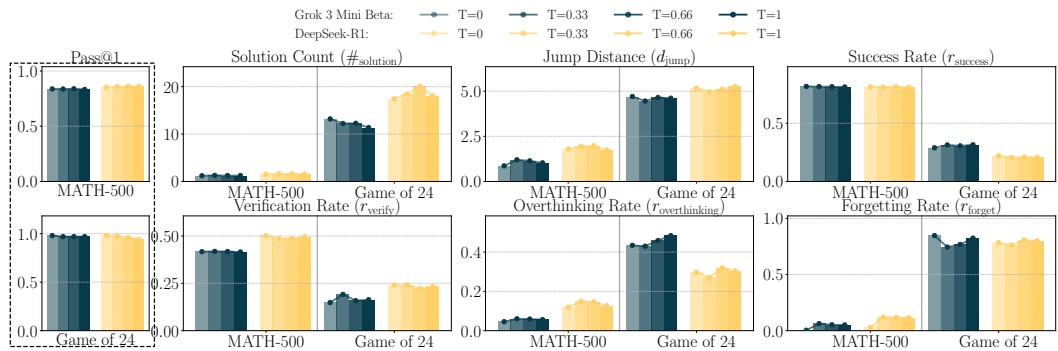

Figure 17: **Impact of decoding temperature on reasoning behaviors across two tasks (MATH-500 and Game of 24) using Grok 3 Mini Beta and DeepSeek-R1.** Each subplot reports one of seven metrics: pass@1, solution count, jump distance, success rate, verification rate, overthinking rate, and forgetting rate. We vary the temperature across $\{0.0, 0.33, 0.66, 1.0\}$ for each model. There is no consistent effect of temperature across models, datasets on reasoning behaviors.

Table 13: **Performance of the majority vote and Best-of-N (BoN) with ReJump on Sudoku and ZebraLogic using Grok 3 Mini Beta.** BoN with ReJump reduces jump distance ($d_{\text{jump}}$) for improving pass@1.

| Task | Method | pass@1 | $d_{\text{jump}}$ |
|---|---|---|---|
| Sudoku | Majority Vote | 0.91 | 6.01 |
| | BoN w. ReJump | **0.96** | **0.71** |
| ZebraLogic | Majority Vote | 0.31 | 12.72 |
| | BoN w. ReJump | **0.38** | **4.48** |

**Results.** We conduct additional experiments using Best-of-N (BoN) with ReJump to further improve reasoning performance on the additional datasets. Based on the heuristics of Sudoku and ZebraLogic, unlike the experiment in Sec. 6.1, where we selected the output with higher jump distance for Game of 24, we instead select the output with the lower jump distance. The results are presented in Tab. 13.

E.2    EXTENSION OF SEC. 6.2: PROMPT SELECTION WITH REJUMP

As discussed in Sec. 5.1, different datasets favor different reasoning strategies; notably, Game of 24 benefits from greater exploration. This aligns with the findings of Stechly et al. (2024), which suggest that effective reasoning requires task-specific prompt designs. To test whether prompting can encourage such exploratory behavior and improve performance, we experiment with four instruction variants inserted into the prompt (Listings 10 to 13) using Phi-4-reasoning-plus on Game of 24 dataset.

```
1 """
2 At each step,try to **make a conceptual leap** rather than a small
      adjustment.
3
4 Do not just continue what you just did – instead, challenge yourself to
      think in a different direction or using a different subset of inputs.
5
6 This approach encourages broader exploration and higher-level reasoning.
7 """
```

Listing 10: Exploration-oriented Instruction A.

```
1 """
2 At each step, instead of thinking locally or making small incremental
      moves,
```

```
3  please consider **making big leaps** in your reasoning.
4
5  Specifically:
6  - Try to **connect concepts or numbers that seem far apart**.
7  - Prefer **longer-range combinations** over adjacent or local steps.
8  - Avoid step-by-step greedy solutions; instead, make bold jumps even if
       they look less obvious at first.
9  - You do not need to go in numerical or structural order.
10 - Think in terms of "maximum novelty".
11
12 Your need to maximize the diversity and distance between steps in your
       reasoning path.
13 """
```

Listing 11: Exploration-oriented Instruction B.

```
1  """
2  Imagine you are exploring a forest, and each tree branch represents a
       line of thought.
3
4  Instead of staying close to your last position, you want to **jump from
       one distant branch to another**, covering as much ground as possible
       with each step.
5
6  At each step, pick the most distant or surprising option you can think of
        - even if it's unconventional. Think globally, not locally.
7  """
```

Listing 12: Exploration-oriented Instruction C.

```
1  """
2  At each step, try to **make a conceptual leap** rather than a small
       adjustment.
3
4  Do not just continue what you just did - instead, challenge yourself to
       think in a different direction or using a different subset of inputs.
5
6  This approach encourages broader exploration and higher-level reasoning.
7  """
```

Listing 13: Exploration-oriented Instruction D.

# F  COMPUTE RESOURCES

All experiments involving models with more than 10B parameters were conducted via API access. Specifically, Gemini models were accessed via the Gemini API[1], DeepSeek-V3 and DeepSeek-R1 via the DeepSeek API[2], Claude models via Anthropic API[3], Qwen-2.5 models via the Qwen API[4], and all other models via the OpenRouter API[5]. The total cost across all APIs was under $2000. For models with fewer than 10B parameters, experiments were run locally on a single NVIDIA H100 GPU. Each experiment on Game of 24 required 510 hours, while experiments on MATH-500 took 1024 hours.

# G  LLM USAGE DISCLOSURE

We used Gemini 2.5 Pro and ChatGPT improve the grammar, clarity, and readability of this manuscript. All LLM-generated content and suggestions were carefully reviewed and edited by

---

[1]https://ai.google.dev/gemini-api/
[2]https://api-docs.deepseek.com/
[3]https://docs.anthropic.com/en/release-notes/api
[4]https://www.alibabacloud.com/help/en/model-studio/use-qwen-by-calling-api
[5]https://openrouter.ai/docs/quickstart

the authors to ensure the final text accurately reflects our scientific contributions and claims. The authors retain full responsibility for the content of this paper.

