# OpenReview forum: "ReJump: A Tree-Jump Representation for Analyzing and Improving LLM Reasoning"
_ICLR.cc/2026/Conference — Submitted to ICLR 2026_

### Official Review · Reviewer_rKX8 · 2025-10-27

**Soundness:** 3
**Presentation:** 3
**Contribution:** 2
**Rating:** 4
**Confidence:** 3

**Summary:**

This paper introduces ReJump, a framework designed to analyze and enhance LLM)reasoning by transforming generated reasoning traces into a structured two-layer representation: a tree layer that captures the hierarchical organization of partial solutions and dependencies, and a jump layer that traces the sequential flow of reasoning steps. Through this representation, the framework enables quantitative analysis of reasoning behaviors such as exploration and exploitation balance, overthinking, and forgetting, facilitating comparison of reasoning dynamics across models, tasks, and settings beyond final accuracy metrics. By revealing reasoning weaknesses independent of task performance, ReJump provides insights for improving model training and inference strategies. Additionally, the framework demonstrates practical utility by improving reasoning quality in applications such as Best-of-N response selection and prompt optimization.

**Strengths:**

This is a work about a proposed framework for converting LLM-generated reasoning traces. This is an interesting direction for refining the understanding on LLM reasoning.

It is an interesting design to define both the two types of similarity measurement, covering both the content semiotics and reasoning jump patterns

A series experimental studied were conducted using the proposed framework, and the authors shared insights/findings from the work

**Weaknesses:**

In the Tree similarity (Sim_T) definition, the authors introduces tree edit distance, a variant of graph edit distance.  How to handle the difference in the nodes, say, corresponding to the newly def

The author should elaborate how to assess the set of metrics proposed for the evaluation. All the proposed metrics look fine, but how to justify they are not overlapping, and jointly cover all key aspects for the evaluation.

It is unclear if the metrics proposed, or the evaluation approach overall, would be sensitive to prompting, for example, if the observed overthinking or increased reasoning would be displayed or not during the LM inference?

The effectiveness of the max flow based approach may be more stronger when the graph. The authors should provide some description/observations on that, so that the value of the work can be better demonstrated.

**Questions:**

n the Tree similarity (Sim_T) definition, the authors introduces tree edit distance, a variant of graph edit distance.  How to handle the difference in the nodes, say, corresponding to the newly def

The author should elaborate how to assess the set of metrics proposed for the evaluation. All the proposed metrics look fine, but how to justify they are not overlapping, and jointly cover all key aspects for the evaluation.

It is a bit unclear if the metrics proposed, or the evaluation approach overall, would be sensitive to prompting, for example, if the observed overthinking or increased reasoning would be displayed or not during the LM inference?

The effectiveness of the max flow based approach may be more stronger when the graph. The authors should provide some description/observations on that, so that the value of the work can be better demonstrated.

---

> ### Author Response · Authors · 2025-11-21
> **Response to Reviewer rKX8 (Part I)**
>
> We thank the reviewer for noting that (i) our approach is interesting, (ii) our experiments are comprehensive, and (iii) our paper provides insights/findings.
>
> ---
>
> > Q1 (Limitation of Tree Similarity): In the Tree similarity (Sim_T) definition, the authors introduces tree edit distance, a variant of graph edit distance. How to handle the difference in the nodes, say, corresponding to the newly def
>
>
> The reviewer is correct that our definition of tree similarity does not consider semantic differences in nodes and instead compares only the tree structure. We adopt this definition for two main reasons:
>
> 1. **Our Focus Is on Structural Similarity Rather Than Full Content Similarity.** The metric focuses on whether two reasonings share similar high-level decomposition patterns, which sheds light on reasoning styles such as exploration, exploitation, and verification, rather than on full content similarity.
>
> 2. **Semantic Node Matching Is Expensive and Difficult.**  One could incorporate semantic equivalence checks by invoking additional LLM inferences to compare whether two nodes represent the same intermediate step. However, matching two trees requires checking many node pairs, which leads to a large number of model calls and becomes prohibitively expensive. Although integrating semantic matching would produce a more precise similarity measure, we found it impractical for this version and leave it for future work.
>
> > Q2 (Independence and Breadth of Evaluation Metrics): The author should elaborate how to assess the set of metrics proposed for the evaluation. All the proposed metrics look fine, but how to justify they are not overlapping, and jointly cover all key aspects for the evaluation.
>
> Our goal in proposing these six metrics is not to obtain mathematically orthogonal statistics, but to capture complementary aspects of reasoning behavior.
>
> * **Redundancy Analysis.** To verify that each metric reflects a distinct aspect rather than overlapping strongly with the others, we conduct an information-theoretic redundancy analysis.
>
>     For each metric $M$, we compute the mutual information between $M$ and all other metrics. We report a normalized redundancy score
>     $$
>         \text{Redundancy}(M) = \frac{I(M ; \text{others})}{H(M)},
>     $$
>     which measures the fraction of $M$'s entropy explained by the remaining metrics. The results for MATH-500 and Game of 24 are shown below, where all entropies and mutual information values are computed by pooling all models' parsed metrics. Across both tasks, all six metrics retain substantial entropy unexplained by the others (low redundancy), indicating that none of them can be reconstructed from the remaining metrics.
>
>     | Dataset | \#$_{\text{solution}}$ | $d_{\text{jump}}$ | $r_{\text{success}}$ | $r_{\text{verify}}$ | $r_{\text{overthinking}}$ | $r_{\text{forget}}$ |
>     |:---:|:---:|:---:|:---:|:---:|:---:|:---:|
>     | MATH-500 | 0.789 | 0.624 | 0.437 | 0.105 | 0.944 | 0.277 |
>     | Game of 24 | 0.731 | 0.761 | 0.882 | 0.687 | 0.840 | 0.840 |
>
> These results are included in Sec. B.3.
>
> * **Positioning Within the Context of Reasoning Metrics.** The six metrics we introduce are not intended to exhaustively cover all dimensions of reasoning behavior. Many additional aspects can be captured, with or without tree-based representations. For example, `Zhou et al. (2025)` proposes metrics that quantify whether the model knows the answer early, how confident it is in intermediate predictions, and how confident it is in its generated thoughts; `Feng et al. (2025)` quantifies failed-step fractions, which is closely related to success rate; and `Xiong et al. (2025)` examines the diversity of alternative reasoning paths, the integration of parallel reasoning threads, and the prevalence of strictly sequential reasoning. Our contribution is to provide a unified tree-based representation that focuses on decomposing reasoning-plan structure and quantifying the associated behaviors. We do not aim to cover all possible dimensions explored in prior work; rather, our metrics are designed to complement existing ones.

---

> > ### Author Response · Authors · 2025-11-21
> > **Response to Reviewer rKX8 (Part II)**
> >
> > > Q3 (Sensitivity of Metrics to Prompt): It is unclear if the metrics proposed, or the evaluation approach overall, would be sensitive to prompting, for example, if the observed overthinking or increased reasoning would be displayed or not during the LM inference?
> >
> > We thank the reviewer for raising this concern. To assess the sensitivity of our metrics to prompting, we perform the following ablation study. Our goal is to modify the prompt while preserving all task requirements needed for the LLM to correctly parse the reasoning. We therefore consider two types of meaning-preserving changes:
> > * **Shuffle**. We shuffle the order of parallel instructions whose order does not affect the semantics. In this experiment, specifically, we permute the order in which we introduce the three transition types (calculation, backtrack, verification).
> > * **Rephrase**. We rephrase the instructions in natural language without changing their semantic content.
> >
> > This yields three prompts: (i) default, (ii) shuffle, and (iii) rephrase. Let $\text{std}\space_{\text{seed}}(M)$ denote the standard deviation of metric $M$ across three runs of the default prompt with different seeds, and $\text{std}\space_{\text{prompt}} (M)$ denote the standard deviation of $M$ across the three prompt variants under a fixed seed. We measure prompt sensitivity using
> >  $$\text{PromptSensitivity}(M) = \frac{\text{std}\space_{\text{prompt}} (M)}{\text{std}\space_{\text{seed}} (M)}.$$
> >
> > We run this experiment using grok-3-mini-beta for its speed and cost efficiency, and we report prompt sensitivity in the table below. The resulting scores are all close to 1, indicating that our method is not sensitive to prompt variation.
> >
> > | Model | \#$_{\text{solution}}$ | $d_{\text{jump}}$ | $r_{\text{success}}$ | $r_{\text{verify}}$ | $r_{\text{overthinking}}$ | $r_{\text{forget}}$ |
> > |:---:|:---:|:---:|:---:|:---:|:---:|:---:|
> > | QwQ-32B | 1.45 | 1.04 | 1.08 | 1.39 | 1.21 | 0.99 |
> > | Phi-4-Reasoning-Plus | 0.82 | 0.93 | 1.25 | 0.96 | 0.95 | 1.08 |
> >
> > These results are included in Sec. B.2.
> >
> > > Q4 (Potential Use of Max Flow for Analyzing Reasoning): The effectiveness of the max flow based approach may be more stronger when the graph. The authors should provide some description/observations on that, so that the value of the work can be better demonstrated.
> >
> > We appreciate the suggestion. The comment is not fully clear to us, so we would be grateful if the reviewer could clarify the specific scenario they had in mind. Based on our current interpretation, we address one plausible meaning below.
> >
> > The only use of max flow that we find applicable here is as an auxiliary metric defined on the ReJump representation rather than as an alternative analysis method. If we treat the initial state as the source, add a unified final state connected to all leaves as the sink, and set all capacities to 1, the resulting max flow increases when the model explores more solution paths and decreases when exploration is limited. This behavior is largely aligned with what our solution count metric already captures.
> >
> > We have not identified a stronger or more informative use of max flow in this context. If the reviewer had other applications in mind, we would appreciate clarification.
> >
> > ---
> >
> > *References*:
> > * `Zhou et al. (2025)` "Landscape of thoughts: Visualizing the reasoning process of large language models." arXiv 2025.
> > * `Feng et al. (2025)` "What characterizes effective reasoning? revisiting length, review, and structure of cot." arXiv 2025.
> > * `Xiong et al. (2025)` "Mapping the Minds of LLMs: A Graph-Based Analysis of Reasoning LLM." arXiv 2025.
> >
> > **Final Note**: Thank you for your detailed comments. If our responses have adequately addressed your concerns, we would greatly appreciate it if you would consider raising your score and supporting our paper’s acceptance. Please feel free to reach out if you have any further questions. We appreciate your time and look forward to your response.

---

> > > ### Comment · Reviewer_rKX8 · 2025-11-25
> > >
> > > Thank you for the detailed rebuttal and clarifications. This message is to acknowledge your response. Based on the additional information provided, I would keep the current score as it is.

---

> > > > ### Author Response · Authors · 2025-11-26
> > > >
> > > > Dear Reviewer rKX8,
> > > >
> > > > Thank you for your response. We would appreciate it if you could clarify which specific concerns from your original review you believe remain unaddressed, as this would help us understand the reasoning behind keeping the current score and further improve our paper. In particular, the comment in Q4 regarding the “max-flow based approach” is unclear to us, so any clarification would be greatly appreciated and would help us address your concern more effectively.
> > > >
> > > > Thanks in Advance,
> > > > Authors of Paper 6806

---

### Official Review · Reviewer_Gh6N · 2025-10-29

**Soundness:** 3
**Presentation:** 3
**Contribution:** 3
**Rating:** 6
**Confidence:** 4

**Summary:**

The paper "ReJump: A Tree-Jump Representation for Analyzing and Improving LLM Reasoning" introduces a novel framework to systematically analyze and compare the reasoning processes of Large Language Models (LLMs), particularly focusing on Large Reasoning Models (LRMs) that generate long-form Chain-of-Thought (CoT) outputs.

At its core, ReJump represents a model's reasoning trace as a tree of intermediate solution steps, augmented with a jump layer that captures non-adjacent transitions—such as backtracking, verification, and recalculation—which are common in human-like reasoning but hard to quantify in raw text. This dual-layer structure enables the extraction of six interpretable metrics: solution count, jump distance (exploration vs. exploitation), success rate, verification rate, overthinking rate, and forgetting rate.

The authors apply ReJump to analyze state-of-the-art LRMs like DeepSeek-R1, QwQ-32B, and Grok 3 Mini Beta across tasks such as MATH-500 and Game of 24. A key finding is that models with similar final accuracy can exhibit fundamentally different reasoning strategies—some favoring broad exploration, others focused exploitation. ReJump also reveals that distilled models inherit reasoning behaviors from their teachers, and that in-context reasoning examples influence action-level behavior more than high-level problem decomposition.
Beyond analysis, ReJump is shown to improve reasoning at test time by enabling Best-of-N selection and prompt selection based on desired reasoning characteristics (e.g., encouraging exploration in Game of 24), without requiring ground-truth labels.

**Strengths:**

1. The approach is both methodologically novel and conceptually comprehensive. The paper introduces ReJump, a dual-layer representation (tree + jump) that models reasoning both structurally and dynamically, offering a novel and interpretable lens on LLM reasoning. Alongside this, the authors define six behavioral metrics (e.g., jump distance, verification rate, overthinking rate) and two similarity metrics that together form a comprehensive toolkit for quantitatively analyzing exploration–exploitation balance and cognitive behaviors such as verification and forgetting.

2. The experiments are solid, covering multiple reasoning models (DeepSeek-R1, Grok 3 Mini Beta, QwQ-32B, Phi-4-Reasoning-Plus, and Claude 3.7 Sonnet) and datasets (MATH-500 and Game of 24), revealing nuanced differences between models that go beyond mere accuracy.

3. A technically sound enhancement built upon meaningful discoveries, ReJump is not only analytical but also practical — it improves reasoning performance through Best-of-N and prompt selection strategies guided by its own metrics, demonstrating clear and measurable gains without the need for supervision.

**Weaknesses:**

1. ReJump extraction requires prompting a large LLM (e.g., Gemini 2.5 Pro), leading to high cost and poor scalability for large-scale or real-time analysis.

2. Experiments focus narrowly on mathematical reasoning and arithmetic-style problems; results on commonsense, coding, or multi-hop reasoning would strengthen generality.

3. Task-specific prompt engineering is still needed to define what a “partial solution node” is; automatic adaptation to new domains (e.g., logic, coding) is not yet solved.

**Questions:**

1. ReJump shows promise as an evaluation and inference-time enhancement tool. Do the authors envision incorporating its metrics as intermediate supervision signals (e.g., via reinforcement learning or reward modeling)?

2. The definitions of overthinking and forgetting are conceptually intuitive, but how are they validated? For example, did human annotators agree with these automatic flags, or were they cross-checked with qualitative examples?

---

> ### Author Response · Authors · 2025-11-21
> **Response to Reviewer Gh6N (Part I)**
>
> We thank the reviewer for the encouraging feedback, especially for recognizing that (i) our approach is novel, comprehensive, interpretable, analytical, practical and sound, (ii) our proposed metrics enable comprehensive and quantitative analysis, (iii) our experiments are solid and comprehensive, and (iv) our approach demonstrates clear and measurable gains.
>
> ---
>
> > Q1: Experiments focus narrowly on mathematical reasoning and arithmetic-style problems; results on commonsense, coding, or multi-hop reasoning would strengthen generality.
>
> As per reviewer's comment, we added two additional logical reasoning datasets: ZebraLogic `(Lin et al. 2025)` and Sodoku.
>
> * **ZebraLogic** `(Lin et al. 2025)` generalizes the classic Zebra Puzzle to an arbitrary $N\times M$ setting, making deductive reasoning significantly harder as entities and attributes increase. We use their $5\times6$, $6\times4$, $6\times5$, and $6\times6$ subsets and sample 500 instances.
>
> * **Sudoku** is a grid-based constraint puzzle. We adopt a compact $6\times6$ variant requiring each row and column to contain digits 1 to 6 exactly once, and generate 500 puzzles.
>
> We further evaluate BoN with ReJump on these two tasks using Grok-3-mini-beta (chosen for its speed and low cost). The full setup is provided in Sec. E.1 of the updated pdf. BoN with ReJump continues to yield improvements on both new datasets:
>
> | Task | Method | Pass@1 |
> |:---:|:---:|:---:|
> | ZebraLogic | Majority Vote | 0.91 |
> | | BoN w. ReJump | **0.96** |
> | Sudoku | Majority Vote | 0.31 |
> | | BoN w. ReJump | **0.38** |
>
> These results support the generality of ReJump beyond arithmetic and mathematical tasks.
>
> > Q2: ReJump shows promise as an evaluation and inference-time enhancement tool. Do the authors envision incorporating its metrics as intermediate supervision signals (e.g., via reinforcement learning or reward modeling)?
>
> Reinforcement Learning (RL) with ReJump as a training signal was too expensive for us because we currently call Gemini API to parse reasoning and produce rewards. Based on the reviewer's comment, we instead use standard RL and applied ReJump only to monitor training progress. We run DAPO on MATH-500 with Qwen-8B and on Game of 24 with Qwen3-1.7B, evaluating each model at four checkpoints corresponding to quarter-length intervals of training. These new experiments are included in Sec. 5.5 of the updated pdf.
>
> The new [Figure 6](https://hackmd.io/_uploads/ryIwaMpxZe.png) summarizes the results. As discussed in Sec. 5.1, MATH-500 favors higher success rates that reflect stronger exploitation, while Game of 24 benefits from both larger jump distance and higher success rates, indicating a need for balanced exploration and exploitation. The curves show that RL progressively shapes the model's reasoning dynamics to match these task-specific demands.
>
> Lastly, using ReJump directly as a reward signal is a natural next step, and we have noted this as a future direction in our limitations section.
>
>
> > Q3: (i: High Cost) ReJump extraction requires prompting a large LLM (e.g., Gemini 2.5 Pro), leading to high cost and poor scalability for large-scale or real-time analysis. (ii: Limited Adaptability) Task-specific prompt engineering is still needed to define what a “partial solution node” is; automatic adaptation to new domains (e.g., logic, coding) is not yet solved.
>
>
> Thanks for highlighting these limitations. We already mentioned both points in the conclusion and mark them as important directions for future work. Here we provide additional clarification on their practical impact.
> 1. **On model size and cost**: our method is not necessarily tied to Gemini-2.5-Pro. A smaller model trained specifically for tree and jump extraction may also be able to handle the task, though we have not tested this. We used Gemini-2.5-Pro simply because it was the most convenient option for demonstrating ReJump. A specialized extractor could further reduce cost, but verifying this would require additional experiments that fall outside the scope of this paper.
> 2. **On adapting ReJump-Extractor to new tasks**: we agree that some task-specific prompt design is needed, but designing such prompts is straightforward and requires only lightweight effort. As additional evidence, we extended our ReJump-Extractor to two logical reasoning tasks in the revision (see our response to Q1), confirming that the adaptation process is practical and easy to apply.

---

> > ### Author Response · Authors · 2025-11-21
> > **Response to Reviewer Gh6N (Part II)**
> >
> > > Q4: The definitions of overthinking and forgetting are conceptually intuitive, but how are they validated? For example, did human annotators agree with these automatic flags, or were they cross-checked with qualitative examples?
> >
> > We thank the reviewer for the insightful question. The correctness of overthinking and forgetting flags depends entirely on the correctness of the extracted ReJump representation. We validated the ReJump-Extractor through both automatic evaluation (by comparing with manually crafted ground truth tree) and human assessment in Sec. 4.1.
> >
> > ---
> >
> > *Reference:*
> >
> > * `Lin et al. (2025)` "Zebralogic: On the scaling limits of llms for logical reasoning." ICML 2025.
> >
> > **Final Note**: We appreciate your thoughtful feedback and are glad that you recognized the strengths of our work. If there are any remaining questions, please do not hesitate to let us know. If our responses have resolved your concerns, we kindly request that you consider increasing your score and support the acceptance of our paper. We look forward to your response.

---

> > > ### Comment · Reviewer_Gh6N · 2025-11-26
> > >
> > > I agree with Reviewer qSFY that the findings in this paper do not reveal any particularly interesting insights. The authors could further polish the manuscript by providing a clearer introduction to the ReJump method and offering more detailed explanations of each metric used. In the methods section, each metric should be defined more clearly and formally using explicit formulas.

---

> > > > ### Author Response · Authors · 2025-12-03
> > > >
> > > > We are happy to report that we provided the formal definition with complete math notation in Sec. B. We avoided heavy notation in the main body to make the method easier for readers to follow. That said, we are fully willing to place more emphasis on the ReJump method in the introduction.

---

### Official Review · Reviewer_qSFY · 2025-11-04

**Soundness:** 3
**Presentation:** 3
**Contribution:** 2
**Rating:** 4
**Confidence:** 3

**Summary:**

The authors aim to better understand the underlying reasoning process of current reasoning LLMs by representing the reasoning process as a walk over a graph, where each node represents a partial solution, and each parent node is a prerequisite of the child node. The authors use Gemini 2.5 Pro to parse and analyze textual LLM outputs into this graph + visitation order representation. Based on this, the authors derive metrics that quantify the model's reasoning behavior. Notably, e.g., the authors identify (1) the level of overthinking by counting the number of extra partial solutions identified by the model, *after* deriving the first correct partial solution. Another example is (2) the level of forgetting, which is derived by the number of times the same node is visited by the model in its reasoning.

The authors apply this method to the MATH-500 and Game-of-24 tasks to derive general insights on LLM reasoning behaviors, including the effect of distillation and R1-style RL training, as well as insights on specific reasoning strategies used for each task.

**Strengths:**

- S1. The graph representation methodology is interesting and the proposed metrics based on this graph representation are intuitive and reasonable.
- S2. The graph representations derived by Gemini 2.5 Pro are verified by human evaluation, and the correlation with human judgement is reasonable.
- S3. Comparison on five main model families and various model variants
- S4. The results quantitatively explain existing insights on *general* reasoning behaviors and show novel *task-specific* insights.

**Weaknesses:**

- W1. The method is complex and costly, given that it is task specific. Applying this analysis to a novel task requires (1) a set of samples for the given task (the authors mention that they use 70 samples from each task), (2) funds for API costs (the authors mention they used approximately $2000 across all experiments), and (3) hand-crafting new prompts to adapt the methodology according to the task (mentioned in the limitations).
- Regarding the findings on *general* reasoning patterns of reasoning LLMs, highlighted in pages 6, 7, 8:
  - W2. The finding on model behaviors are not too surprising, and the quantification of those behaviors offers limited actionable conclusions. Regarding actionability, the study of direct applications of ReJump as inference time strategies in Section 5 does not sufficiently demonstrate strong real-world potential, given the complexity of the method, and limited investigation (of task performance comparison with baselines, cost analysis, etc.).
  - W3. It is unclear if the proposed method (graph-based analysis) had a key role in deriving these findings. E.g., it could be possible to conduct large scale behavioral analysis by simply employing Gemini 2.5 Pro to identify these behaviors from the reasoning paths directly and calculating statistics. How is the proposed method better than this simpler baseline, as a tool for analysis?

I'm willing to increase my score if the authors can provide *strong* arguments against W2 and W3.

Please note that my assessment differs on the paper's *task-specific* insights vs *general* insights. To clarify my assessment: the former are novel and actionable (S4), but hard to apply on new tasks (W1). The later have limited novelty and not very actionable (W2, W3).

**Questions:**

Suggestions on presentation:
- I think that reducing the information density of the figures could make them more effective at delivering key insights. Examples:
  - Font sizes could be larger to improve legibility (they are too small)
  - There are too many models in Figure 4, making it hard to read the radar chart, and recognizing the difference in shape between models. The main figure could include the 2-3 top models that achieve similar performance, to better highlight the point that `even when models achieve similar final performance, their underlying reasoning processes can differ significantly`.
  - In Figure 4 and 5, the metric labels could be made bigger, and written in plain English, similar to those used in the caption: Solution Count, Jump Distance, Success Rate, Inv. Forget Rate, Verification Rate, Inv. Overthinking Rate.
  - Figure 6 is too small and dense.

---

> ### Author Response · Authors · 2025-11-21
> **Response to Reviewer qSFY (Part I)**
>
> We thank the reviewer for the thoughtful comments, in particular for noting that (i) our approach is interesting and verified by human evaluation (ii) our proposed metrics are intuitive and reasonable, (iii) our experiments are comprehensive, and (iv) our results provide quantitative and novel insights.
>
> ---
>
> > Q1 (W1: Hard to Apply to New Tasks). The method is complex and costly, given that it is task specific. Applying this analysis to a novel task requires (1) a set of samples for the given task (the authors mention that they use 70 samples from each task), (2) funds for API costs (the authors mention they used approximately $2000 across all experiments), and (3) hand-crafting new prompts to adapt the methodology according to the task (mentioned in the limitations).
>
> Thanks for the question. A few clarifications help address the concern.
> * **Data**: ReJump does not require a fixed data size. The 70 samples in our human study were only for validating that our ReJump-Extractor, built on Gemini, can reliably parse reasoning traces. In practice, with such highly capable LLMs, the extractor works well, and users can sanity check it on a few examples and proceed without computing accuracy systematically.
> * **Cost**: The reported \\$2000 reflects a broad set of experiments across models and tasks, including inference with non-Gemini models like Claude that were unrelated to the ReJump-Extractor. For a new task with around 100 samples, ReJump-Extractor runs cost under \\$20 when calling the Gemini API to obtain ReJump representations. More importantly, our method is not necessarily tied to Gemini-2.5-Pro. A smaller model trained specifically for tree and jump extraction may also be able to handle the task, though we have not tested this. We used Gemini-2.5-Pro simply because it was the most convenient option for demonstrating ReJump. A specialized extractor could further reduce cost, but verifying this would require additional experiments that fall outside the scope of this paper.
> * **Prompt design**: Each task does need its own prompt, but creating one is strightforward. As additional evidence, we extended our ReJump-Extractor to two logical reasoning tasks (ZebraLogic `(Lin et al. 2025)` and a simplified 6x6 Sudoku) in our revision, confirming that the adaptation process is lightweight and practical. The details of the experiments are provided in Sec. E.1. With these two additional datasets, we also replicate our experiments using Best-of-N (BoN) selection with ReJump and again observe improved performance, as shown below:
>     | Task | Method | Pass@1 |
>     |:---:|:---:|:---:|
>     | ZebraLogic | Majority Vote | 0.91 |
>     | | BoN w. ReJump | **0.96** |
>     | Sudoku | Majority Vote | 0.31 |
>     | | BoN w. ReJump | **0.38** |
>
> In short, applying ReJump to a new task costs about \\$20 or less plus only light prompt engineering. Given the insights it provides into model-specific limitations, directions for improving them, and differences across training algorithms, we believe this overhead is well justified, especially for those aiming to train or refine their own reasoning models.
>
> > Q2 (W2. Limited Novelty and Actionability). The finding on model behaviors are not too surprising, and the quantification of those behaviors offers limited actionable conclusions. Regarding actionability, the study of direct applications of ReJump as inference time strategies in Section 5 (**Authors: the reference here seems inconsistent and may be intended to point to Section 6**) does not sufficiently demonstrate strong real-world potential, given the complexity of the method, and limited investigation (of task performance comparison with baselines, cost analysis, etc.).
>
> We appreciate the reviewer highlighting this confusion and would like to clarify our intent.
>
> **The focus of our work is ReJump representation itself, rather than specific findings or a fully-developed reasoning-improvement method**: Our representation enables quantitative analysis of different reasoning behaviors such as exploration-exploitation tradeoff, overthinking, forgetting, etc.  The analysis and improvement experiments are intended not to claim surprising phenomena or to propose a fully developed inference-time method, but to demonstrate what the ReJump representation enables: (i) making deliberate reasoning behaviors measurable and comparable, which can provide valuable insights for practitioners building reasoning models, and (ii) showing that the representation can be used in simple ways to improve models beyond analysis alone. While a deeper investigation of actionability, including stronger baselines and a detailed cost–performance evaluation, is beyond our current scope, we agree that it is a interesting future direction.

---

> > ### Author Response · Authors · 2025-11-21
> > **Response to Reviewer qSFY (Part II)**
> >
> > > Q3 (W3. Comparison to Simpler Baselines). It is unclear if the proposed method (graph-based analysis) had a key role in deriving these findings. E.g., it could be possible to conduct large scale behavioral analysis by simply employing Gemini 2.5 Pro to identify these behaviors from the reasoning paths directly and calculating statistics. How is the proposed method better than this simpler baseline, as a tool for analysis?
> >
> > We thank the reviewer for this great question. To examine whether one can simply ask Gemini-2.5-Pro to directly obtain the values of the metrics, we ran the following experiments.
> >
> > * For metrics not defined on the graph (e.g., #$\space_{\text{solution}}$, $r_{\text{success}}$, $r_{\text{forget}}$), we use the ground truth datasets created for evaluating ReJump-Extractor (see Sec. 4.1). Each sample is annotated with its correct metric value. We then compare two approaches: (i) directly asking the model to output the metric, and (ii) first extracting the ReJump representation and then computing the metric from that representation. The results below indicate that using ReJump-Extractor produces more accurate analyses compared to the simpler baseline.
> >
> >     | | \#$_{\text{solution}}$ (MAE $\downarrow$) | $r_{\text{success}}$ (MAE $\downarrow$) | $r_{\text{forget}}$ (Accuracy $\uparrow$) |
> >     |:---:|:---:|:---:|:---:|
> >     | Direct | 2.12 | 0.11 | 0.87 |
> >     | ReJump-Extractor | **0.62** | **0.08** | **0.89** |
> >
> > * For metrics defined specifically on the graph (i.e., $d_{\text{jump}}$ and $r_{\text{verify}}$), it is difficult to obtain them by directly querying Gemini-2.5-Pro. Since jump distance measures exploration, we instead prompt the model to classify exploration level using three examples. We then repeat the experiment of Sec. 6.1, where each prompt has three outputs and we apply Best-of-N (BoN) to choose the output with highest predicted exploration level. The table below shows that BoN with ReJump consistently achieves the best performance, further supporting that ReJump-Extractor enables more accurate analysis compared to the simpler baseline.
> >     | Model | Majority Vote | BoN w. Direct | BoN w. ReJump |
> >     |:---:|:---:|:---:|:---:|
> >     | Phi-4-reasoning-plus | 0.77 | 0.77 | **0.84** |
> >     | QwQ-32B | 0.76 | **0.82** | **0.82** |
> >
> > These results are included in Sec. B.4.
> >
> > > Q4: Suggestions on presentation.
> >
> > Thank you for the detailed presentation suggestions.
> >
> > * Regarding the comment "there are too many models in Figure 4, making it hard to read the radar chart, and recognizing the difference in shape between models. The main figure could include the 2-3 top models that achieve similar performance, to better highlight the point that even when models achieve similar final performance, their underlying reasoning processes can differ significantly.":
> >
> >     This is an excellent suggestion. However, because the later sections (Sec. 6) focus on improving the relatively weaker models (QwQ-32B and Phi-4-Reasoning-Plus), we decided to keep these models in the main figure for consistency. To improve readability, we have added an additional figure in Appendix D.1 that includes only the three top-performing models with similar final accuracy.
> >
> > For the remaining presentation-related comments, we agree and have made the corresponding revisions, including increasing the font size in Figures 3, 4, 5, and 6.
> >
> > ---
> >
> > *Reference*:
> >
> > * `Lin et al. (2025)` "Zebralogic: On the scaling limits of llms for logical reasoning." ICML 2025.
> >
> > **Final Note**: Thank you for the thoughtful feedback and for `explicitly noting your openness to revisiting the score`. If you feel our revisions resolve your concerns, we would greatly appreciate your consideration in updating the score. Thank you again for your time, and we look forward to your response.

---

> ### Comment · Reviewer_qSFY · 2025-11-26
>
> Thank you for addressing my comments and for the additional results.
>
> W1 has been partially resolved but my concerns with W2 and W3 remain.
>
> Regarding W1, while the author rebuttal explains that the cost of applying this analysis to new tasks is not significant, **concerns remain regarding the reliability of the analysis on new tasks**. The authors show that the metrics derived by ReJump are reliable, i.e., align with ground-truth or human judgement, on Game-of-24 and Math-500 in Section 4.1. This does not ensure the reliability of ReJump on novel tasks with novel prompts. To ensure reliability, this analysis will need to be repeated for each task, which remains a major cost factor.
>
> The rebuttal of W2 is incoherent. The authors explain that "The focus of our work is ReJump representation itself, rather than specific findings or a fully-developed reasoning-improvement method", but go on to mention that "The analysis and improvement experiments are intended ... to demonstrate what the ReJump representation enables:... provide **valuable insights for practitioners**... the representation can be used in simple ways to **improve models beyond analysis**".
>
> Based on the metrics shown in the rebuttal of W3, it seems that the findings from the paper could have also been revealed through directly prompting Gemini-2.5-Pro without the use of a tree representation. The metrics regarding r_success and r_forget are similar. While the MAE of #solution from direct prompting is 3x higher than that from the proposed method, it is unclear if the higher MAE would lead to different conclusions when using direct prompting for #solution analysis such as those in Figures 3, 4, 6. The high scale of #solution in Figure 6 in Game-of-24 (ranging between 6 and 8) suggests otherwise. As noted in W2, the BoN with ReJump methodology does not strongly demonstrate real-world potential, invalidating the comparison of the BoN results.
>
> Based on the above, I maintain my score.

---

> > ### Author Response · Authors · 2025-12-03
> >
> > Thank you for the follow-up and for clarifying the remaining concerns. We appreciate the opportunity to further explain the focus and contributions of our work.
> >
> > > W2: The rebuttal of W2 is incoherent. The authors explain that "The focus of our work is ReJump representation itself, rather than specific findings or a fully-developed reasoning-improvement method", but go on to mention that "The analysis and improvement experiments are intended ... to demonstrate what the ReJump representation enables:... provide valuable insights for practitioners... the representation can be used in simple ways to improve models beyond analysis".
> >
> > Sorry for the confusion. Let us try to make it more clear. Our focus is on developing the ReJump representation for analyzing reasoning traces. The representation itself is the core contribution. The small improvement experiments are **further extensions** meant only to show what the representation can **additionally** enable, not to introduce or evaluate a full reasoning-improvement method. Because improvement is not the focus, we do not aim to provide full cost analyses or comparisons with methods specifically designed to optimize final accuracy.
> >
> > When we say the representation “provides valuable insights for practitioners building reasoning models,” we refer to its ability to make deliberate reasoning behaviors measurable and comparable across models, tasks, training setups, and decoding strategies. This analytic value does not depend on turning ReJump into a complete inference-time strategy. We hope this clarification resolves the concern.
> >
> > > W3: Based on the metrics shown in the rebuttal of W3, it seems that the findings from the paper could have also been revealed through directly prompting Gemini-2.5-Pro without the use of a tree representation. The metrics regarding r_success and r_forget are similar. While the MAE of #solution from direct prompting is 3x higher than that from the proposed method, it is unclear if the higher MAE would lead to different conclusions when using direct prompting for #solution analysis such as those in Figures 3, 4, 6. The high scale of #solution in Figure 6 in Game-of-24 (ranging between 6 and 8) suggests otherwise. As noted in W2, the BoN with ReJump methodology does not strongly demonstrate real-world potential, invalidating the comparison of the BoN results.
> >
> > Regarding W3, we respectfully disagree that our findings could be reproduced through direct prompting alone, for three reasons.
> >
> > 1. **Irreplaceable information**: Metrics such as jump distance, which are central to capturing the exploration–exploitation tradeoff, are not even definable without the ReJump representation. This structural information is essential to several key findings in our analysis.
> > 2. **Higher Precision**: (i) For metrics that are not defined without ReJump, such as jump distance and verification rate, even if one tries to approximate the corresponding characteristics using direct prompting, the precision will be much lower. For instance, for jump distance, which aims to capture exploration-explotation tradeoff, direct prompting can only give coarse labels (e.g., “high exploration” and "low exploration"), while ReJump provides quantitative evaluation. Our second experiment to Q3 show that jump distance indeed offer meaningfully more precise measurements. (ii) For the other metrics, we also observe that the ReJump representation provides more precise evaluation from our first experiment for Q3 in our rebuttal.
> > 3. **Lower Cost at Scale**: Even if all metrics could be obtained via direct prompting precisely, ReJump is more efficient. For each reasoning trace, our method only requires using ReJump-Extractor once to obtain a single ReJump representation, and then compute all metrics at once, whereas direct prompting requires separate prompts enginerring for each metric. Based on the reviewer's suggestion that reliability must be human-verified, our method requires only one verification of extraction quality, and directly prompting LLM for obtaining the metrics repeated human studies for six times. This is a substantial difference in cost and scalability.

---

### Author Response · Authors · 2025-11-21
**To AC and All Reviewers (Part I)**

We thank the reviewers for providing valuable suggestions that helped us improve our paper. We are particularly encouraged that the reviewers found that (i) our approach is sound (`R-Gh6N`), interesting (`R-qSFY`, `R-rKX8`), novel (`R-Gh6N`), interpretable (`R-Gh6N`), analytical (`R-Gh6N`), practical (`R-Gh6N`), verified by human evaluation (`R-qSFY`), and demonstrates clear gains (`R-Gh6N`);  (ii) our proposed metrics are intuitive (`R-qSFY`), reasonable (`R-qSFY`), and enable comprehensive (`R-Gh6N`) and quantitative (`R-Gh6N`) analysis; (iii) our experiments are solid (`R-Gh6N`) and comprehensive (`R-qSFY`, `R-Gh6N`, `R-rKX8`); and (iv) our paper provides quantitative (`R-qSFY`) and novel (`R-Gh6N`) insights and findings (`R-rKX8`).

As for the concerns/questions raised, we believe that we successfully addressed every single one, as replied to each reviewer. We integrated most of the answers and new results in the newly updated version (attached).

Here we first restate the scope of the paper. **Our goal is to introduce and analyze the ReJump representation itself rather than to claim surprising findings or present a complete reasoning-improvement method.** With this clarified, we identify three key concerns raised by the reviewers. To address them, we conducted additional experiments, and the key concerns and our responses are summarized below.

---

### Key Concern 1: Adaptation to Other Tasks

Two specific questions under this concern are: (i) that applying ReJump to other tasks may appear to require substantial effort, and (ii) that the experiments focus primarily on mathematical and arithmetic-style problems, and would benefit from additional tasks. Regarding (i), we argue that adapting ReJump to new tasks only requires lightweight prompt engineering. To demonstrate this and address (ii), we added two additional logical reasoning datasets: ZebraLogic `(Lin et al. 2025)` and Sodoku.

* **ZebraLogic** `(Lin et al. 2025)` generalizes the classic Zebra Puzzle to an arbitrary $N\times M$ setting, making deductive reasoning significantly harder as entities and attributes increase. We use their $5\times6$, $6\times4$, $6\times5$, and $6\times6$ subsets and sample 500 instances.

* **Sudoku** is a grid-based constraint puzzle. We adopt a compact $6\times6$ variant requiring each row and column to contain digits 1 to 6 exactly once, and generate 500 puzzles.

We further evaluate BoN with ReJump on these two tasks using Grok-3-mini-beta (chosen for its speed and low cost). The full setup is provided in Sec. E.1 of the updated pdf. BoN with ReJump continues to yield improvements on both new datasets:

| Task | Method | Pass@1 |
|:---:|:---:|:---:|
| ZebraLogic | Majority Vote | 0.91 |
| | BoN w. ReJump | **0.96** |
| Sudoku | Majority Vote | 0.31 |
| | BoN w. ReJump | **0.38** |


### Key Concern 2: Benefit of ReJump over Simpler LLM-based Analysis

This concern stems from the complexity of the representation itself and whether ReJump-based analysis offers real advantages over a simpler baseline, such as prompting an LLM to directly output the metric values without extracting a ReJump representation. To test this, we ran the following experiments.

* For metrics not defined on the graph (e.g., $\#_{\text{solution}}$, $r_{\text{success}}$, $r_{\text{forget}}$), we use the ground truth datasets created for evaluating ReJump-Extractor (see Sec. 4.1). Each sample is annotated with its correct metric value. We then compare two approaches: (i) directly asking the model to output the metric, and (ii) first extracting the ReJump representation and then computing the metric from that representation. The results below indicate that using ReJump-Extractor produces more accurate analyses compared to the simpler baseline.

    | | #$_{\text{solution}}$ (MAE $\downarrow$) | $r_{\text{success}}$ (MAE $\downarrow$) | $r_{\text{forget}}$ (Accuracy $\uparrow$) |
    |:---:|:---:|:---:|:---:|
    | Direct | 2.12 | 0.11 | 0.87 |
    | ReJump-Extractor | **0.62** | **0.08** | **0.89** |

* For metrics defined specifically on the graph (i.e., $d_{\text{jump}}$ and $r_{\text{verify}}$), it is difficult to obtain them by directly querying Gemini-2.5-Pro. Since jump distance measures exploration, we instead prompt the model to classify exploration level using three examples. We then repeat the experiment of Sec. 6.1, where each prompt has three outputs and we apply Best-of-N (BoN) to choose the output with highest predicted exploration level. The table below shows that BoN with ReJump consistently achieves the best performance, further supporting that ReJump-Extractor enables more accurate analysis compared to the simpler baseline.
    | Model | Majority Vote | BoN w. Direct | BoN w. ReJump |
    |:---:|:---:|:---:|:---:|
    | Phi-4-reasoning-plus | 0.77 | 0.77 | **0.84** |
    | QwQ-32B | 0.76 | **0.82** | **0.82** |

These results are included in Sec. B.4.

---

> ### Author Response · Authors · 2025-11-21
> **To AC and All Reviewers (Part II)**
>
> ### Key Concern 3: Redundancy Analysis of Proposed Metrics
>
> To verify that each metric reflects a distinct aspect rather than overlapping strongly with the others, we conduct an information-theoretic redundancy analysis.
>
> For each metric $M$, we compute the mutual information between $M$ and all other metrics. We report a normalized redundancy score
> $$
>     \text{Redundancy}(M) = \frac{I(M ; \text{others})}{H(M)},
> $$
> which measures the fraction of $M$'s entropy explained by the remaining metrics. The results for MATH-500 and Game of 24 are shown below, where all entropies and mutual information values are computed by pooling all models' parsed metrics. Across both tasks, all six metrics retain substantial entropy unexplained by the others (low redundancy), indicating that none of them can be reconstructed from the remaining metrics.
>
> | Dataset | \#$_{\text{solution}}$ | $d_{\text{jump}}$ | $r_{\text{success}}$ | $r_{\text{verify}}$ | $r_{\text{overthinking}}$ | $r_{\text{forget}}$ |
> |:---:|:---:|:---:|:---:|:---:|:---:|:---:|
> | MATH-500 | 0.789 | 0.624 | 0.437 | 0.105 | 0.944 | 0.277 |
> | Game of 24 | 0.731 | 0.761 | 0.882 | 0.687 | 0.840 | 0.840 |
>
> These results are included in Sec. B.3.
>
> ### Other New Experiments
>
> * Monitoring behavior changes during RL training with ReJump: see `R-Gh6N` Q2, Sec. 5.5 in the updated pdf
> * Sensitivity of metric values to prompt variations: see `R-rKX8` Q3, Sec. B.2 in the updated pdf
>
> ---
>
> *Reference*:
>
> * `Lin et al. (2025)` "Zebralogic: On the scaling limits of llms for logical reasoning." ICML 2025.

---

### Author Response · Authors · 2025-12-03
**Summary of the Discussion Period**

Dear AC,

Thank you for taking the time to read our discussion. Below is a brief summary of what occurred during the discussion period to help clarify the situation.

* For Reviewer qSFY, who raised three weaknesses (W1, W2, and W3) and explicitly noted that addressing W2 and W3 *could lead to a higher score*, we did our best to respond to all concerns. The reviewer was only partially convinced, so we provided a second response to ensure each point was thoroughly addressed.

* For Reviewer Gh6N, who initially gave a score of 6 and maintained it after the rebuttal, they offered a suggestion regarding the presentation of our method. We clarified the issue and committed to placing more emphasis on ReJump in the introduction, which fully resolves their concern.

* For Reviewer rKX8, we were concerned about the overall quality of the review. The reviewer raised an issue about a “max flow based approach,” which does not appear anywhere in our paper and was introduced without context, making it unclear how it relates to our work. Although we attempted to infer the reviewer’s intent, the point remained ambiguous. Additionally, the Weaknesses section was copied directly into the Questions section, typos included. During the discussion, the reviewer gave only a brief acknowledgment of our rebuttal and decided to keep the current score, without providing any explanation or addressing our explicit request for clarification. This suggests they may not have engaged closely with our response.

We hope this summary helps clarify the state of the discussion.

Best Regards,

On behalf of the authors

---

### Meta-Review · Area_Chair_ThoC · 2025-12-10

**Summary:**

The reviewers were split 1 positive and 2 negative. Major concerns included the high cost of the method (due to prompting LLM APIs), whether the method generalizes well or not to new tasks, and whether the insights are sufficiently interesting.

**Reviewer Concerns:**

High cost of prompting APIs
- This was not resolved by the rebuttal

Generalization to other tasks; insights are not interesting enough
- Authors and reviewer qSFY engaged in substantial discussion, but reviewer qSFY remained negative on their concerns.

**Reviewer Scores:**

qSFY: I don't think they would have changed their score, as they seemed unconvinced during their discussion

Gh6N: they were already positive, but replied that they had similar unresolved concerns as qSFY (whether the insights are interesting or not). I doubt they would change their score.

rKX8: The other two reviews shared similar unresolved concerns, and I don't think this reviewer would have changed their score.

Overall, the paper has concerns on generalizability, cost, and insightfulness that were not addressed by the rebuttal. On the other hand, it also shows good task-specific performance, substantial experiments, and a comprehensive appendix.

I appreciate the effort that the authors have put into their rebuttal. LLM reasoning is a highly competitive topic this year, and I think this paper would benefit from another round of revision before submission. In the revised paper, the authors should consider explaining the unresolved limitations of their work, and also toning down the claims/insights flagged by qSFY.

---

### Decision · Program_Chairs · 2026-01-26

Reject